


# Experimental and diagnostic protocol for the physical component of the CMIP6 Ocean Model Intercomparison Project (OMIP)

Stephen M. Griffies [1], Gokhan Danabasoglu [2], Paul J. Durack [3], Alistair J. Adcroft [1], V. Balaji [1], Claus W. Böning [4], Eric P. Chassignet [5], Enrique Curchitser [6], Julie Deshayes [7], Helge Drange [8], Baylor Fox-Kemper [9], Peter J. Gleckler [3], Jonathan M. Gregory [10], Helmuth Haak [11], Robert W. Hallberg [1], Helene T. Hewitt [12], David M. Holland [13], Tatiana Ilyina [11], Johann H. Jungclaus [11], Yoshiki Komuro [14], John P. Krasting [1], William G. Large [2], Simon J. Marsland [15], Simona Masina [16], Trevor J. McDougall [17], A. J. George Nurser [18], James C. Orr [19], Anna Pirani [20], Fangli Qiao [21], Ronald J. Stouffer [1], Karl E. Taylor [3], Anne Marie Treguier [22], Hiroyuki Tsujino [23], Petteri Uotila [24], Maria Valdivieso [25], Michael Winton [1], and Stephen G. Yeager [2]

[1]NOAA Geophysical Fluid Dynamics Laboratory, Princeton, New Jersey, USA
[2]National Center for Atmospheric Research, Boulder, Colorado, USA
[3]Program for Climate Model Diagnosis and Intercomparison, Lawrence Livermore National Laboratory, Livermore, California, USA
[4]GEOMAR Helmholtz Centre for Ocean Research Kiel, Germany
[5]Center for Ocean-Atmospheric Prediction Studies (COAPS), Florida State University, Tallahassee, FL, USA
[6]Rutgers University, New Brunswick, New Jersey, USA
[7]Sorbonne Universités (UPMC, Univ Paris 06-CNRS-IRD-MNHN), LOCEAN Laboratory, Paris, France
[8]Geophysical Institute, University of Bergen, Norway
[9]Department of Earth, Environmental, and Planetary Sciences (DEEPS), Brown University, USA
[10]Met Office Hadley Centre and University of Reading, UK
[11]Max Planck Institute for Meteorology Bundesstrasse 53, 20146 Hamburg, Germany
[12]Met Office Hadley Centre, Exeter, UK
[13]New York University, New York, USA
[14]Japan Agency for Marine-Earth Science and Technology, Kanagawa, Japan
[15]CSIRO Oceans and Atmosphere, Aspendale, Victoria, Australia
[16]Centro Euromediterraneo sui Cambiamenti Climatici, and Istituto Nazionale di Geofisica e Vulcanologia, Bologna, Italy
[17]University of New South Wales, Sydney, Australia
[18]National Oceanography Centre Southampton (NOCS), Southampton, UK
[19]IPSL/LSCE, UMR8212, CNRS-CEA-UVSQ, Gif sur Yvette, France
[20]Université Paris Saclay, France and Abdus Salam Institute for Theoretical Physics, Italy
[21]First Institute of Oceanography, State Oceanic Administration, China
[22]Laboratoire d'Oceanographie Physique et Spatiale, Ifremer, Plouzane, France
[23]Meteorological Research Institute (MRI), Japan Meteorological Agency, Tsukuba, Japan
[24]Finnish Meteorological Institute, Helsinki, Finland
[25]University of Reading, Reading, UK

*Correspondence to:* Stephen M. Griffies (Stephen.Griffies@noaa.gov)

**Draft from April 12, 2016**





**Abstract.** The Ocean Model Intercomparison Project (OMIP) aims to provide a framework for evaluating, understanding, and improving the ocean and sea-ice components of global climate and earth system models contributing to the Coupled Model Intercomparison Project Phase 6 (CMIP6). OMIP addresses these aims in two complementary manners: (A) by providing an experimental protocol for global ocean/sea-ice models run with a prescribed atmospheric forcing, (B) by providing a protocol for ocean diagnostics to be saved as part of CMIP6. We focus here on the physical component of OMIP, with a companion paper (Orr et al., 2016) offering details for the inert chemistry and interactive biogeochemistry. The physical portion of the OMIP experimental protocol follows that of the interannual Coordinated Ocean-ice Reference Experiments (CORE-II). Since 2009, CORE-I (Normal Year Forcing) and CORE-II have become the standard method to evaluate global ocean/sea-ice simulations and to examine mechanisms for forced ocean climate variability. The OMIP diagnostic protocol is relevant for any ocean model component of CMIP6, including the DECK (Diagnostic, Evaluation and Characterization of Klima experiments), historical simulations, FAFMIP (Flux Anomaly Forced MIP), C4MIP (Coupled Carbon Cycle Climate MIP), DAMIP (Detection and Attribution MIP), DCPP (Decadal Climate Prediction Project), ScenarioMIP (Scenario MIP), as well as the ocean-sea ice OMIP simulations. The bulk of this paper offers scientific rationale for saving these diagnostics.

## Contents



## 1   OMIP and this paper

We document here the CMIP6 Ocean Model Intercomparison Project (OMIP). The key aims of OMIP are to provide a framework for evaluating, understanding, and improving the ocean and sea-ice components of global climate and earth system
models contributing to CMIP6. OMIP addresses these aims in two complementary ways.

1. OMIP AS AN EXPERIMENTAL MIP: OMIP provides an experimental protocol for global ocean/sea-ice simulations forced with common atmospheric data sets. OMIP ocean/sea-ice simulations include physical, inert chemical, and interactive biogeochemical components, thus bringing together a broad community of ocean and climate scientists making use of global ocean/sea-ice models. OMIP offers an opportunity for contributions from a wide number of groups capable
of running global ocean/sea-ice models.

2. OMIP AS A DIAGNOSTICS MIP: OMIP provides a diagnostics protocol to coordinate and rationalize ocean diagnostics for CMIP6 simulations that include an ocean component, including the following.

   (a) CMIP6 DECK (Diagnostic, Evaluation and Characterization of Klima experiments) and historical simulations (Eyring et al., 2015)

(b) FAFMIP (Flux Anomaly Forced MIP) (Gregory et al., 2016);

   (c) C4MIP (Coupled Carbon Cycle Climate MIP) (Jones et al., 2016)

   (d) DAMIP (Detection and Attribution MIP) (Gillett and Shiogama, 2016)

   (e) DCPP (Decadal Climate Prediction Project) (Boer et al., 2016)

   (f) ScenarioMIP (Scenario MIP) (O'Neill et al., 2016)

(g) OMIP ocean/sea-ice simulations (this paper).

In this paper, we detail elements of the physical portion of OMIP, including the experimental protocol and diagnostics protocol. Details for the chemical and biogeochemical portions of OMIP are provided in the companion paper by Orr et al. (2016).



## 1.1 A mandate based on enhanced observational and modelling capabilities

Observational oceanography continues to experience a growth in measurement capability that supports critical insights into the changing earth climate system. This growth largely results from the Argo Program (Riser et al., 2016). Argo has revolutionized physical oceanography by providing comprehensive temperature and salinity profiles since 2005 for the upper 2000 meters with near global coverage. Most centrally for studies of the earth's climate, Argo has enabled revised assessments of ocean heat content (e.g., Roemmich et al., 2012, 2015; von Schuckmann et al., 2016), documenting the ongoing and unabated ocean warming. These measurements also point to persistent salinity changes (e.g., Hosoda et al., 2009; Durack and Wijffels, 2010; Helm et al., 2010; Skliris et al., 2014), hypothesized to result from water cycle amplification (Stott et al., 2008; Durack et al., 2012; Pierce et al., 2012; Terray et al., 2012). Measurements furthermore suggest that the Southern Hemisphere is responsible for 67-98% of the global ocean heating during 2006 to 2013 (Roemmich et al., 2015).

Additional deep ocean measurements and analysis (e.g., Purkey and Johnson, 2010, 2012, 2013; Kouketsu et al., 2011) reflect changes in the global energy budget as well as ongoing sustained contributions to sea level rise. Recent augmentation of space-borne ocean observations include ocean salinity, thanks to the SMOS (Berger et al., 2002), Aquarius (Lagerloef et al., 2008) and SMAP (Piepmeier et al., 2015) satellites, which complement the longer-standing ocean surface temperature measurements.

Global ocean/sea-ice and climate models are powerful tools to help mechanistically interpret ocean measurements, and in some cases to identify key limitations of the measurements (Durack et al., 2014a). Conversely, ocean measurements, particularly those maintained over many decades, offer the means to assess simulation fidelity. As noted by Durack et al. (2016), we cannot presume measurements will continue indefinitely. It is therefore critical that we further the relationship between modelling and observations, as doing so supports both. A grounding in ocean and climate science, in the midst of enhanced capabilities in both observations and modelling, enables the Ocean Model Intercomparison Project.

## 1.2 Uses of global ocean/sea-ice models

Although the bulk of CMIP involves coupled climate and earth system models, it is important to complement these more comprehensive systems with a hierarchy of model configurations aiming to uncover mechanisms and understand biases. Global ocean/sea-ice models provide a tool for doing so, in a manner motivated by similar efforts in other climate components, particularly the Atmosphere Model Intercomparison Project (AMIP) (Gates, 1993). Specifically, OMIP experiments and diagnostics provide a framework for the following types of research studies:

1. To investigate oceanic physical, chemical, and biogeochemical mechanisms that drive seasonal, interannual, and decadal variability;

2. To assess and understand biases in the ocean/sea-ice component of coupled climate models;

3. To attribute ocean climate variations to boundary forced (including volcanoes) versus natural (without volcanoes);

4. To evaluate robustness of mechanisms across models and forcing data sets;





5. To bridge observations and modelling by providing a complement to ocean reanalysis from data assimilation (Karspeck et al., 2015);

6. To provide consistent ocean and sea-ice states useful for initialization of decadal predictions (Yeager et al., 2012).

Further specific examples of recent global ocean/sea-ice studies are noted in Section 2 where we discuss the Coordinated Ocean-ice Reference Experiments (CORE), the predecessor to OMIP.

### 1.3 Content of this paper

We start the main portion of this paper in Section 2 by defining the OMIP experimental protocol for the physical components of ocean/sea-ice simulations. Chemical and biogeochemical protocols are detailed in Orr et al. (2016). The following sections define the OMIP diagnostics protocol.

In Section 3 we provide an overview of OMIP as a diagnostics MIP, and summarize many of the related issues. In particular, we raise questions about native grid versus spherical grid output for diagnosed fields. The remaining sections then detail the various diagnostics. In Section 4 we describe the static fields and functions to define the particular ocean model configuration. In Section 5 we provide details for the scalar fields such as tracers, and in Section 6 we discuss the requested components of vector fields such as velocity and transport. In Section 7 we describe the requested mass transports through a suite of predefined straits and throughflows. We describe the requested boundary fluxes of mass, heat, salt, and momentum in Section 8. In Section 9 we formulate the diagnostics for examining three-dimensional heat and salt budgets. Finally, in Sections 10 and 11 we detail the requests for vertical and lateral subgrid scale parameters. We close the main portion of the paper with a brief summary in Section 12.

We offer a number of appendices. In Appendix A we detail grid cell volume and area; in Appendix B we discuss spatial sampling; in Appendix C we make suggestions regarding data precision; in Appendix D, we summarize elements of seawater thermodynamics of relevance for sampling the temperature and salinity fields; and in Appendix E we show that the evolution of ocean heat content is invariant when changing temperature scales. Finally, in Appendix F we summarize elements of a finite volume formulation of the tracer equation.

## 2 OMIP as a global ocean/sea-ice MIP

The physical component of OMIP simulations follows the protocol of the Coordinated Ocean-ice Reference Experiments (CORE) (Griffies et al., 2009b, 2012; Danabasoglu et al., 2014) interannually varying simulations (CORE-II). The interannual CORE-II protocol shares much with the Normal Year Forcing protocol of CORE-I (Griffies et al., 2009b).[1] However, CORE-I makes use of an idealized repeating annual cycle, whereas CORE-II includes interannual variations. The interannual variations allow CORE-II simulations to be directly compared to observation-based measures, especially on interannual to decadal time scales.

---

[1]CORE-I simulations, based on CMIP3-era ocean/sea-ice models, are documented in Griffies et al. (2009b).





CORE-II makes use of the interannually varying atmospheric state of Large and Yeager (2009) to force physical ocean fields. The data covers the 62-year period from 1948-2009, and it is collaboratively supported by the U.S. National Center for Atmospheric Research (NCAR) and the NOAA Geophysical Fluid Dynamics Laboratory (NOAA/GFDL). All data sets, codes for the bulk flux formulae, technical report, and other support codes along with the release notes are available at the CLIVAR

Ocean Model Development Panel (OMDP) web page

http://www.clivar.org/omdp/core

Importantly, groups should make use of the "corrected" forcing dataset available from the web site, since these files incorporate modifications from Large and Yeager (2009) aiming to address biases in the reanalysis product. The "uncorrected" data that are also available represent the raw fields without the Large and Yeager (2009) modifications. These raw fields should be used

only for those who have coded the Large and Yeager (2009) modifications directly into their flux coupler. Finally, we note that within the CMIP6 lexicon of experiment names, the identification for this simulation is

$\mathtt{experiment\_id = omip - ocore2}$.

## 2.1  CORE-II analysis papers

A broad community of scientists have provided a thorough analysis for an ensemble of 15 to 20 CORE-II simulations making

use of CMIP5-class ocean/sea-ice models. This work has been documented in the following research papers.

- Danabasoglu et al. (2014): North Atlantic mean

- Griffies et al. (2014): Global and regional sea level

- Downes et al. (2015): Southern Ocean watermasses and sea ice

- Farneti et al. (2015): Antarctic Circumpolar Current and Southern Ocean meridional overturning circulation

- Danabasoglu et al. (2016): North Atlantic variability

- Wang et al. (2016a): Arctic sea ice and solid freshwater

- Wang et al. (2016b): Arctic liquid freshwater

- Ilicak et al. (2016): Arctic hydrography

- Tseng et al. (2016): North and equatorial Pacific.

These papers help to define the state-of-the-science in forced global ocean/sea-ice simulations. They furthermore identify model diagnostics and metrics useful for evaluating the ocean component of all CMIP6 simulations.

Scientific and practical limitations of forced global ocean/sea-ice simulations are important to acknowledge (e.g., Section 3 of Griffies et al. (2009b)), though such models have their advantages, particularly in their ability to help interpret the observational record. In general, the utility of forced global ocean/sea-ice models depends on the scientific questions to be addressed.





The above papers detail their limitations and exemplify their scientific value across a broad suite of processes and ocean basins. By extension, these studies provide a compelling case for both the limitations and potentials for OMIP simulations included as part of CMIP6.

## 2.2 OMIP/CORE-II experimental protocol

The experimental protocol for the physical component of OMIP is detailed in Griffies et al. (2009b), Griffies et al. (2012) and Danabasoglu et al. (2014), with a summary provided here.

- INITIALIZATION

  1. Potential temperature (or Conservative Temperature as in IOC et al. (2010)) and salinity are initialized using January mean observational-based climatology based on Locarnini et al. (2013) for temperature and Zweng et al.
  (2013) for salinity. The January fields from Locarnini et al. (2013) and Zweng et al. (2013) stop at 1500 m, and should be filled with annual mean values for depths below 1500 m. These initial conditions are provided at the OMIP web site. Interpolation should then be made to the respective model grid.

  2. Velocity starts from a state of rest.

  3. Sea ice fields are generally initialized from an existing state taken from another simulation, set to the January mean
  state from that simulation.

  4. There is no recommended protocol for solid land ice calving, as iceberg models are generally disabled in OMIP experiments.

- FORCING

  1. HEAT FLUXES include radiative and turbulent components. The radiative fluxes are determined from Large and
  20 Yeager (2009). Turbulent heat fluxes are computed based on the evolving ocean state and the prescribed atmospheric state along with bulk formula. There is no restoring term applied to the surface ocean temperature. Rather, there is an implied surface temperature restoring due to the prescribed atmospheric state.

  2. BULK FORMULAE for computing turbulent fluxes for heat and momentum must follow Large and Yeager (2009). In particular, properties of moist air relevant to flux computation, such as air density, saturation specific humidity,
  and latent heat of vaporization, follow Large and Yeager (2009).

  3. SURFACE WATER FLUXES are provided by Large and Yeager (2009) for precipitation and Dai and Trenberth (2002) interannual for river runoff. Evaporation is computed by the model.

  4. SURFACE OCEAN SALINITY is damped to a monthly observational-based climatology provided by the OMIP web site above. Details of the damping time scale are not specified by the protocol (see Section 2.3 for more comments).

  5. BIOGEOCHEMICAL FORCINGS are detailed in Orr et al. (2016).





- SIMULATION LENGTH: For many purposes, simulations run for no less than five cycles of the 1948-2009 forcing have proven useful to remove dependence on details of the initial conditions, and to reach a quasi-equilibrium for at least the upper portion of the ocean. For OMIP in CMIP6, we ask for output from the fifth cycle. Doing so ensures that the simulations are compared at the same point in time after initialization from the same initial conditions.

## 2.3 Comments on surface salinity restoring

The real climate system has no direct feedback between sea surface salinity and surface water fluxes (see Durack et al. (2013) for further discussion). Hence, the need to provide a surface salinity restoring boundary condition is an unsatisfying aspect of OMIP, as for CORE. However, absent some form of restoring, models typically drift over decadal time scales. Such drift can reduce the physical utility of the simulations, particularly for studies of the high latitude circulation in the Atlantic and Southern Ocean. The key difficulty is associated with high latitude thermohaline and ocean/sea-ice processes that impact on the meridional overturning circulation (see Section 3 of Griffies et al. (2009b) as well as Behrens et al. (2013)).

Details of the salinity restoring appropriate to reduce thermohaline drift are sensitive to model details and the prescribed precipitation and river runoff data. Consequently, a single salinity restoring boundary condition for all models (i.e., a common piston velocity) has proven elusive. Example settings for a suite of CORE-II simulations can be found in Table 2 of Danabasoglu et al. (2014), including whether there is any restoring under sea-ice and/or normalization of the restoring flux over the globe. A full examination of model sensitivity to the salinity restoring has not been published, though some examination for specific models is provided in Section 16 of Griffies et al. (2009b) and Appendix C of Danabasoglu et al. (2014).

There is no proven correlation between model physical integrity, from a process perspective, and the strength of the salinity boundary condition required to stabilize the simulated overturning circulation. Claims that a "perfect" ocean/sea-ice model should be able to make use of zero surface salinity restoring ignore the many uncertainties in the prescribed precipitation and river runoff. More fundamentally, such claims ignore the otherwise undamped thermohaline feedbacks that emerge in the absence of an interactive atmosphere model that responds to changes in the ocean and sea-ice state. Salinity restoring is thus a stop-gap measure aimed to reduce runaway feedbacks that can arise in the global ocean/sea-ice models run with a prescribed atmospheric state.

It remains an ongoing research task to examine suitable salinity boundary conditions for global ocean/sea-ice models. Consequently, we do not offer protocol specifications. Rather, we recommend that modellers consult Appendix C in Danabasoglu et al. (2014) to see what has worked for other models. Choosing a weak salinity restoring is useful to reduce its impact on variability, and since restoring is itself unphysical.

## 2.4 Future plans for an OMIP Phase II

The present document focuses on Phase I of OMIP, which is a straightforward extension of the CORE-II project making use of the Large and Yeager (2009) atmospheric state. There is ongoing research coordinated through the CLIVAR Ocean





Model Development Panel (OMDP) to develop a Phase II experiment for OMIP.[2] The OMIP Phase II experiment will use an atmospheric state and runoff dataset based on the JRA-55 reanalysis (Kobayashi et al., 2015). JRA-55 has a finer spatial resolution, starts at year 1958, and remains updated to recent months. More details will be forthcoming when available.

## 3 OMIP as a diagnostics MIP

5  The diagnostic portion of OMIP coordinates ocean diagnostics for CMIP6 experiments involving an ocean component. The diagnostic suite is an updated and revised version of the CMIP5 ocean diagnostics detailed in Griffies et al. (2009a). For the remainder of this paper, we provide scientific rationales for the diagnostics, and detail the protocol for sampling the physical ocean fields used to create the diagnostics. Orr et al. (2016) focus on the chemical and biogeochemical tracers. Diagnostics for the sea ice component are coordinated by the CMIP6 Sea-ice Model Intercomparison Project (SIMIP) (Notz et al., 2016).

### 3.1 The needs of comparative analysis

Comparative ocean analysis involves computing differences between two realizations of a particular ocean property. One can then compute statistics, such as the mean square difference, thus rendering quantitative measures of the distance between model realizations (model-model comparisons), or between a model realization and an observation-based realization. Such *metric-based* analyses are the dominant means for assessing the precision and accuracy of CMIP simulations.

Analysts sometimes aspire for relatively complex manipulations of diagnosed fields. One example concerns an estimate of tracer transport given the archived tracer concentration, velocity, and grid geometry. Such offline *secondary analyses* are generally less accurate than when the analogous diagnostics are directly sampled online. Reasons for the loss of accuracy concern missing temporal correlations and/or offline numerical methods that only approximate online numerics. These limitations become particularly egregious when refining grid resolution so that simulations admit more of the ocean's dynamical spectrum

(e.g., mesoscale eddies, unstable boundary currents, inertia-gravity waves).

Whether computing differences of diagnosed fields, or performing more complex manipulations, any form of comparative analysis requires fields to be placed on a common grid. We propose that the most useful grid for analysis is based on depth/pressure for the vertical and spherical (latitude-longitude) in the horizontal.

### 3.2 Specifications for time sampling

There are three time sampling periods requested for various OMIP diagnostics: day, month, and year. Each period is sampled as a time average. Time averages include all model time steps over the given period; there is no sub-sampling. Products of time dependent fields are time averaged as a product, again using all model time steps to build the average.

---

[2]See preliminary discussions of this effort from the report of the Ocean Model Development Panel workshop on forcing ocean and sea-ice models, (OMDP, 2015).





## 3.3 Specifications for spatial sampling

Spatial integrals for regional diagnostics or transports across lines are computed using all grid points within the region or line. There is no sub-sampling.

## 3.4 Specifications for horizontal and vertical gridding

As for CMIP5, the CMIP6 ocean diagnostics should be remapped to depth/pressure vertical levels for those models not based on one of the vertical coordinates $z$, $z^*$, $p$, or $p^*$, with details of these vertical coordinates provided in Appendix B5.[3] For the horizontal, CMIP5 recommended that all two-dimensional (horizontal) and three-dimensional ocean diagnostics remain on the model native grid. We conjecture that this specification contributed to the dearth of analysis performed on the CMIP5 ocean diagnostics. We are thus motivated to revise that specification for CMIP6 by encouraging the remapping of diagnostics

to spherical grids.

### 3.4.1 Specifications for horizontal grids

Spherical latitude/longitude grids are rarely used for global ocean simulations given the desire to remove the spherical coordinate singularity from the ocean domain. Indeed, native ocean model grids commonly used today are rather complex, spanning the spectrum from general orthogonal (e.g., Murray (1996) and Madec and Imbard (1996)), unstructured finite volume (e.g.,

Ringler et al. (2013)) to unstructured finite element (e.g., Danilov (2013)).

    Horizontal grid cell dimensions are static in CMIP models. Remapping from native horizontal grids to a spherical grid can thus be performed offline, given sufficient information about grid cell areas. When choosing a remapping method, we note that conservative methods may be unnecessary for some purposes, such as for visualization and producing difference maps. Remapping technology for geophysical models has matured since CMIP5, although there is no community-wide consensus

on the best practices. Issues related to gridding and remapping for CMIP6 are discussed more thoroughly in the WGCM Infrastructure Panel (WIP) contribution to this CMIP6 special issue (Balaji et al., 2016).

    In considering whether to remap to a sphere or keep a field native, it is useful to consider the practice common for CMIP atmospheric diagnostics, which are archived on spherical grids.[4] Multiplicative manipulations (i.e., products of velocity times tracer) are avoided, since they are inaccurate, particularly given the high frequency variability inherent in the atmosphere. As

ocean model grids are refined, such high frequency transients will also corrupt correlations computed with time mean fields.

    Here is a summary of the specifications for horizontal gridding of ocean diagnostics in CMIP6.

  – Two-dimensional diagnostics are provided on a spherical grid; ideally the standard 1-degree grid from Levitus (1982) and Locarnini et al. (2013) (see more discussion in Appendix B).

---

[3]There is one exception for the vertical; namely, the overturning mass transport is of scientific interest on both depth/pressure surfaces and potential density surfaces (with 2000 dbar referencing). We detail this diagnostic in Section 6.5.

[4]The spherical grids are not the same, however, with each group generating their own.





- Three-dimensional diagnostics of Priority=1 are strongly recommended on a spherical grid. Native grid diagnostics can be provided as an alternative.

- Remapping of scalar fields should be conservative for those diagnostics meant for budget analysis (e.g., boundary fluxes in Tables 6, 7, 8, and budget terms in Table 10).

- Horizontal vector fields can be interpolated onto a common A-grid or B-grid (with geographic north-south and east-west components).

- Diagnostics stored in either native or spherical grids should have the same standard name and the same CMIP/CMOR (Climate Model Output Rewriter) name. Within files, the gridded fields should be distinguished by the cell_measures, coordinates, and auxiliary CF (Climate and Forecast) attributes.[5] The same variable in native and spherical form can

never coexist in the same file. The WGCM Infrastructure Panel (WIP) will provide means to distinguish the gridded fields (Balaji et al., 2016).

### 3.4.2 Specification for vertical grids

There are three general paradigms of vertical coordinates in use by the ocean model community (Griffies et al., 2000), with the choice guided by the scientific use of the model. The traditional geopotential coordinate has dominated ocean climate modelling

since the pioneering work of Bryan (1969). This choice matches well to the predominant means for mapping observation-based data, which typically occurs on geopotential or pressure levels. However, the alternative paradigms of terrain following coordinates and isopycnal coordinates are gaining traction in the climate community, along with even more general features offered by the Arbitrary Lagrangian-Eulerian (ALE) method (e.g., Bleck (2002), Donea et al. (2004), Adcroft and Hallberg (2006), Petersen et al. (2015)).

- Three-dimensional diagnostics should be archived on one of the depth or pressure based vertical coordinates $z$, $z^*$, $p$ or $p^*$ (Section B5).

- Models making use of alternative vertical coordinates should remap diagnosed fields to one of the coordinates $z$, $z^*$, $p$ or $p^*$. Vertical remapping should be conservative and performed online each model time step so to ensure scalar content (e.g., heat, salt, water) is preserved across the vertical native and vertical remapped grids.

### 3.4.3 Comments on the grid specifications

Remapping to a sphere is arguably more trustworthy if sanctioned by the respective modelling centres, which generally means the remapping is performed directly by the model centres prior to submitting data to the CMIP archives. The alternative is for the centres to provide sufficient grid information for analysts to perform the remapping, with details of such grid information provided in the WGCM Infrastructure Panel (WIP) contribution to this CMIP6 special issue (Balaji et al., 2016).

---

[5]See http://cfconventions.org/ for information about the CF conventions.




We consider here a case in point for what centres may wish to consider. Namely, for many CMIP6 simulations, GFDL is considering a $1/4°$ horizontal ocean grid, with 75-levels in the vertical. To reduce archive burden, GFDL has decided to coarsen the ocean diagnostics to a one-degree spherical grid with 33-levels in the vertical, with this grid defined by the World Ocean Atlas (Levitus, 1982; Locarnini et al., 2013). Besides reducing archive space, this approach will greatly facilitate use of

5 the diagnostics across a broad element of the analysis community. For others coarsening their output to a one-degree class of resolution, we encourage use of the same World Ocean Atlas grid.

Refer to Appendix A for further details of grid cell volume and area, and Appendix B for details of spatial sampling.

### 3.5 Specification for precision

All model data submitted to CMIP6 should follow the netCDF4 protocol (Appendix C). Precision of the data should be

sufficient to ensure robust statistical analyses. For most applications, single precision (seven significant digits) is sufficient.

### 3.6 Prioritizing the diagnostics

We make use of three priorities for OMIP diagnostics.[6]

- **Priority=1** diagnostics serve as a baseline for the CMIP physical ocean diagnostics. These diagnostics are of highest priority as they support a broad baseline of CMIP ocean related studies. Specifically, they facilitate

– characterizing the model configuration;

    – assessing the simulated climate state;

    – assessing the simulated climate change.

- **Priority=2** diagnostics support more in-depth understanding of the simulations. Specifically, they facilitate

    – measuring mass and tracer transports over the globe, within semi-enclosed basins, and across sections;

– quantifying mass, heat, and salt budget terms on global and/or regional scales;

    – documenting auxiliary fields that render a more complete quantitative characterization of the simulation, such as ventilation;

    – subsampling selected priority=1 fields to facilitate a quick assessment.

- **Priority=3** diagnostics serve process-based analyses of CMIP ocean simulations. Specifically, they facilitate

– quantifying three-dimensional heat and salt budgets;

    – studying sub-monthly transients and/or variability;

    – documenting parameterized eddy coefficients to characterize subgrid scale schemes;

    – quantifying impacts on energetics from parameterizations.

---

[6]CMIP5 level=0 diagnostics detailed in Griffies et al. (2009a) are now termed CMIP6 priority=1 diagnostics, and so on for the other priorities. This change in prioritization nomenclature brings the OMIP diagnostic suite into alignment with other CMIP6 MIPs.



### 3.7 Sponsors for diagnostics

Our recommended suite of OMIP diagnostics reflect the needs and interests of numerous analysts, including the authors of this document and their extended network of collaborators. The recommendations generally resulted from surveys and discussions within the ocean and climate community, with particular feedback from the CMIP5 process. Furthermore, the list reflects our expert judgement regarding what diagnostics can be useful for scientifically assessing the simulations, and for providing mechanistic understanding of various physical processes.

Although tempting to ask for an extensive suite of diagnostics that allow for complete process-based examination of the simulations, any diagnostic suite must confront the reality of finite archive space and limited human resources. Furthermore, CMIP6 diagnostics should have a sponsor, such as a WCRP science panel or a CMIP6 MIP. Sponsors are expected to lead in analyzing selected diagnostics for peer-review publications. Note, however, sponsoring a diagnostic does not constitute ownership. The CMIP6 archive is public and thus available to anyone.

We identify the following CMIP6 sanctioned MIPs having directly sponsored ocean diagnostics listed in this document.

1. OMIP, as represented by the authors of this document, is the sponsor for the bulk of the diagnostics requested here.

2. FAFMIP (Flux-Anomaly-Forced Model Intercomparison Project) (Gregory et al., 2016) supports the WCRP Grand Challenge on sea level rise and regional impacts, with particular interest in identifying mechanisms for regional sea level variations/projections. Critical needs for this MIP include budget terms for heat and salt summarized by Table 10. These terms will help to understand mechanisms for the large spread in ocean heat uptake efficiency found in CMIP5 simulations (Kuhlbrodt and Gregory, 2012), which in turn impact on the spread in projected sea level rise (Slangen et al., 2012; Church et al., 2013b; Slangen et al., 2014).

3. C4MIP (Coupled Climate Carbon Cycle Model Intercomparison Project) (Jones et al., 2016) aims to improve and accelerate development of global-scale, three-dimensional, coupled earth system models that include the carbon cycle and related biogeochemical and ecosystem components. C4MIP is the primary sponsor of chemical and biogeochemical tracers discussed in Orr et al. (2016).

4. HIGHRESMIP (High Resolution Model Intercomparison Project) (Haarsma, 2016) aims to assess the robustness of improvements in the representation of important climate processes with "weather-resolving" global model resolutions (roughly 25 km or finer), within a simplified framework using the physical climate system with constrained aerosol forcing.

5. DCPP (Decadal Climate Prediction Project) (Boer et al., 2016) is a WCRP project and a CMIP6 MIP that aims to improve and accelerate development of global climate prediction systems on annual, multi-annual, and decadal time scales.

Although not explicitly sponsoring a MIP for CMIP6, the following communities have provided input to the OMIP diagnostics.



1. GSOP (Global Synthesis and Observational Panel) is a CLIVAR panel that has encouraged analysis of various physical processes, such as those proposed by FAFMIP in support of understanding the role of physical processes in heat and salt budgets (Table 10).

2. The AMOC (Atlantic Meridional Overturning Circulation) community, as represented by the U.S. CLIVAR AMOC Science Team, has emphasized the needs for diagnostics measuring the mass, heat, and salt transport in the Atlantic. The budget terms in Table 10 will also be of prime interest for regional AMOC and Arctic analysis, with particular use to diagnose the role of ocean processes in impacting the AMOC (Roberts et al., 2013, 2014).

3. SOUTHERN OCEAN: The Southern Ocean community, as represented by members of the CLIVAR/CliC/SCAR Southern Ocean Region Panel, has a particular interest in the detailed workings of mesoscale eddy parameterizations, particularly as they impact on vertical heat transport (e.g., Gregory (2000)), and response of the Antarctic Circumpolar Current (ACC) and Southern Ocean meridional overturning circulation (MOC) to changes in surface forcing (e.g., Downes and Hogg (2013), Downes et al. (2015), Farneti et al. (2015)).

4. ECOSYSTEM COMMUNITY: Certain physical fields are of direct use for assessing the impacts of physical climate on ecosystems, with bottom temperature and salinity fields of particular interest.

5. OCEAN MIXING COMMUNITY: Focusing on ocean subgrid scale processes, the mixing community is served by documenting model parameterizations. In CMIP5, there were numerous fields requested to serve this community (see Tables 11 and 12). Unfortunately, few model submissions were made, thus leading to little published analysis of these fields.[7] For CMIP6, we only recommend a subset of the CMIP5 information in hopes that more model groups will submit the reduced number of fields, thus better serving the needs of subgrid scale parameterization studies.

## 3.8 Tabulation of ocean diagnostics

The following sections provide a tabulated list of ocean model diagnostics recommended for the CMIP6 archive, with the tables containing the following information.

- diagnostic name according to the CMOR short name and the CF standard name;

- community sponsoring the diagnostic (Section 3.7);

- relation to CMIP5 as detailed in Griffies et al. (2009a) (same, new, or modification);

- physical units;

- time sampling for output (time mean over day, month, or year);

- spatial shape (i.e., 1d, 2d, or 3d);

---

[7]Downes and Hogg (2013) characterize the Southern Ocean in CMIP5 simulations in terms of the ocean mesoscale eddy parameterizations. Unfortunately, few models submitted the relevant eddy diffusivities and overturning streamfunctions.



- recommended grid (native, spherical, depth/pressure);

- prioritization guidance (Section 3.6);

- experiment for which the diagnostic should be saved. Unless otherwise specified, results should be submitted for the full length of each experiment.

All fields are reported as "missing" over grid cells that are entirely land. The spatial shape of a field that has no horizontal dimension(s) is indicated by a 0 (i.e., a time series); one-dimensional spatial fields are denoted by Y (e.g., meridional heat transport); horizontal two dimensional fields are denoted XY; vertical two dimensional fields are denoted YZ or Y$\rho$; three-dimensional fields are denoted XYZ.[8]

### 3.9    Names and units for the diagnostics

The CMOR names and units for diagnostics are listed in the tables. Model diagnostics submitted to the CMIP6 archive must follow this name and unit convention. Notably, the CMOR names are becoming a community standard given the widespread development of CMIP analysis software.

There is an additional name that is supported by the CF metadata conventions. We list these CF "standard names" in the following sections, and relate these names to the CMOR names. The CF standard names are self-explanatory and interdisciplinary, which can result in rather long names. We thus include them as sub-tables within the diagnostic tables.

### 3.10    Diagnostics for ocean spin-up

The deep ocean is ventilated on multi-centennial timescales (Sen Gupta and England, 2004; Stouffer, 2004) which can lead to drifts in many ocean properties on long timescales (Banks et al., 2007). Drift needs to be taken into account when analysing ocean diagnostics from CMIP6.

Five cycles of OMIP atmospheric forcing is insufficient to reach full equilibrium of the full ocean (Danabasoglu et al. (2014) and Griffies et al. (2014)). Nonetheless, comparison of different models simulations at the same point in their fifth forcing cycle of OMIP simulations is meaningful, since they have all integrated for the same time starting from the same initial conditions (see simulation length discussed in Section 2.2).

In contrast, many of the other CMIP6/MIPs are run relative to the CMIP DECK experiment piControl. The experiment piControl is itself initialised from piControl-spinup, which has an unspecified length (Eyring et al., 2015). The spin-up length determines the residual drift in ocean properties from ocean temperature and salinity (Pardaens et al., 2003) to biogeochemistry (Seferian et al., 2015), with shorter spin-up generally exhibiting larger drifts.

The CMIP6 experiments abrupt4xCO2, 1pctCO2 and Historical Simulations, FAFMIP, C4MIP and ScenarioMIP are all concerned with the ocean response to forcing. In these cases, the underlying climate drift from piControl should be subtracted from the response. If absolute quantities are required for analysis, it is recommended that the underlying climate drift in

---

[8]XYZ is a shorthand for the more detailed prescription of both horizontal and vertical grids, with details given in Appendix B.





piControl is analysed and the length and details of the spin-up in piControl-spinup is considered. The priority 1 diagnostics in Tables 2, 3, 6, 7, and 8 should be saved as decadal time means at decadal intervals.

## 4 Static fields and functions

We list here the static fields and functions needed to describe elements of the ocean model, with a summary in Table 1.

| STATIC DIAGNOSTICS | | | | | | | | | |
|---|---|---|---|---|---|---|---|---|---|
| ITEM | CMOR NAME | SPONSOR | CMIP5/CMIP6 | UNITS | TIME | SHAPE | GRID | PRIORITY | EXPT |
| 1 | equation of state | OMIP | same | $\rho(S, \Theta, p)$ or $\rho(S, \Theta, z)$ | | | | 1 | once |
| 2 | freezing point | OMIP | same | function of $(S, p)$ or $(S, z)$ | | | | 1 | once |
| 3 | rhozero | OMIP | same | $kg/m^3$ | static | 0 | 0 | 1 | once |
| 4 | cpocean | OMIP | new | $J/(kg\,K)$ | static | 0 | 0 | 1 | once |
| 5 | deptho | OMIP | same | m | static | XY | native & sphere | 1 | once |
| 6 | basin | OMIP | same | dimensionless | static | XY | native & sphere | 1 | once |
| 7 | areacello | OMIP | same | $m^2$ | static | XY | native & sphere | 1 | once |
| 8 | masscello | OMIP | same | $kg/m^2$ | static | XYZ | native & sphere, z/p | 1 | once |

| CMOR NAME RELATED TO CF STANDARD NAME | | |
|---|---|---|
| ITEM | CMOR NAME | CF STANDARD NAME |
| 3 | rhozero | reference_sea_water_density_for_boussinesq_approximation |
| 4 | cpocean | specific_heat_capacity_of_sea_water |
| 5 | deptho | sea_floor_depth_below_geoid |
| 6 | basin | region |
| 7 | areacello | cell_area |
| 8 | masscello | sea_water_mass_per_unit_area |

**Table 1.** Static fields and functions to be saved for the ocean model component in CMIP6, as well as how the CMOR name for a diagnostc is related to its CF standard name. These fields provide basic information about the model configuration, and need only be archived once for all the model experiments in the CMIP6 repository (hence the "once" entry for the experiment column). Blank entries signal a characteristic that is not applicable for this particular diagnostic. Furthermore, the bottom topography, grid length and areas, and basin regions should be made available on *both* the model native grid and on the spherical latitude/longitude grid onto which scalars are remapped. The entry for masscello applies only to Boussinesq models with static grid cell volumes (Appendix A). For other kinds of models, masscello is generally time-dependent and the entry in Table 2 applies instead.

### 4.1 Equation of state

- sea_water_equation_of_state

This diagnostic is in fact a mere citation to the literature source for the model equation of state used to compute *in situ* density ($kg/m^3$). Its functional dependence should also be noted (see IOC et al. (2010)):

- potential temperature $\theta$ or Conservative Temperature $\Theta$

- practical salinity $S_P$ or Absolute Salinity $S_A$





- pressure (dbars) or depth (metres).

## 4.2 Freezing temperature for seawater

- sea_water_freezing_temperature_equation

Ocean models use a variety of equations to determine when liquid seawater freezes to form frazil and then sea ice (McDougall
et al., 2014). It is thus useful for studies of high latitude processes to document the equation used to compute the freezing point
(in degrees C) of seawater, as a function of salinity and pressure.

## 4.3 Boussinesq reference density

- rhozero

Many ocean climate models employ the Boussinesq approximation, in which there appears a constant reference density $\rho_o$
within budgets for tracer and momentum, and volume is conserved rather than mass. It is useful to have an archive of this
constant for CMIP6.

As noted on page 47 of Gill (1982), with the exception of only a small percentage of the ocean, *in situ* density in the World
Ocean varies by no more than 2% from $1035\,\mathrm{kg\,m^{-3}}$. Hence, $\rho_o = 1035\,\mathrm{kg\,m^{-3}}$ is a sensible choice for the reference density
used in a Boussinesq ocean climate model. However, some models may use a different value. For example, early versions of the
GFDL ocean model (Cox, 1984) set $\rho_o = 1000\,\mathrm{kg\,m^{-3}}$. Others choose the average density corresponding to the thermocline
region. Roquet et al. (2015) present a summary of various choices.

## 4.4 Seawater heat capacity

- cpocean

As detailed in McDougall (2003) and IOC et al. (2010), the heat capacity of seawater is a constant when measuring the
heat content of a parcel in terms of Conservative Temperature. As discussed in Appendix D, not all models choose to use
Conservative Temperature as their prognostic heat variable. However, all ocean models use a constant heat capacity, $c_p^o$, to
convert between prognostic model temperature and heat content, though not all models use the TEOS-10 value for heat capacity
(see equation (D3) in Appendix D3). Hence, to enable accurate comparisons between ocean model heat contents, we ask that
all models archive their choice for the constant seawater heat capacity.

A useful method to archive this constant is to include it as part of the metadata for the potential temperature or Conservative
Temperature diagnostic (Sections 5.10 and 5.11), *as well as* for all heat related diagnostics (e.g., boundary heat fluxes in Table
8 and heat budget terms in Table 10). Doing so will facilitate computation of ocean heat content consistent with how the model
converts boundary enthalpy fluxes into temperature tendencies.





### 4.5 Bathymetry

- deptho

For global primitive equation ocean models, the geoid is assumed to correspond to the geopotential surface $z = 0$. The distance from $z = 0$ to the ocean bottom defines the ocean depth field, $H(x, y)$, or the ocean *bathymetry*, and the vertical position of the

bottom is

$$z = -H(x, y). \tag{1}$$

This solid earth boundary used by the model should be archived. Precisely, the bathymetry representing the ocean bottom from the perspective of the model tracer fields defines the field deptho.

If the lateral area for exchange of fluid between columns (e.g., mass transport) is anything other than a simple function of

the tracer column depths, then the modulated areas affecting the exchange is useful to archive. For example, this additional information is necessary for models that allow a strait to be more narrow than the nominal width of the cell. However, at this time there is no OMIP specification for this area.

### 4.6 Tracer region masks

- basin

Analysis of budgets and properties over ocean basins is commonly performed for the purpose of assessing the integrity of simulations. This analysis generally involves the use of a mask that partitions the model grid into ocean basins (some enclosed seas may be missing from the model). We recommend the following ocean regions, with the names corresponding to standard CF basin names found at `http://cfconventions.org/`

1. southern_ocean

2. atlantic_ocean

3. pacific_ocean

4. arctic_ocean

5. indian_ocean

6. mediterranean_sea

7. black_sea

8. hudson_bay

9. baltic_sea



10. red_sea

These region masks are set according to the following flag_values and flag_meanings, which should be recorded as attributes of the variable:

- flag_values=1,2,3,4,5,6,7,8,9,10

- flag_meanings="southern_ocean, atlantic_ocean, pacific_ocean, arctic_ocean, indian_ocean, mediterranean_sea, black_sea, hudson_bay, baltic_sea, red_sea"

Many budget analyses are computed over regions, in which case it is very useful for the analyst to have access to the regional tracer mask specific to each model grid. Additionally, for some grid staggering (such as B-grid), the tracer mask will differ from velocity mask, in which case a mask for the velocity cells should be provided to the CMIP6 archive as a distinct output

variable, with the same standard name of region. The two variables are distinguished in netCDF by their coordinates, one being on the tracer grid and the other on the velocity grid.

### 4.7 Horizontal area of a tracer cell

- areacello

The static field areacello provides a measure (in m$^2$) of the horizontal area for a tracer cell. More details for areacello are

15 provided in Appendix A.

### 4.8 Mass of a grid cell for Boussinesq models with static cell volumes

- masscello

The three-dimensional masscello field measures the mass per horizontal area of a grid cell. As discussed in Appendix A1, some volume conserving Boussinesq models maintain static cell volumes, in which case masscello should be reported as part of the

20 Table 1 request. Otherwise, the mass of a grid cell is time dependent and so should be reported as in Section 5.3.

### 4.9 Extra grid information to facilitate remapping

There are ongoing discussions within the CMIP community regarding the conventions required to horizontally remap fields from the native grids to the sphere. Refer to (Balaji et al., 2016) for guidance from the WGCM Infrastructure Panel (WIP).

## 5 Scalar fields

In this section, we present specifications for the scalar fields to be archived as part of CMIP6, with a summary in Table 2.





## SCALAR FIELDS

| ITEM | CMOR NAME | SPONSOR | CMIP5/CMIP6 | UNITS | TIME | SHAPE | GRID | PRIORITY | EXPT |
|---|---|---|---|---|---|---|---|---|---|
| 1 | pbo | OMIP | dbar → Pa | Pa | month | XY | sphere | 1 | all |
| 2 | pso | OMIP | dbar → Pa | Pa | month | XY | sphere | 1 | all |
| 3 | masscello | OMIP | same | $kg/m^2$ | month | XYZ | sphere, z/p | 1 | all |
| 4 | thkcello | OMIP | same | m | month | XYZ | sphere, z/p | 1 | all |
| 5 | masso | OMIP | same | kg | month | 0 | | 1 | all |
| 6 | volo | OMIP | same | $m^3$ | month | 0 | | 1 | all |
| 7 | zos | OMIP | same | m | month | XY | sphere | 1 | all |
| 8 | zossq | OMIP | same | $m^2$ | month | XY | sphere | 3 | all |
| 9 | zostoga | OMIP | same | m | month | 0 | | 1 | all |
| 10 | thetao | OMIP | K → $^\circ C$ | $^\circ C$ | month | XYZ | sphere (native), z/p | 1 | all |
| 11 | thetaoga | OMIP | K → $^\circ C$ | $^\circ C$ | month | 0 | | 1 | all |
| 12 | bigthetao | OMIP | new (if TEOS-10 based model) | $^\circ C$ | month | XYZ | sphere (native) z/p | 1 | all |
| 13 | bigthetaoga | OMIP | new (if TEOS-10 based model) | $^\circ C$ | month | 0 | | 1 | all |
| 14 | tos | OMIP | K → $^\circ C$ | $^\circ C$ | month | XY | sphere (native) | 1 | all |
| 15 | tosga | OMIP | new | $^\circ C$ | month | 0 | | 1 | all |
| 16 | tos | OMIP | K → $^\circ C$ | $^\circ C$ | day | XY | sphere | 3 | all |
| 17 | tossq | OMIP | K → $^\circ C$ | $^\circ C^2$ | month | XY | sphere | 3 | all |
| 18 | tossq | OMIP | K → $^\circ C$ | $^\circ C^2$ | day | XY | sphere | 3 | all |
| 19 | tob | OMIP | new | $^\circ C$ | month | XY | sphere | 1 | all |
| 20 | so | OMIP | same | 1e-3 | month | XYZ | sphere (native), z/p | 1 | all |
| 21 | soga | OMIP | same | 1e-3 | month | 0 | | 1 | all |
| 22 | sos | OMIP | same | 1e-3 | month | XY | sphere | 1 | all |
| 23 | sosga | OMIP | new | 1e-3 | month | 0 | | 1 | all |
| 24 | sos | OMIP | new | 1e-3 | day | XY | sphere | 3 | all |
| 25 | sob | OMIP | new | 1e-3 | month | XY | sphere | 1 | all |
| 26 | obvfsq | OMIP | new | $s^{-2}$ | month | XYZ | sphere (native), z/p | 1 | all |
| 27 | agessc | OMIP | same | year | month | XYZ | sphere (native), z/p | 1 | all |
| 28 | mlotst | OMIP | same | m | month | XY | sphere | 1 | all |
| 29 | mlotstsq | OMIP | same | $m^2$ | month | XY | sphere | 3 | all |
| 30 | msftbarot | OMIP | same | $kg/s$ | month | XY | sphere | 1 | all |

| | CMOR NAME RELATED TO CF STANDARD NAME | |
|---|---|---|
| ITEM | CMOR NAME | CF STANDARD NAME |
| 1 | pbo | sea_water_pressure_at_sea_floor |
| 2 | pso | sea_water_pressure_at_sea_water_surface |
| 3 | masscello | sea_water_mass |
| 4 | thkcello | cell_thickness |
| 5 | masso | sea_water_mass |
| 6 | volo | sea_water_volume |
| 7 | zos | sea_surface_height_above_geoid |
| 8 | zossq | square_of_sea_surface_height_above_geoid |
| 9 | zostoga | global_average_thermosteric_sea_level_change |
| 10 | thetao | sea_water_potential_temperature |
| 11 | thetaoga | sea_water_potential_temperature |
| 12 | bigthetao | sea_water_conservative_temperature |
| 13 | bigthetaoga | sea_water_conservative_temperature |
| 14/16 | tos | sea_surface_temperature |
| 15 | tosga | sea_surface_temperature |
| 17 | tossq | square_of_sea_surface_temperature |
| 18 | tossq | square_of_sea_surface_temperature |
| 19 | tob | sea_water_potential_temperature_at_sea_floor |
| 20 | so | sea_water_salinity |
| 21 | soga | sea_water_salinity |
| 22/24 | sos | sea_surface_salinity |
| 23 | sosga | sea_surface_salinity |
| 25 | sob | sea_water_salinity_at_sea_floor |
| 26 | obvfsq | square_of_brunt_vaisala_frequency_in_sea_water |
| 27 | agessc | sea_water_age_since_surface_contact |
| 28 | mlotst | ocean_mixed_layer_thickness_defined_by_sigma_t |
| 29 | mlotstsq | square_of_ocean_mixed_layer_thickness_defined_by_sigma_t |
| 30 | msftbarot | ocean_barotropic_mass_streamfunction |

**Table 2.** Scalar fields to be saved from the ocean component in CMIP6 ocean model simulations. Entries with grids denoted "sphere (native)" denote diagnostics where spherical output is strongly recommended to facilitate analysis, though where native output is accepted if spherical is unavailable. The column indicating the experiment for saving the diagnostics generally says "all", in which case we recommend the diagnostic be saved for CMIP6 experiments in which there is an ocean model component, including the DECK, historical simulations, FAFMIP, DAMIP, DCPP, ScenarioMIP, and C4MIP, as well as the ocean-sea ice OMIP simulations. The Priority=1 diagnostics should be saved as decadal time means at decadal intervals for the piControl-spinup. Blank entries signal a characteristic that is not applicable for this particular diagnostic. Squared diagnostics are computed online by accumulating each time step, so that they can be useful for computing variance fields. The entry for masscello applies for models with time-dependent cell masses (Appendix A). For Boussinesq models with static grid cell volumes, the entry in Table 1 applies instead. The variables bigthetao and bigthetaoga are requested only for models enacting the TEOS-10 Conservative Temperature field as a prognostic model variable (see Section 5.12 and Appendix D). The lower sub-table lists the CMOR names and their corresponding CF standard names.



## 5.1 Pressure at ocean bottom

- pbo

The bottom pressure in a hydrostatic ocean is given by the gravitational acceleration acting on the mass per area of a fluid column, plus any pressure applied at the ocean surface from the overlying atmosphere or ice.[9] In a discrete model, pbo is given
by the vertical sum over the levels/layers in the column

$$p_\mathrm{b} = p_\mathrm{a} + g \sum_k \rho \, \mathrm{d}z \qquad (2)$$

where $p_\mathrm{a}$ is the pressure applied at the ocean surface (pso discussed in Section 5.2), and we assumed a constant gravitational acceleration.[10] Hence, $g^{-1}(p_\mathrm{b} - p_\mathrm{a})$ is the mass per horizontal area of a fluid column. The bottom pressure is a prognostic field in non-Boussinesq hydrostatic models, whereas it is diagnosed in Boussinesq hydrostatic models. Anomalies of bottom
pressure with respect to a suitable reference value, such as $\rho_o \, g \, H$, provide a means for measuring mass adjustments throughout the water column.

If the model is Boussinesq (very common), then an adjustment must be made to account for spurious mass sources in the Boussinesq fluid. In particular, if interested in the mass distribution of seawater, as needed for angular momentum (Bryan, 1997), bottom pressure (Ponte, 1999), or geoid perturbations (Kopp et al., 2010), one must account for this spurious mass
change that arises due to the oceanic Boussinesq approximation. Details for how to compute these adjustments are provided in Section D.3.3 of Griffies and Greatbatch (2012). Please make a note in the meta-data whether an adjustment has been made to correct for the Boussinesq error.

## 5.2 Pressure applied to ocean surface

- pso

The pressure applied to the ocean surface from the overlying atmosphere is often neglected in climate simulations, in which case it should not be included in the diagnostic pso. However, models that incorporate this effect offer the means to simulate the inverse barometer response of the sea surface (for a review of the basics, see Appendix C in Griffies and Greatbatch (2012)). Changes in atmospheric pressure present a rapid barotropic forcing to the ocean (Arbic, 2005). Additionally, changes in the distribution of mass in the atmosphere can lead to noticeable changes in regional sea level (e.g., Goddard et al. (2015)).

In addition to atmospheric mass impacting on the ocean, there is mass from overlying sea ice, ice shelves, and icebergs. If the ocean model feels this mass through undulations of its free surface, then the mass per area should be included in pso.

Note that solid runoff is defined as all frozen water that enters the ocean from land, such as from snow and land-ice, lake-ice and river-ice. For example, snow can enter in its frozen state when a land model has a buffer layer of a certain thickness, with

---

[9]If the model is non-hydrostatic, the bottom pressure is affected by the mass per area of the ocean fluid, plus non-hydrostatic fluctuations in the pressure field. Non-hydrostatic dynamics become important for the ocean at spatial scales finer than roughly 100 m (e.g., Marshall et al. (1997)). These grid scales are finer than any ocean model used for CMIP6, hence all CMIP6 ocean models are hydrostatic.

[10]All CMIP6 simulations assume a constant gravitational acceleration, $g$. However, the precise value may slightly differ between the models.




all snow exceeding this buffer conveyed to the ocean. Land-ice can enter the ocean as icebergs resulting from an ice sheet/shelf model, or formed from snow excess (e.g., Jongma et al. (2009), Martin and Adcroft (2010), Marsh et al. (2015)). There is no increase in liquid ocean water until the solid runoff melts. However, the presence of solid ice affects the pressure felt within the liquid ocean column, and affects the heat budget of the ocean through the latent heat of fusion.

In rigid lid ocean models, the term "surface pressure" refers to the hydrostatic pressure at $z = 0$ associated with the layer of liquid water between $z = 0$ and $z = \eta$ (Pinardi et al., 1995). This pressure is also sometimes referred to as the "lid pressure." It can be positive or negative depending on whether the free surface, $\eta$, is positive or negative. This "surface pressure" field is distinctly *not* what we refer to here by pso. Instead, pso records the non-negative pressure applied at $z = \eta$ due to media above the ocean surface interface. Such pressure may be set to zero in some approximate model formulations, such as the rigid lid, in

which case the ocean pressure is not influenced by movement of mass outside the liquid ocean.

### 5.3  Mass per area of grid cell

- masscello = cell mass per unit horizontal area

To estimate tracer budgets offline, we require the mass per horizontal area of seawater in the grid cell

$$\text{masscello} = \rho\,\mathrm{d}z, \tag{3}$$

with units of kg/m$^2$. For a hydrostatic model, the mass per area is proportional to the pressure increment $\mathrm{d}p$ according to $\mathrm{d}p = -g\,\rho\,\mathrm{d}z$, so that

$$\text{masscello} = -g^{-1}\mathrm{d}p \qquad \text{hydrostatic.} \tag{4}$$

For a Boussinesq model, the *in situ* density factor, $\rho$, in equation (3) is set to the constant reference density, $\rho_o$, so that

$$\text{masscello} = \rho_o\,\mathrm{d}z \qquad \text{hydrostatic / Boussinesq,} \tag{5}$$

in which case the mass per area is equivalent to $\rho_o$ times the grid cell thickness (Section 5.4).

### 5.4  Thickness (i.e., volume per area) of grid cell

- thickcello = cell volume per unit horizontal area

The tracer cell thickness

$$\text{thkcello} = \mathrm{d}z \tag{6}$$

measures the distance (in metres) between surfaces of constant vertical coordinate. This information is useful, in particular, for measuring changes in thickness between pressure surfaces in a non-Boussinesq pressure-based model exposed to increasing anthropogenic warming. If available, the thickness of a velocity cell should also be archived, which is particularly useful for C-grid models.





## 5.5 Total mass of liquid seawater

- masso

### 5.5.1 Summary of the diagnostic

This diagnostic is the global sum of the grid cell area (areacello in Section 4.7) multiplied by the cell mass per area (masscello
in Section 5.3). For the purpose of global budgets in non-Boussinesq models, it is essential to have the total mass of liquid
seawater in the ocean domain. This scalar field includes all seawater contained in the liquid ocean, including any enclosed seas
that are part of the ocean model integration. As a discrete sum of the three-dimensional grid cells, masso is given by

$$\text{masso} = \mathcal{M} = \sum_{i,j,k} \rho \, dA \, dz \qquad \text{non-Boussinesq,} \tag{7}$$

with $\rho$ the *in situ* density,

$$dA = dx \, dy \tag{8}$$

the horizontal area of a grid cell, and $dz$ the vertical thickness. For a hydrostatic fluid, $dp = -g \, \rho \, dz$ so that the total seawater
mass in a non-Boussinesq model is given by

$$\text{masso} = -g^{-1} \sum_{i,j,k} dA \, dp \qquad \text{hydrostatic non-Boussinesq.} \tag{9}$$

For a Boussinesq model, the density factor in equation (7) becomes a constant, $\rho_o$, so that the net mass is given by

$$\text{masso} = \sum_{i,j,k} \rho_o \, dA \, dz \qquad \text{Boussinesq,} \tag{10}$$

in which case the mass is equal to $\rho_o$ times the total volume of liquid in the ocean (Section 5.6).

### 5.5.2 Theoretical considerations

In a non-Boussinesq ocean, the total mass of liquid seawater evolves according to the budget

$$\frac{d\mathcal{M}}{dt} = \sum_{i,j} Q^{\mathrm{m}} \, dA, \tag{11}$$

where $Q^{\mathrm{m}}$ (kg m$^{-2}$ s$^{-1}$) is the net mass flux that crosses the liquid ocean boundaries, per horizontal cross-sectional area, due to
evaporation, precipitation, runoff, and material tracers such as salt.[11] Maintenance of this mass budget is a fundamental feature
of a conservative non-Boussinesq ocean model.

---

[11]Most ocean models do not add or remove mass associated with the transfer of material tracers across the ocean surface.



## 5.6 Total volume of liquid seawater

- volo

This diagnostic measures the sum of the three-dimensional tracer grid cell volumes

$$\text{volo} = \mathcal{V} = \sum_{i,j,k} \mathrm{d}A\,\mathrm{d}z. \tag{12}$$

It is computed as the global sum of the product of the grid cell horizontal area (areacello in Section 4.7) and the grid cell thickness (thkcello in Section 5.4).

In a Boussinesq fluid, volo evolves according to the budget

$$\frac{\mathrm{d}\mathcal{V}^{\text{Bouss}}}{\mathrm{d}t} = \frac{1}{\rho_o} \sum_{i,j} Q^{\text{m}}\,\mathrm{d}A. \tag{13}$$

Maintenance of this volume budget is a fundamental feature of a conservative Boussinesq ocean model. In particular, if there are no net boundary fluxes of volume, then a conservative Boussinesq model will retain a constant total volume to within computational roundoff.

In contrast, a non-Boussinesq model will generally alter its volume in cases even with zero boundary mass fluxes, since non-Boussinesq models conserve mass rather than volume. Hence, the non-Boussinesq model's total volume changes through changes in the global mean ocean density, with such changes referred to as *steric effects* (e.g., Griffies and Greatbatch (2012)).

## 5.7 Dynamic sea level

- zos

This diagnostic field has a zero global area mean, so that it measures sea level pattern fluctuations around the ocean geoid defined via a resting ocean state at $z = 0$. That is, zos is the *dynamic sea level* as defined in Griffies and Greatbatch (2012) or Griffies et al. (2014). The dynamic sea level reflects fluctuations due to ocean dynamics. Consequently, this diagnostic is not used to map the global mean sea level changes due to thermal expansion or changes in ocean mass. Rather, global mean changes due to thermosteric effects are archived in zostoga (Section 5.9). In the following, we identify various technical points regarding the dynamic sea level diagnostic.

### 5.7.1 Non-Boussinesq versus Boussinesq

Non-Boussinesq models incorporate global steric effects contributing to sea level changes, such as those related to thermal expansion. In contrast, the prognostic sea surface height in Boussinesq models does not incorporate global steric effects (Greatbatch, 1994). When removing the global mean, sea level patterns from Boussinesq and non-Boussinesq models are directly comparable (Losch et al., 2004; Griffies and Greatbatch, 2012).





### 5.7.2   Algorithm for computing sea surface height

It should be noted in the "comment" attribute whether zos is obtained directly, as in a free-surface model, or diagnostically derived. Diagnostic methods can follow from assuming velocities are geostrophic at some level, or from geostrophy relative to an assumed level of quiescence. Gregory et al. (2001) summarizes various methods of estimating sea-level in rigid-lid models.

Notably, these methods are largely obsolete, since CMIP6 ocean models generally do not make the rigid lid approximation.

### 5.7.3   Inverse barometer from sea ice loading

In some coupled climate models, sea ice at a grid cell depresses the liquid seawater through its mass loading (appearing as an applied surface pressure on the ocean model as discussed in Section 5.2). This depression occurs independent of the subgrid scale distribution of sea ice, as it is a result of the mass of sea ice in a grid cell acting on the liquid ocean. There is, however, no

dynamical effect associated with these depressions in the liquid ocean sea level, so there are are no associated ocean currents. See Appendix C in Griffies and Greatbatch (2012) for a discussion of this inverse barometer effect of sea ice.

- For OMIP, do *not* record inverse barometer responses from sea ice loading in zos. Rather, zos is the effective sea level as if sea ice (and snow) at a grid cell were converted to liquid seawater (Campin et al., 2008).

- A means to measure the *effective* dynamic sea level is to remove the inverse barometer response to applied pressure
loading on the ocean from sea ice (e.g., see equation (206) of Griffies and Greatbatch (2012))

$$\eta_{\text{effective}} = \eta_{\text{model}} + \frac{p_{\text{ice loading}}}{g\,\rho_{\text{surf}}}. \tag{14}$$

In this equation, $\eta_{\text{model}}$ is the sea level computed by the ocean model, $p_{\text{ice loading}}/g$ is the mass per unit area of the applied surface loading on the ocean, and $\rho_{\text{surf}}$ is the surface ocean density.[12] For OMIP purposes, the surface ocean density can be approximated by a constant $\rho_o$.

- The term $p_{\text{ice loading}}/(g\,\rho_{\text{surf}})$ in equation (14) acts to remove the sea ice loading inverse barometer response contained in $\eta_{\text{model}}$. Thereafter, we normalize to zero (global area integral vanishes) to render the effective dynamic sea level for OMIP

$$\eta_{\text{omip}} = \eta_{\text{effective}} - \left( \frac{\sum_{i,j} \eta_{\text{effective}}\, \mathrm{d}A}{\sum_{i,j} \mathrm{d}A} \right). \tag{15}$$

It is the effective dynamic sea level, $\eta_{\text{omip}}$, that should be reported in zos.

### 5.7.4   Inverse barometer from atmospheric loading

The inverse barometer response of sea level arising from atmospheric loading has, for many applications on long time scales, minimal dynamical impact (Wunsch and Stammer, 1997). Indeed, most, if not all, ocean components of CMIP models ignore the atmospheric loading on the ocean (see Arbic (2005) for an exception).

---

[12]Note that $p_{\text{ice loading}} = 0$ for models that do not depress the sea surface under the weight of sea ice.





- For those models that do apply atmospheric loading, and thus have an inverse barometer response in $\eta_{\text{model}}$, we request that such loading remain part of the dynamic sea level archived in CMIP. That is, the dynamic sea level *will* be depressed or raised according to the weight of the atmosphere. We thus do *not* remove the inverse barometer from atmospheric loading. This treatment for the atmosphere loading contrasts to the recommendation for sea ice loading detailed in Section 5.7.3.

- If the ocean model feels the effects from the applied atmospheric forcing, then include this fact in the "comments" section for zos.

- The key point is that the global area integral of dynamic sea level, zos, should be zero, even if the ocean model feels the weight of the atmosphere.

## 5.8 Squared dynamic sea level

- zossq

The field zossq is the square of the dynamic sea level and accumulated each model time step. This quadratic quantity helps to measure the variability simulated in the dynamic sea level by computing the variance. For that purpose, we can make use of the following identity

$$\text{Var}(\text{zos}) = \frac{1}{T} \int_{-T/2}^{T/2} (\text{zos} - \overline{\text{zos}})^2 \, \mathrm{d}t \tag{16a}$$

$$= \overline{\text{zossq}} - (\overline{\text{zos}})^2, \tag{16b}$$

where $T$ is the time interval for the time average, and $\overline{\text{zos}}$ is the time mean dynamic sea level.

### 5.9 Global thermosteric sea level changes

- zostoga

The potential for increased sea level due to anthropogenic climate change presents some of the most pressing issues for adaptation to a warmer world (Church et al., 2011, 2013a; Gregory et al., 2013). Sea level changes also provide a baseline assessment of the changing ocean climate in the simulations (Yin et al., 2010a; Yin, 2012; Kuhlbrodt and Gregory, 2012). It is thus of primary importance to consider the effects from sea level rise as simulated in the CMIP models. Results from model simulations should be carefully documented in order to properly interpret the CMIP archive.

There are three main reasons for global mean sea level to increase. First, thermal expansion arises due to the warming ocean (thermosteric change). Second, changes in the mass of sea water in the ocean (barystatic change) affects an increase in ocean volume. Third, global halosteric effects (Durack et al., 2014b), though they are far smaller than either thermosteric or barystatic changes.





### 5.9.1 Increases in ocean mass

The mass effect arises most importantly from increasing melt of land glaciers and ice sheets. These changes will be registered by changes in the bottom pressure (pbo in Section 5.1) and ocean mass (masscello in Section 5.3). However, the CMIP6-based global climate models do not generally have reliable values for these contributions. The reason is that the climate models

generally do not include reliable interactive and evolving land glacier and ice sheet models. Estimates for mass effects generally come from specialized process-based models using CMIP scenarios as input (for example, see Church et al. (2013c)). Hence, changes to global sea level due to changes in ocean mass (barystatic sea level change) are not trustworthy within the CMIP6 climate and earth system models.

### 5.9.2 Global halosteric changes

There is notably no significant global mean sea level rise from changes in salinity, given that the global halosteric effect is very tiny in comparison to the global thermosteric effect (see Griffies and Greatbatch (2012) for discussion).[13] Furthermore, net changes from the global halosteric effect in a CMIP simulation are associated with inaccurate estimates of ocean mass changes in these models. In general, the global halosteric effect is only a fraction of the volume change that results from adding freshwater to the ocean (Munk (2003), Lowe and Gregory (2006)).

### 5.9.3 Global thermosteric changes

Thermal expansion of seawater accounts for roughly one-third to one-half of the observed global mean sea level rise in the 20th and early 21st centuries (Church et al., 2011, 2013a; Gregory et al., 2013). Measuring the thermal expansion from CMIP simulations is thus of primary importance.

### 5.9.4 As compared to the CMIP5 request

For the reasons noted above, CMIP6 does *not* ask for the following CMIP5 diagnostics:

- global_average_sea_level_change = zosga
- global_average_steric_sea_level_change = zossga

Rather, only the diagnostic zostoga, measuring global thermal expansion, is requested for CMIP6.

### 5.9.5 Theoretical considerations

To understand the basics of how the global mean sea level changes, we summarize some salient points from Section 4.5 of Griffies and Greatbatch (2012). Here, we consider the relation between the total mass of liquid seawater, total volume of seawater, and global mean seawater density,

$$\mathcal{M} = \mathcal{V} \langle \rho \rangle, \tag{17}$$

[13]Regionally, halosteric effects can be sizable, as recently discussed for models by Griffies et al. (2014) and for observations by Durack et al. (2014b).





where $\mathcal{M}$ is the total liquid ocean mass (masso in equation (7)), $\mathcal{V}$ is the total ocean volume (volo in equation (12)), and $\langle \rho \rangle$ is the global mean *in situ* density

$$\langle \rho \rangle = \frac{\sum \rho \, \mathrm{d}A \, \mathrm{d}z}{\sum \mathrm{d}A \, \mathrm{d}z} = \frac{\mathcal{M}}{\mathcal{V}}. \tag{18}$$

Temporal changes in total ocean mass are affected by a nonzero net mass flux through the ocean boundaries (equation (11))

$$\frac{\mathrm{d}\mathcal{M}}{\mathrm{d}t} = \mathcal{A} \, \overline{Q^{\mathrm{m}}} \tag{19}$$

where $Q^{\mathrm{m}}$ is the mass crossing the ocean surface (the diagnostic wfo in Section 8.2),

$$\overline{Q^{\mathrm{m}}} = \mathcal{A}^{-1} \sum Q^{\mathrm{m}} \, \mathrm{d}A \tag{20}$$

is the global mean mass per horizontal area per time of water crossing the ocean boundaries, with

$$\mathcal{A} = \sum \mathrm{d}A \tag{21}$$

the area of the global ocean surface. Note that for most CMIP6 models, there is no land ice sheet or ice shelf components, in which case the mass flux entering the ocean is missing the ice sheet melt component of sea level rise.

For an ocean with a constant horizontal area (i.e., no wetting and drying, as is the case for typical CMIP models), then temporal changes in the ocean volume are associated with sea level changes via

$$\frac{\mathrm{d}\mathcal{V}}{\mathrm{d}t} = \mathcal{A} \frac{\mathrm{d}\overline{\eta}}{\mathrm{d}t}, \tag{22}$$

where

$$\overline{\eta} = \mathcal{A}^{-1} \sum \eta \, \mathrm{d}A \tag{23}$$

is the global mean sea level.[14] Bringing these results together leads to the evolution equation for the global mean sea level

$$\frac{\mathrm{d}\overline{\eta}}{\mathrm{d}t} = \frac{\overline{Q^{\mathrm{m}}}}{\langle \rho \rangle} - \left( \frac{\mathcal{V}}{\mathcal{A} \langle \rho \rangle} \right) \frac{\mathrm{d}\langle \rho \rangle}{\mathrm{d}t}. \tag{24}$$

The first term in equation (24) alters sea level by adding or subtracting mass from the ocean. The second term arises from temporal changes in the global mean density; i.e., from *steric* effects.

We can approximate each of the terms in equation (24) over a finite time $\Delta t$ via

$$\Delta \overline{\eta} \approx \frac{\overline{Q^{\mathrm{m}}} \, \Delta t}{\langle \rho \rangle} - \left( \frac{\mathcal{V}}{\mathcal{A}} \right) \frac{\Delta \langle \rho \rangle}{\langle \rho \rangle}, \tag{25}$$

where the $\Delta$ operator is a finite difference over the time step of interest. The global steric term is defined by

$$\mathcal{S} \equiv - \left( \frac{\mathcal{V}}{\mathcal{A}} \right) \frac{\Delta \langle \rho \rangle}{\langle \rho \rangle}. \tag{26}$$

---

[14]Contrary to the dynamic sea level $\eta_{\mathrm{omip}}$ considered in Section 5.7, we are interested here in evolution of the global mean of the sea level $\eta$, with this global mean distinctly nonzero due to thermosteric effects.




It is straightforward to diagnose from a model simulation, given temporal changes in the global mean density. Note that this diagnostic is relevant for *both* Boussinesq and non-Boussinesq ocean simulations, as it depends only on changes in the *in situ* density.

For CMIP, we are interested in the change in sea level in a global warming scenario experiment, with respect to a reference state defined by the initial conditions of the experiment. In this case, the steric term at a time $n$ is given by

$$
\begin{aligned}
\mathcal{S} &= -\left(\frac{\mathcal{V}^0}{\mathcal{A}}\right) \frac{\langle \rho^n \rangle - \langle \rho^0 \rangle}{\langle \rho^0 \rangle} \\
&= \left(\frac{\mathcal{V}^0}{\mathcal{A}}\right) \left(1 - \frac{\langle \rho^n \rangle}{\langle \rho^0 \rangle}\right),
\end{aligned}
\tag{27}
$$

where $\rho^0 = \rho(\theta^0, S^0, p^0)$ is the *in situ* density for a grid cell as determined by the grid cell's reference temperature, reference salinity, and reference pressure; $\rho^n = \rho(\theta^n, S^n, p^n)$ is the *in situ* density at time step $n$; and $\mathcal{V}^0$ is the reference global volume of seawater.

As stated earlier, we are most interested in the global steric changes from CMIP models associated with changes in ocean temperature. These thermosteric effects are recorded in zostoga, which represents that part of the global mean sea level change due to changes in ocean density arising just from changes in temperature. We can estimate this thermosteric effect via

$$
\mathcal{S}^{\text{thermo}} = \left(\frac{\mathcal{V}^0}{\mathcal{A}}\right) \left(1 - \frac{\langle \rho(\theta^n, S^0, p^0) \rangle}{\langle \rho^0 \rangle}\right).
\tag{28}
$$

That is, the density in the numerator is computed as a function of the time evolving potential temperature (or Conservative Temperature), with salinity and pressure held constant at their reference value.

There is *no* appearance of sea ice in the expression (28) for thermosteric sea level rise. Sea ice changes impact on the steric sea level only by their impacts on the halosteric effect. As discussed, we are not concerned with global halosteric changes due to the nontrivial uncertainties in land ice melt, with such processes generally not considered in CMIP simulations.

### 5.9.6 Specifying the global thermosteric changes

To compute the global mean thermosteric sea level changes, we need to specify a reference state, as per equation (28).

– For OMIP simulations (global ocean/sea-ice models), we recommend taking the reference state to be that from the start of the first cycle of the forcing. As a comparison, note that Figure 3 of Griffies et al. (2014) shows the global steric rise over five cycles of forcing. The thermosteric rise will be quite close to the steric rise for this simulation.

– For coupled model historical or double $CO_2$ simulations, we recommend taking the reference state to be the first year of the simulation.

### 5.10 Potential temperature of liquid seawater

• thetao



The three dimensional monthly mean potential temperature should be archived, where the reference pressure is at the ocean surface.

Recommendations from IOC et al. (2010) promote the alternative *Conservative Temperature* to measure ocean heat. Conservative Temperature is the potential enthalpy divided by a reference heat capacity (Appendix D). Conservative Temperature is far more conservative than potential temperature, and so provides a solid foundation for prognosing heat movement in the ocean. However, as discussed in Appendix D, for comparison to other models and to observational data, as well as to previous CMIPs, we recommend that ocean components in CMIP6 archive potential temperature thetao, regardless whether the models consider this field as prognostic, or as diagnostic (when Conservative Temperature is prognostic).

### 5.11 Conservative Temperature of liquid seawater

- bigthetao

For models that make use of the TEOS-10 Conservative Temperature as their prognostic field (Appendix D), they should archive the Conservative Temperature. Doing so will allow for meaningful comparison across CMIP6 with future CMIPs, which will predominantly use Conservative Temperature.

### 5.12 Global mean ocean temperature

- thetaoga

- bigthetaoga

In addition to the three-dimensional field of potential temperature, we ask CMIP6 models to archive the global mean potential temperature. Models enacting TEOS-10 (Appendix D) should archive *both* thetaoga and bigthetaoga. These global mean time series provide a measure of the model drift and reflect on the net heating at the ocean boundaries (see below).

### 5.12.1 Summary of the diagnostic

For potential temperature, its global mean has the same standard name as the three-dimensional potential temperature, but is distinguished by the cell methods attribute (area and depth mean). For Conservative Temperature, its global mean is requested just for those models enabling the TEOS-10 thermodynamics.

The calculation of global mean prognostic temperature differs depending on the use of Boussinesq or non-Boussinesq ocean equations. In a non-Boussinesq model, the mean is given by the mass weighted mean

$$\langle T \rangle_{\text{non-Bouss}} = \frac{\sum_{i,j,k} \rho T \, dA \, dz}{\sum_{i,j,k} \rho \, dA \, dz}, \tag{29}$$

where $T$ is the model potential temperature, $\theta$, or Conservative Temperture, $\Theta$. In a Boussinesq model, the mean is computed as the volume weighted mean

$$\langle T \rangle_{\text{Bouss}} = \frac{\sum_{i,j,k} T \, dA \, dz}{\sum_{i,j,k} dA \, dz}. \tag{30}$$





The distinction between non-Boussinesq and Boussinesq models arises from the differences in the underlying conserved fields in the two model formulations. For both cases, it is necessary to accumulate each model time step when producing the time mean, since the mean is built from the product of time dependent terms (e.g., density and grid cell thicknesses are generally time dependent).

Time series of the global mean prognostic temperature provides a measure of simulation drift. Furthermore, as discussed next, when combined with the boundary fluxes and total mass/volume, one can diagnose the degree to which the ocean model conserves heat.

### 5.12.2    Theoretical considerations

According to the results of Griffies et al. (2009b) and Griffies et al. (2014), one should *not* assume that all ocean models are

written with numerical methods that ensure the conservation of scalar fields such as mass, heat, and salt. One means to check for heat conservation is to compute the change in total heat over a specified time (say over a year), and compare that change to the total boundary heat input to the ocean system. The change in heat should agree to the heat input through the boundaries, with agreement to within computational roundoff expected from a conservative model. See Appendix F for dissussion of this point in the context of finite volume scalar budgets.

If there is a difference greater than computational roundoff, then how significant is the difference? To answer this question, consider an order of magnitude calculation to determine the temperature trend that one may expect, given a nonzero net heat flux through the ocean boundaries. For simplicity, assume a Boussinesq fluid with constant volume (i.e., no net volume fluxes), and assume the model prognostic field is potential temperature. The global mean liquid ocean potential temperature evolves according to

$$(\mathcal{V}\rho_o c_p^o) \frac{\mathrm{d}\langle\theta\rangle}{\mathrm{d}t} = \mathcal{A}\,\overline{Q^{\scriptscriptstyle\mathrm{H}}} \tag{31}$$

where $\overline{Q^{\scriptscriptstyle\mathrm{H}}} = \mathcal{A}^{-1}\sum Q^{\scriptscriptstyle\mathrm{H}}\,\mathrm{d}A$ is the global average boundary heat flux. Typical values for the World Ocean yield $\mathcal{V}\rho_o c_p^o \approx 5.4 \times 10^{24}\,\mathrm{J}/^\circ C$ and $\mathcal{A} = 3.6 \times 10^{14}\,\mathrm{m}^2$, leading to the decadal scale potential temperature trend

$$\frac{\Delta\langle\theta\rangle}{\mathrm{decade}} \approx 0.02\,\overline{Q^{\scriptscriptstyle\mathrm{H}}}. \tag{32}$$

For example, with a $1\,\mathrm{W\,m}^{-2}$ ocean area average heating of the ocean over the course of a decade,[15] we expect a global mean

temperature trend of roughly $0.02^\circ C$ per decade, or $0.2^\circ C$ per century. If there is an error in the balance (31), we may define a global mean heat flux

$$Q^{\mathrm{error}} \equiv \overline{Q^{\scriptscriptstyle\mathrm{H}}} - \left(\frac{\mathcal{V}\rho_o c_p^o}{\mathcal{A}}\right)\frac{\mathrm{d}\langle\theta\rangle}{\mathrm{d}t}. \tag{33}$$

---

[15]Otto et al. (2013) infer a net signal from global warming on the order of $0.7\,\mathrm{W\,m}^{-2}$, as averaged over the Earth (ocean + land) surface area. Roemmich et al. (2015) compute warming rates from *in situ* ocean measurements over years 2006–2013 using Argo, with values ranging from $0.35\,\mathrm{W\,m}^{-2}$ to $0.49\,\mathrm{W\,m}^{-2}$ (normalized by Earth surface area). Assuming the 17% of the ocean area not well sampled by Argo warms at the same rate as the observed 83%, the total warming for the upper 2000 m of ocean is $0.4\,\mathrm{W\,m}^{-2}$ to $0.6\,\mathrm{W\,m}^{-2}$.



To translate the error in the net heating into an error in the temperature trend, use relation (32) to define

$$\frac{\Delta \langle \theta \rangle^{\text{error}}}{\text{decade}} \approx 0.02 Q^{\text{error}}. \tag{34}$$

### 5.13 Monthly mean SST of liquid water

- tos

- tosga

In the CMIP archive, it is quite valuable to have the full three-dimensional fields, such as potential temperature and salinity. However, for many purposes, just the top model fields are sufficient. The SST (sea surface temperature) field is archived as tos, and its global area average is tosga

$$\text{tosga} = \frac{\sum_{i,j} \theta_{i,j,k=1} \, \mathrm{d}A}{\sum_{i,j} \mathrm{d}A}. \tag{35}$$

We offer the following points to clarify the SST archived from OMIP simulations.

– SST (tos) is the interface temperature at the upper boundary of the ocean. In regions of open ocean, SST recorded for CMIP is the temperature used by the model to calculate the sensible heat transfer between the ocean surface and air above, and to compute upwelling longwave radiation at the surface. If the ocean is covered with sea ice, tos is the temperature just below the sea ice (used to calculate heat conduction between the two media). In regions of open ocean the "surface_temperature" will be the same as "sea_surface_temperature". However, in the presence of sea ice the two will generally differ. In nearly all ocean climate models, the surface temperature is given by the prognostic temperature in the upper-most ocean grid cell.

– Surface tracer fields produced from a climate model generally do *not* correspond to skin properties. Rather, they are bulk properties averaged over the top grid cell, which is generally no less than a metre thick.

– Since the potential temperature is referenced to 0 gauge pressure, the surface potential temperature is the same as surface *in situ* temperature.

– SST is the surface model value for potential temperature, which is distinct from the surface value of Conservative Temperature (IOC et al., 2010).

### 5.14 Daily mean SST of liquid water

- tos

We recommend that daily mean SST be saved for the purpose of computing space-time diagrams to diagnose propagating signals, such as Tropical Instability Waves. The daily mean SST is also useful for understanding the potential for enhanced



coral bleaching in a warming world. Remotely sensed estimates of coral bleaching have converged on a measure based on degree-heating weeks (Strong et al., 2004). Quantifying this measure in models requires an archive of daily mean sea surface temperatures.

### 5.15 Daily and monthly mean squared SST of liquid water

The quadratic field

- tossq

is accumulated each model time step. It is requested to help measure the variability simulated in the sea surface temperature, so that one may compute the variance (see Section 5.8 for sea level variance).

### 5.16 Bottom potential temperature

- tob

For studies of impacts on ecosystem from climate change, it is useful to measure changes in bottom salinity (Section 5.20) and temperature (Cheung et al., 2013; Gehlen and Dunne, 2014; Saba et al., 2015). As with the request to save SST and SSS, we request for CMIP6 the bottom temperature and bottom salinity in order to facilitate easier analysis using these fields.

### 5.17 Salinity of liquid water

- so

- soga

We request the three dimensional monthly mean ocean salinity field.[16] In addition, as for potential temperature, we recommend saving the global mean salinity of liquid seawater. This mean is computed in a non-Boussinesq model by the mass weighted mean

$$\langle S \rangle_{\text{non-Bouss}} = \frac{\sum_{i,j,k} \rho S \, \mathrm{d}A \, \mathrm{d}z}{\sum_{i,j,k} \rho \, \mathrm{d}A \, \mathrm{d}z}, \tag{36}$$

whereas for a Boussinesq model it is the volume weighted mean

$$\langle S \rangle_{\text{Bouss}} = \frac{\sum_{i,j,k} S \, \mathrm{d}A \, \mathrm{d}z}{\sum_{i,j,k} \mathrm{d}A \, \mathrm{d}z}. \tag{37}$$

In either case, the time series of the global mean salinity provides a measure of simulation drift, and a means to check for conservation of total salt. As for the global mean temperature, it is generally necessary to compute each of the terms in the average on each time step, since the average is generally built from the product of time dependent terms.

---

[16]We discuss salinity and TEOS-10 in Appendix D.





## 5.18 Sea surface salinity (SSS)

- sos

- sosga

The sea surface salinity (SSS) provides a useful means for detecting changes in the high latitude thermohaline forcing, which can present the analyst with a quick diagnosis of whether a simulation is more or less prone to modification of the overturning circulation. For example, fresh water capping can be seen by diagnosis of the SSS. In this case, signals in SSS may motivate more detailed analysis of the three-dimensional fields. In addition, to further reduce the size of the diagnostic, we request the global area average of the SSS, sampled as monthly means

$$\text{sosga} = \frac{\sum_{i,j} \mathrm{d}A\, S_{i,j,k=1}}{\sum_{i,j} \mathrm{d}A}. \tag{38}$$

## 5.19 Daily mean sea surface salinity (SSS)

- sos

Recent remote measures of surface salinity are available from SMOS (Berger et al., 2002), Aquarius (Lagerloef et al., 2008) and SMAP (Piepmeier et al., 2015) satellites. These measures allow for higher temporal features from models to be compared to observations, thus motivating the archival of daily mean SSS from models.

## 5.20 Bottom salinity

- sob

To study impacts on ecosystems from climate change, it is useful to measure changes in bottom temperature (Section 5.16) and salinity (Cheung et al., 2013; Gehlen and Dunne, 2014; Saba et al., 2015).

## 5.21 Squared ocean buoyancy frequency

- obvfsq

This diagnostic is the squared buoyancy frequency in units of $\sec^{-2}$. We recommend use of locally referenced potential density for computing this measure of vertical gravitational stability.

Previous CMIPs requested the potential density referenced to the ocean surface ($\sigma_0$). However, the buoyancy frequency is a more useful diagnostic for measuring vertical stability, which motivated us to replace potential density with squared buoyancy frequency. Additionally, buoyancy frequency is commonly used as part of various ocean parameterizations, such as gravity wave mixing (Simmons et al., 2004; Melet et al., 2013), and mesoscale eddy closures (Gent et al., 1995; Griffies et al., 1998).



## 5.22 Ideal age tracer

- agessc

The ideal age tracer (Thiele and Sarmiento, 1990; England, 1995) provides a useful measure of ocean ventilation (Bryan et al., 2006; Gnanadesikan et al., 2007). This tracer is set to zero in the model surface level/layer at each time step. Beneath the surface level, the ideal age tracer grows older according to the model time. Ideal age is particularly useful for revealing surface-to-deep connections in regions such as the Southern Ocean. It can also be used to estimate uptake of anthropogenic tracers such as carbon dioxide (Russell et al., 2006).

Ideal age satisfies the following advection-diffusion-source equation

$$\frac{\partial A}{\partial t} + \nabla \cdot (\boldsymbol{v} A) = 1 - \nabla \cdot \boldsymbol{F} + \gamma (A^* - A) \delta_{\text{surf}}. \tag{39}$$

In this equation, $A$ is the ideal age with dimensions of time; a unit source (the "1" on the right hand side) adds time to the age tracer over each time step; $\boldsymbol{F}$ is the subgrid scale (SGS) flux; and a damping is applied in a surface region back to $A^* = 0$. The surface damping is often applied just to the top grid cell. Alternatively, it can be applied over a region of specified thickness. If the damping time $\gamma^{-1}$ is zero (infinitely strong damping), then $A = A^* = 0$ is specified for the surface region; i.e., the top cell value of the age tracer is set to $A = 0$. Some groups take $\gamma^{-1} = 0$ whereas others use a finite value. So long as the restoring strength is sufficiently strong, there should be only minor distinctions between the two approaches, although there is no documented study testing this conjecture.

To facilitate direct comparison of ideal age in the different model simulations, we recommend initializing age globally to zero at 1 January 1850 in the historical experiments, or at the start of any of the various scenario experiments. Measuring age in years, rather than seconds, is the traditional approach in ocean modelling, and is recommended for ideal age in CMIP6.

## 5.23 Mixed layer depth

- mlotst

### 5.23.1 Summary of the diagnostic

An assessment of model mixed layer depth (MLD) is useful for understanding how water-mass formation in the simulations is regulated by upper ocean stratification and surface water overturn. Unfortunately, there is no universally agreed upon criterion for defining the mixed layer depth. For the purpose of fostering a consistent comparison of simulated mixed layers from ocean model components in CMIP6, the "sigma-t" criterion introduced by Levitus (1982) should be followed.

### 5.23.2 Theoretical and practical considerations

The mixed layer depth is based on measuring ocean gravitational stability under a vertical displacement. To determine whether vertical transfer is favoured requires a thought experiment, in which a surface ocean fluid parcel is displaced downward without





changing its temperature or salinity, but feeling the local *in situ* pressure. If the density of the displaced parcel is sufficiently far from the local *in situ* density, then the displacement is not favoured, and we are thus beneath the mixed layer and into the stratified interior. What determines "sufficiently far" is subjective, with convention determining the precise value.

The mixed layer has near-zero vertical gradients of temperature, salinity, and density, as well as tracers such as CFCs. So
most techniques to estimate the MLD rely on either a threshold gradient or a threshold change in one of these quantities, normally in potential temperature $\theta$ or density (see, for example Lorbacher et al., 2006; de Boyer Montégut et al., 2004; Monterey and Levitus, 1997). Relying solely on $\theta$ has the advantage of good observational data coverage, but this approach neglects salinity stratification associated with barrier layers (see e.g., Sprintall and Tomczak, 1992) and high latitudes where salinity greatly impacts on density. In contrast, relying solely on density overlooks density-compensating changes in $\theta - S$,
thus over-estimating the thickness of the mixed layer.

The method we recommend for OMIP comes from Levitus (1982). Here, the MLD is defined based on meeting a "sigma-t" criterion. This method is readily employed in off-line mode, thus supporting the use of monthly mean model fields. However, we recommend computing mlotst online in order to avoid aliasing. Also, we ask for the squared mixed layer depth mlotstsq (see Section 5.24) to allow for computation of the variance. The variance calculation is served best by online mixed layer depth
calculations.

We here provide some details for the diagnostic. Mathematically, we compute the difference between the following two densities

$$\rho_{\text{displaced from surface}} = \rho[S(k=1), \Theta(k=1), p(k)] \tag{40a}$$

$$\rho_{\text{local}} = \rho[S(k), \Theta(k), p(k)], \tag{40b}$$

and convert that density difference to a buoyancy difference

$$\delta B = -\left( \frac{g \left( \rho_{\text{displaced from surface}} - \rho_{\text{local}} \right)}{\rho_{\text{local}}} \right). \tag{41}$$

This buoyancy difference is computed from the surface down to the first depth at which $\delta B > \Delta B_{\text{crit}}$, where the OMIP recommended value is

$$\Delta B_{\text{crit}} = 0.0003 \text{ m s}^{-2}, \tag{42}$$

with this value also used in Levitus (1982). Other values may be more suitable for regional studies, such as for the Southern Ocean. The mixed layer depth, $H^{(\text{mld})}(x, y, t)$ is then approximated by interpolating between the depth where $\delta B > \Delta B_{\text{crit}}$ and the shallower depth.[17] With $g = 9.8 \text{ m s}^{-2}$ and $\rho_{\text{local}} \approx 1035 \text{ kg m}^{-3}$, then $\Delta B_{\text{crit}} = 0.0003 \text{ m s}^{-2}$ corresponds to a critical density difference of

$$\Delta \rho_{\text{crit}} = 0.03 \text{ kg m}^{-3}, \tag{43}$$

---

[17]Linear interpolation is common, though other methods may be used depending on modeller preference.





as used by de Boyer Montégut et al. (2004). Note that some studies employ the larger critical value, $\Delta\rho_{\text{crit}} = 0.125 \text{ kg m}^{-3}$, which will generally result in a deeper mixed layer depth due to the need to penetrate deeper into the stratified water. The choice is subjective, but should be compatible across models and observations to ensure suitable comparisons.

### 5.24 Squared mixed layer depth

- mlotstsq

This diagnostic is the square of mlotst (Section 5.23). Diagnosing both mlotst and mlotstsq online during each model time step allows one to compute variance of the mixed layer depth (see Section 5.8 for the analogous calculation of dynamic sea level variance).

### 5.25 Barotropic or quasi-barotropic streamfunction

- msftbarot

#### 5.25.1 Summary of the diagnostic

The barotropic streamfunction is a useful field for mapping the vertically integrated fluid transport. However, many ocean models have jettisoned the rigid lid assumption of Bryan (1969) for both computational and physical reasons. Absent a rigid lid assumption, the vertically integrated mass transport[18]

$$\boldsymbol{U}^\rho = \int\limits_{-H}^{\eta} \rho\boldsymbol{u}\,\mathrm{d}z \tag{44}$$

generally has a non-zero divergence, thus precluding it from being fully specified by a single scalar field. Instead, both a streamfunction and velocity potential are needed to specify the transport. For those models that do not compute a barotropic streamfunction, we introduce the notion of a *quasi-barotropic streamfunction* $\psi^U$ in the following theoretical considerations, with this field serving as a useful approximate alternative to the barotropic streamfunction.

In summary, we request either of the following scalar fields be archived for purposes of mapping the vertically integrated mass transport:

- Barotropic streamfunction for those models that compute this function using an elliptic solver;

- The quasi-barotropic streamfunction $\psi^U$ for cases when the model does not distinguish the streamfunction from the velocity potential.

We recommend that the dimensions of the streamfunction be mass transport (kg/s), rather than volume transport (m$^3$/s) (see start of Section 6).

---

[18]The density factor $\rho$ in a non-Boussinesq fluid becomes the constant $\rho_o$ for Boussinesq fluids.




### 5.25.2 Theoretical considerations

For a mass conserving non-Boussinesq fluid, the vertically integrated mass transport $\boldsymbol{U}^\rho = \int_{-H}^{\eta} \boldsymbol{u}\,\rho\,\mathrm{d}z$ has a divergence given by

$$\nabla \cdot \boldsymbol{U}^\rho = -\frac{\partial (D\,\langle\rho\rangle)}{\partial t} + Q^{\mathrm{m}}, \tag{45}$$

where $D = H + \eta$ is the thickness of a fluid column. Similarly, for a Boussinesq fluid the depth integrated velocity $\boldsymbol{U} = \int_{-H}^{\eta} \boldsymbol{u}\,\mathrm{d}z$ has a divergence

$$\rho_o \nabla \cdot \boldsymbol{U} = -\rho_o \left(\frac{\partial \eta}{\partial t}\right) + Q^{\mathrm{m}}. \tag{46}$$

Given that neither $\boldsymbol{U}^\rho$ nor $\boldsymbol{U}$ are non-divergent, a barotropic streamfunction is insufficient to fully describe the vertically integrated flow. In general, it is necessary to solve an elliptic boundary value problem to diagnose the barotropic streamfunction. However, for CMIP purposes, it is sufficient to compute an approximate streamfunction, with details now given.

Consider the function

$$\psi^U(x,y) = -\int_{y_o}^{y} U^\rho(x,y')\,\mathrm{d}y', \tag{47}$$

where the southern limit $y_o$ is at Antarctica. Note that all intermediate ranges of latitude bands are included, so there are no shadow regions that may otherwise be isolated due to land/sea arrangements. By definition, the $y$ derivative $\psi^U(x,y)$ yields the transport in the $\hat{\boldsymbol{x}}$ (eastward) direction

$$\frac{\partial \psi^U}{\partial y} = -U^\rho, \tag{48}$$

yet the $x$ derivative does not yield the $\hat{\boldsymbol{y}}$-transport due to the divergent nature of the vertically integrated flow. A complement function

$$\psi^V(x,y) = \psi^U(x_o,y) + \int_{x_o}^{x} V^\rho(x',y)\,\mathrm{d}x', \tag{49}$$

yields $\partial_x \psi^V = V^\rho$. In the special case of a Boussinesq rigid lid model absent surface water fluxes, $\psi^U$ and $\psi^V$ reduce to the single rigid lid barotropic streamfunction. In the more general case, comparison of $\psi^U$ and $\psi^V$ in climate model simulations at GFDL reveal that after just a few years of spin-up, patterns for the monthly means of $\psi^U$ and $\psi^V$ are very similar. This result provides evidence that much of the large-scale vertically integrated circulation is nearly non-divergent. In this case, either function $\psi^U$ and $\psi^V$ renders a useful map of the vertically integrated mass transport. Due to its simplicity, we recommend that the quasi-barotropic streamfunction $\psi^U$ be archived for CMIP6.

## 6 Vector fields

We now consider components to vector fields, with a summary of the diagnostics given in Table 3. Refer to Section B3 for a discussion of remapping vector fields, and Section B4 for remapping of transport components to estimate meridional



transports. In general, remapping vector fields is less straightforward than remapping scalar fields. Modellers may thus choose to only submit native grid vector components, even though spherical grids greatly facilitates comparative analyses.

## 6.1  Residual mean velocity and transport units

The *residual mean velocity*, $\boldsymbol{v}^{\dagger}$, transports seawater mass and tracer in an ocean model, where

$$\boldsymbol{v}^{\dagger} = \boldsymbol{v} + \boldsymbol{v}^{*} \tag{50}$$

is the sum of the model prognostic velocity $\boldsymbol{v}$ (the Eulerian mean) plus a parameterized eddy-induced velocity, $\boldsymbol{v}^{*}$. We identify two commonly used eddy-induced velocities.

- MESOSCALE: The parameterized eddy-induced velocity for mesoscale processes commonly follows that suggested by Gent et al. (1995) or related methods such as Ferrari et al. (2010).

- SUBMESOSCALE: Fox-Kemper et al. (2008, 2011) propose a parameterized eddy-induced velocity for mixed layer submesoscale processes.

We know of no other processes now commonly parameterized in global climate simulations according to eddy-induced transport, though such may appear in the future. Notably, if there are no parameterizations of eddy-induced transport, $\boldsymbol{v}^{*} = 0$ so that the residual mean is the Eulerian mean, $\boldsymbol{v}^{\dagger} = \boldsymbol{v}$. We focus our transport diagnostics on the residual mean field, though we still recommend archiving the raw horizontal velocity field, $\boldsymbol{u}$ in Section 6.2.

The mass transport

$$\mathcal{V}^{(\hat{\boldsymbol{n}})} = \rho\,\boldsymbol{v}^{\dagger} \cdot \hat{\boldsymbol{n}}\,\mathrm{d}A \tag{51}$$

measures the mass per time passing through the $\hat{\boldsymbol{n}}$ face of a grid cell, with $\mathrm{d}A$ the area of the cell face and $\hat{\boldsymbol{n}}$ the outward normal. This transport is conveniently quantified using the mass Sverdrup

$$\text{mass Sv} = 10^{9}\,\text{kg s}^{-1} \tag{52}$$

rather than the volume Sverdrup

$$\text{volume Sv} = 10^{6}\,\text{m}^{3}\,\text{s}^{-1}. \tag{53}$$

For Boussinesq fluids, the density factor $\rho$ becomes a constant reference density $\rho_{o}$ (see rhozero in Table 1), which trivially allows for use of the mass Sverdrup as the unit of transport in Boussinesq fluids. Therefore, we request archiving mass transport in the units kg/s rather than a volume transport (m$^{3}$/s).

## 6.2  Horizontal velocity field from resolved flow

- uo = zonal velocity component ($\hat{\boldsymbol{x}} \cdot \boldsymbol{v}$)





**VECTOR FIELD COMPONENTS**

| ITEM | CMOR NAME | SPONSOR | CMIP5/CMIP6 | UNITS | TIME | SHAPE | GRID | PRIORITY | EXPT |
|---|---|---|---|---|---|---|---|---|---|
| 1 | uo | OMIP | same | m/s | month | XYZ | sphere (native), z/p | 1 | all |
| 2 | vo | OMIP | same | m/s | month | XYZ | sphere (native), z/p | 1 | all |
| 3 | umo | OMIP | resolved + parameterized | kg/s | month | XYZ | sphere (native), z/p | 1 | all |
| 4 | vmo | OMIP | resolved + parameterized | kg/s | month | XYZ | sphere (native), z/p | 1 | all |
| 5 | wmo | OMIP | resolved + parameterized | kg/s | month | XYZ | sphere (native), z/p | 1 | all |
| 6 | msftmyz | OMIP | same | kg/s | month | YZ-basin | (latitude,z/p) | 1 | all |
| 7 | msftmrho | OMIP | msftmrhoz $\longrightarrow$ msftmrho | kg/s | month | Y$\rho$-basin | (latitude,$\rho$) | 1 | all |
| 8 | msftyyz | OMIP | same | kg/s | month | YZ-basin | (native,z/p) | 1 | all |
| 9 | msftyrho | OMIP | msftyrhoz $\longrightarrow$ msftyrho | kg/s | month | Y$\rho$-basin | (native,$\rho$) | 1 | all |
| 10 | msftmzmpa | OMIP | new | kg/s | month | YZ-basin | (latitude,z/p) | 1 | all |
| 11 | msftmrhompa | OMIP | new | kg/s | month | Y$\rho$-basin | (latitude,$\rho$) | 1 | all |
| 12 | msftympa | OMIP | new | kg/s | month | YZ-basin | (native,z/p) | 1 | all |
| 13 | msftyrhompa | OMIP | new | kg/s | month | Y$\rho$-basin | (native, $\rho$) | 1 | all |
| 14 | msftmzsmpa | OMIP | new | kg/s | month | YZ-basin | (latitude,z/p) | 1 | all |
| 15 | msftyzsmpa | OMIP | new | kg/s | month | YZ-basin | (native, z/p) | 1 | all |
| 16 | hfx | OMIP | same | W | month | XYZ | (native, z/p) | 2 | all |
| 17 | hfy | OMIP | same | W | month | XYZ | (native, z/p) | 2 | all |
| 18 | hfbasin | OMIP | same | W | month | Y-basin | latitude | 1 | all |
| 19 | hfbasinpmadv | OMIP | new | W | month | Y-basin | latitude | 1 | all |
| 20 | hfbasinpsmadv | OMIP | new | W | month | Y-basin | latitude | 1 | all |
| 21 | hfbasinpmdiff | OMIP | new | W | month | Y-basin | latitude | 1 | all |
| 22 | hfbasinpadv | OMIP | new | W | month | Y-basin | latitude | 1 | all |
| 23 | htovgyre | OMIP | same | W | month | Y-basin | latitude | 2 | all |
| 24 | htovovrt | OMIP | same | W | month | Y-basin | latitude | 2 | all |
| 25 | sltovgyre | OMIP | same | kg/s | month | Y-basin | latitude | 2 | all |
| 26 | sltovovrt | OMIP | same | kg/s | month | Y-basin | latitude | 2 | all |

| | CMOR NAME RELATED TO CF STANDARD NAME | |
|---|---|---|
| ITEM | CMOR NAME | CF STANDARD NAME |
| 1 | uo | sea_water_x_velocity |
| 2 | vo | sea_water_y_velocity |
| 3 | umo | ocean_mass_x_transport |
| 4 | vmo | ocean_mass_y_transport |
| 5 | wmo | upward_ocean_mass_transport |
| 6 | msftmyz | ocean_meridional_overturning_mass_streamfunction |
| 7 | msftmrho | ocean_meridional_overturning_mass_streamfunction |
| 8 | msftyyz | ocean_y_overturning_mass_streamfunction |
| 9 | msftyrho | ocean_y_overturning_mass_streamfunction |
| 10 | msftmzmpa | ocean_meridional_ overturning_mass_streamfunction_due_to_parameterized_mesoscale_advection |
| 11 | msftmrhompa | ocean_meridional_ overturning_mass_streamfunction_due_to_parameterized_mesoscale_advection |
| 12 | msftympa | ocean_y_overturning_mass_streamfunction_due_to_parameterized_mesoscale_advection |
| 13 | msftyrhompa | ocean_y_overturning_mass_streamfunction_due_to_parameterized_mesoscale_advection |
| 14 | msftmzsmpa | ocean_meridional_overturning_mass_streamfunction_due_to_parameterized_submesoscale_advection |
| 15 | msftyzsmpa | ocean_y_overturning_mass_streamfunction_due_to_parameterized_submesoscale_advection |
| 16 | hfx | ocean_heat_x_transport |
| 17 | hfy | ocean_heat_y_transport |
| 18 | hfbasin | northward_ocean_heat_transport |
| 19 | hfbasinpmadv | northward_ocean_heat_transport_due_to_parameterized_mesoscale_advection |
| 20 | hfbasinpsmadv | northward_ocean_heat_transport_due_to_parameterized_submesoscale_advection |
| 21 | hfbasinpmdiff | northward_ocean_heat_transport_due_to_parameterized_mesocale_diffusion |
| 22 | hfbasinpadv | northward_ocean_heat_transport_due_to_parameterized_eddy_advection |
| 23 | htovgyre | northward_ocean_heat_transport_due_to_gyre |
| 24 | htovovrt | northward_ocean_heat_transport_due_to_overturning |
| 25 | sltovgyre | northward_ocean_salt_transport_due_to_gyre |
| 26 | sltovovrt | northward_ocean_salt_transport_due_to_overturning |

**Table 3.** Diagnostic table for vector components, including a sub-table relating the CMOR name to its CF standard name. Entries with grids denoted "sphere (native)" denote diagnostics where spherical output is recommended to facilitate analysis, though where native output will be accepted if spherical is unavailable. The column indicating the experiment for saving the diagnostics generally says "all", in which case we recommend the diagnostic be saved for CMIP6 experiments in which there is an ocean model component, including the DECK, historical simulations, FAFMIP, DAMIP, DCPP, ScenarioMIP, and C4MIP, as well as the ocean-sea ice OMIP simulations. The priority 1 diagnostics should be saved as decadal time means at decadal intervals for the piControl-spinup. Certain of the fields in this table should be partitioned into Atlantic-Arctic, Indian-Pacific, and Global regions. Spherical remapping (mapping to north-south and east-west vector components) is discussed in Sections B3 and B4.





- vo = meridional velocity component ($\hat{\boldsymbol{y}} \cdot \boldsymbol{v}$)

These diagnostics save the horizontal velocity components, as diagnosed from the velocity field time stepped as part of the model prognostic equations. This diagnostic does *not* include any extra velocity that may arise from parameterized subgrid scale eddy-advection.

## 6.3 Horizontal residual mean mass transport

- umo = $\rho\, u^{\dagger}\, \mathrm{d}y\, \mathrm{d}z$

- vmo = $\rho\, v^{\dagger}\, \mathrm{d}x\, \mathrm{d}z$

This diagnostic asks for the horizontal mass transport through faces of a grid cell, where transport arises from the residual mean (sum of Eulerian plus parameterized eddy-induced, as discussed in Section 6.1).

## 6.4 Vertical residual mean mass transport

- wmo = $\rho\, w^{\dagger}\, \mathrm{d}x\, \mathrm{d}y$.

This diagnostic is the vertical mass transport across the model coordinate surface, diagnosed from the model residual mean velocity field (sum of Eulerian plus parameterized eddy-induced, as discussed in Section 6.1). This mass transport (measured in kg/s) is more valuable for analysis than the vertical velocity component. However, if there is reason to determine the vertical residual velocity, it can be diagnosed by

$$w^{\dagger} = \frac{\mathrm{wmo}}{\rho\, \mathrm{d}x\, \mathrm{d}y}. \tag{54}$$

For a Boussinesq model, the $\rho$ factor becomes the constant $\rho_o$, in which case the diagnostic produces an exact expression for the time mean vertical velocity component.

## 6.5 Meridional and $\hat{\boldsymbol{y}}$-ward overturning streamfunction from residual mean transport

- msftmyz = meridional-depth mass streamfunction

- msftmrho = meridional-density mass streamfunction

- msftyyz = y-depth mass streamfunction

- msftyrho = y-density mass streamfunction

We have interest in diagnosing the meridional transport of fluid by the residual mean velocity (sum of Eulerian plus parameterized eddy-induced, as discussed in Section 6.1) in each of the basins Atlantic-Arctic, Indian-Pacific, and World Ocean. To separate the Indian and Pacific Oceans is not sensible, since there is no meridional boundary separating these basins. Instead,





the Atlantic-Arctic, Indian-Pacific, and World Ocean are the only three physically relevant partitions available. We ask for the transport as a function of depth/pressure as well as potential density referenced to 2000 db.

The issue of generalized horizontal coordinates adds complexity to the diagnosis of the northward mass transport when using non-spherical grids. As stated in Appendix B4, instead of remapping mass fluxes to a spherical grid, and then computing
the basin transports, we recommend computing the transports across native grid lines that approximate latitude circles, and reporting these as a function of latitude. Such algorithms can be implemented in a conservative manner for finite volume based models, even those with complex grids (e.g., see Figure C2 of Forget et al. (2015)). Finite element models, in contrast, require extra care (Sidorenko et al., 2009).

For those models using a non-spherical coordinate horizontal grid, in addition to archiving the meridional overturning
streamfunction, we recommend archiving the model native grid $\hat{y}$-ward overturning streamfunction, where $(\hat{x}, \hat{y})$ are directions defined according to the model native grid. We also use the synonyms $(iward, jward)$, using the familiar $(i, j)$ notation for horizontal grid indices. For many purposes and for many of the most commonly used non-spherical structured grids, the $\hat{y}$-ward native grid streamfunction is sufficient since it closely approximates the spherical meridional streamfunction.

A general expression for the ocean mass transport overturning streamfunction is given by

$$
\quad \Psi(y, s, t) = -\int\limits_{x_a}^{x_b} \mathrm{d}x \int\limits_{-H}^{z(s)} \rho \, v^{\dagger} \, \mathrm{d}z, \tag{55}
$$

where $v^{\dagger}$ is the meridional residual mean velocity (see the vmo diagnostic in Section 6.3). $\Psi$ is in fact a transport streamfunction only for the rigid lid Boussinesq case. We nonetheless retain the name "streamfunction" for historical reasons. Note that the zonal integral is computed along surfaces of constant $s$, where $s$ is either a geopotential/pressure surface, or a potential density surface. That is, we recommend that the following versions of the overturning streamfunction be archived at monthly time
averages in the CMIP6 repository, with results for the Atlantic-Arctic, Indian-Pacific, and Global Oceans:

– **meridional-depth** overturning streamfunction and $\hat{y}$-**ward-depth** overturning streamfunction: The depth $z(s)$ corresponds to either the depth of a geopotential or the depth of a pressure surface, depending on whether the model is Boussinesq or non-Boussinesq, respectively.

– **meridional-density** overturning streamfunction and $\hat{y}$-**ward-density** overturning streamfunction: The depth $z(s)$ cor-
responds to the depth of a predefined set of $\sigma_{2000}$ isopycnals, with the definition of these isopycnals at the modeler's discretion. This field presents complementary information relative to the $\hat{y}$-ward-depth overturning streamfunction, and is very useful particularly for diagnosing water mass transformation processes.[19]

– Consistent with the discussion in Section 3.2, it is critical that the time average of the streamfunction be accumulated using each model time step, in order to avoid problems with aliasing and problems ignoring correlations.

---

[19]We do not request plotting overturning on the neutral density coordinate from McDougall and Jackett (2005) in order to facilitate direct comparison of the density overturning streamfunction between isopycnal models, which are based on $\sigma_{2000}$, and non-isopycnal models. Additionally, the McDougall and Jackett (2005) neutral density is based on present-day observational properties, which becomes less relevant for climate change simulations.





### 6.6 Meridional and $\hat{y}$-ward overturning streamfunction from SGS processes

- msftmzmpa = meridional-depth mass streamfunction from parameterized mesoscale

- msftmrhompa = meridional-density mass streamfunction from parameterized mesoscale

- msftympa = y-depth mass streamfunction from parameterized mesoscale

- msftyrhompa = y-density mass streamfunction from parameterized mesoscale

- msftmzsmpa = meridional-depth mass streamfunction from parameterized sub-mesoscale

- msftyzsmpa =y-depth mass streamfunction from parameterized sub-mesoscale

We follow the same philosophy as in Section 6.5 for here diagnosing the meridional and $\hat{y}$-ward overturning streamfunction arising from parameterized subgrid scale (SGS) transport. The following points should be considered for this diagnostic.

– The CMIP5 CF standard name for these fields is "bolus_advection". The new CMIP6 names in Table 3 are preferable since "bolus" advection is a term of limited applicability.

– Gent et al. (1995) represents the canonical parameterization scheme for mesoscale eddies. It is the mass transport from this, or alternative mesoscale closures, that should be archived in the fields "due_to_parameterized_mesoscale_advection". For the Gent et al. (1995) streamfunction, it is useful to map this in both depth and density space.

– Fox-Kemper et al. (2008, 2011) represents the canonical parameterization scheme for mixed layer submesoscale transport. It is the mass transport from this, or alternative submesoscale closures, that should be archived in the fields "due_to_parameterized_submesoscale_advection". Note that since the Fox-Kemper et al. (2008, 2011) scheme applies only in the mixed layer, only its meridional-depth and $\hat{y}$-depth version is relevant.

– For the Gent et al. (1995) parameterization, the meridional overturning streamfunction for mass transport is

$$\Psi^{\mathrm{gm}}(y,s,t) = -\int_{x_a}^{x_b}\mathrm{d}x\int_{-H}^{z(s)}\rho\,v^{\mathrm{gm}}\,\mathrm{d}z \tag{56a}$$

$$= \int_{x_a}^{x_b}\mathrm{d}x\int_{-H}^{z(s)}\partial_z\left(\rho\,\kappa_{\mathrm{gm}}S^y\right)\mathrm{d}z \tag{56b}$$

$$= \int_{x_a}^{x_b}\rho\,\kappa_{\mathrm{gm}}S^y(z(s))\,\mathrm{d}x, \tag{56c}$$

where $\kappa_{\mathrm{gm}} > 0$ is the eddy diffusivity, $S^y$ is the $\hat{y}$ neutral slope, and $\kappa_{\mathrm{gm}}$ vanishes at the ocean bottom. As for the residual mean streamfunction $\Psi$ defined by equation (55), we recommend archiving $\Psi^{\mathrm{gm}}$ on both depth/pressure levels and isopycnal ($\sigma_{2000}$) levels.



## 6.7 Net heat transport from resolved + parameterized processes

- hfx = x-component to net ocean heat transport

- hfy = y-component to net ocean heat transport

- hfbasin = northward net ocean heat transport integrated in basins

There are many processes in the ocean that affect heat transport: resolved advective transport, diffusion, parameterized eddy-induced advection or skew transport, overflow parameterizations, etc. In the analysis of ocean model simulations, it is useful to have a measure of each component of the heat transport, particularly in the horizontal. We follow CMIP5 in requesting the vertically integrated $\hat{x}$-ward and $\hat{y}$-ward heat transport from *all* ocean processes. The horizontal components to this depth integrated heat transport are archived in hfx and hfy. Note that the heat transports are computed using the celsius temperature scale.

Following from the approach taken for the meridional overturning streamfunction, each ocean model using non-spherical coordinate horizontal grids should compute the northward heat transport in each of the basins (northward_ocean_heat_transport), approximated using the model native grid fields without remapping. For models using a spherical latitude-longitude grid, there will be no difference. The approximated poleward transport in non-spherical grids will generally consist of transports crossing a "zig-zag" path (Appendix B4). The resulting poleward heat transport should be reported as a function of latitude, with latitudinal resolution comparable to the model native grid resolution.

## 6.8 Advective heat transport from parameterized mesocale and submesoscale processes

- hfbasinpmadv = northward heat transport from parameterized mesoscale advection

- hfbasinpsmadv = northward heat transport from parameterized submesoscale advection

- hfbasinpmdiff = northward heat transport from parameterized mesoscale diffusion

- hfbasinpadv = northward heat transport from parameterized advection (meso + submeso)

In support of understanding the importance of various subgrid scale (SGS) parameterizations, we recommend that depth and basin integrated northward heat transports should be archived for the Atlantic-Arctic, Indian-Pacific, and World Ocean basins. Additional notes for this diagnostic follow.

– Parameterized SGS advection from mesoscale closures (such as Gent et al. (1995)) and submesocale closures (such as Fox-Kemper et al. (2008, 2011)) are included. They occur with the suffix "advection", even if the implementation of the schemes appears as a skew transport.

– If the eddy-induced advection from the mesoscale and submesoscale closures are combined operationally in the model, and cannot be separately diagnosed, then their net effect is archived in fields with suffix "due_to_parameterized_eddy_advection".



- In addition to eddy-induced advection, mesocale eddies are commonly parameterized through neutral diffusion as in Solomon (1971) and Redi (1982). Contributions to heat transport from neutral diffusion should be placed in the fields with suffix "due_to_parameterized_mesoscale_diffusion".

- The vertically integrated northward transports can be approximated using the a "zig-zag" path method discussed in Appendix B4. The components should be archived as monthly means for the Atlantic-Arctic, Indian-Pacific, and World Ocean. The transports should be reported as a function of latitude, with the latitudinal spacing comparable to the model native grid spacing.

## 6.9 Gyre and overturning decomposition of heat & salt residual mean advective transport

- htovgyre = northward heat transport from gyres

- htovovrt = northward heat transport from overturning

- sltovgyre = northward salt transport from gyres

- sltovovrt = northward salt transport from overturning

### 6.9.1 Summary of the diagnostic

The $\hat{\boldsymbol{y}}$-ward advective transport of a tracer within a particular ocean basin is given by the integral

$$\mathcal{H}^{(\hat{\boldsymbol{y}})}(y,t) = \int\limits_{x_1}^{x_2} \mathrm{d}x \int\limits_{-H}^{\eta} \rho\, C\, v^\dagger\, \mathrm{d}z, \tag{57}$$

where $C$ is the tracer concentration, $v^\dagger$ is the residual mean meridional velocity component (sum of resolved plus parameterized advection), $z = -H(x,y)$ is the ocean bottom, $z = \eta(x,y,t)$ is the ocean free surface, and $x_1$ and $x_2$ are the boundaries of the basin or global ocean. It is useful for some analysis to decompose the transport (57) into "gyre" and "overturning" components, with these terms defined in the following. See Section 3.1.1 of Farneti and Vallis (2009) for a recent example of this diagnostic in use.

We recommend that the monthly means for the components to heat and salt transport be archived, partitioned according to Atlantic-Arctic, Indian-Pacific, and Global Oceans The transports should be reported as a function of latitude, with the latitudinal spacing comparable to the model native grid spacing.

### 6.9.2 Theoretical considerations

The total mass transport leaving the $\hat{\boldsymbol{y}}$-ward face of a grid cell is written

$$V\, \mathrm{d}x = v^\dagger \rho\, \mathrm{d}z\, \mathrm{d}x, \tag{58}$$





and so $C\,V\,\mathrm{d}x$ measures the mass per time (kg/s) or heat per time (watt) of tracer leaving the $\hat{\boldsymbol{y}}$-ward face, including transport from resolved and parameterized advection. We now consider a decomposition of this transport by defining the basin average transport and basin average tracer concentration as follows

$$[V] = \frac{\sum_i V\,\mathrm{d}x}{\sum_i \mathrm{d}x} \tag{59a}$$

$$[C] = \frac{\sum_i C\,\mathrm{d}x}{\sum_i \mathrm{d}x}, \tag{59b}$$

along with the deviations from basin average

$$V = [V] + V^* \tag{60a}$$

$$C = [C] + C^*. \tag{60b}$$

The discrete $i$-sum extends over the basin or global domain of interest, so that $\sum_i \mathrm{d}x\, V$ is the total $\hat{\boldsymbol{y}}$-ward transport of seawater at this band at a particular ocean model vertical level. The resulting $\hat{\boldsymbol{y}}$-ward tracer transport becomes

$$\mathcal{H}(y,t) = \sum_{i,k} V\,C\,\mathrm{d}x = \sum_{i,k} ([V]\,[C] + V^*\,C^*)\,\mathrm{d}x, \tag{61}$$

where the $k$ sum extends over the vertical cells in a column. It is common to identify three components:

$$\text{y\_flux\_advect} = \sum_i \sum_k V\,C\,\mathrm{d}x \tag{62}$$

$$\text{y\_flux\_over} = \sum_i \sum_k [V]\,[C]\,\mathrm{d}x \tag{63}$$

$$\text{y\_flux\_gyre} = \sum_i \sum_k V^*\,C^*\,\mathrm{d}x, \tag{64}$$

with

$$\text{y\_flux\_gyre} = \text{y\_flux\_advect} - \text{y\_flux\_over}. \tag{65}$$

This identity follows very simply when the advective flux takes on the form of either first order upwind or second order centered differences. It becomes more complex when considering higher order, or flux limited, advection schemes. In the more general cases, the expression (65) serves to define the gyre transport component. In this way, the advective flux is built from the advection scheme used in the ocean model.

## 7 Mass transports through pre-defined transects

- mfo

There are a number of climatologically important straits, throughflows, and current systems whose mass transport has been measured observationally.[20] These mass transports provide a useful means to characterize the rates that water flows through key

---

[20]The CLIVAR Ocean Model Development Panel maintains the REOS website, http://www.clivar.org/clivar-panels/omdp/reos, (Repository for Evaluating Ocean Simulations), on which the transports in Table 4 are listed, with updates provided when available.



| MASS TRANSPORT THROUGH SECTIONS | | | | | | | | |
|---|---|---|---|---|---|---|---|---|
| ITEM | CMOR NAME | GEOGRAPHICAL ENDPOINTS | DEPTH | SPONSOR | CMIP5/CMIP6 | UNITS | TIME | PRIORITY | EXPT |
| 1 | mfo (barents_opening) | $(16.8^{\circ}E, 76.5^{\circ}N)$ $(19.2^{\circ}E, 70.2^{\circ}N)$ | full | OMIP | same | kg/s | month | 2 | all |
| 2 | mfo (bering_strait) | $(171^{\circ}W, 66.2^{\circ}N)$ $(166^{\circ}W, 65^{\circ}N)$ | full | OMIP | same | kg/s | month | 2 | all |
| 3 | mfo (caribbean_windward_passage) | $(75^{\circ}W, 20.2^{\circ}N)$ $(72.6^{\circ}W, 19.7^{\circ}N)$ | full | OMIP | same | kg/s | month | 2 | all |
| 4 | mfo (davis_strait ) | $(50^{\circ}W, 65^{\circ}N)$ $(65^{\circ}W, 65^{\circ}N)$ | full | OMIP | new | kg/s | month | 2 | all |
| 5 | mfo (denmark_strait) | $(37^{\circ}W, 66.1^{\circ}N)$ $(22.5^{\circ}W, 66^{\circ}N)$ | full | OMIP | same | kg/s | month | 2 | all |
| 6 | mfo (drake_passage) | $(68^{\circ}W, 54^{\circ}S)$ $(60^{\circ}W, 64.7^{\circ}S)$ | full | OMIP | same | kg/s | month | 2 | all |
| 7 | mfo (english_channel) | $(1.5^{\circ}E, 51.1^{\circ}N$ $(1.7^{\circ}E, 51.0^{\circ}N)$ | full | OMIP | same | kg/s | month | 2 | all |
| 8 | mfo (faroe_scotland_channel) | $(6.9^{\circ}W, 62^{\circ}N)$ $(5^{\circ}W, 58.7^{\circ}N)$ | full | OMIP | same | kg/s | month | 2 | all |
| 9 | mfo (florida_bahamas_strait) | $(78.5^{\circ}W, 26^{\circ}N)$ $(80.5^{\circ}W, 27^{\circ}N)$ | full | OMIP | same | kg/s | month | 2 | all |
| 10 | mfo (fram_strait) | $(11.5^{\circ}W, 81.3^{\circ}N$ $(10.5^{\circ}E, 79.6^{\circ}N)$ | full | OMIP | same | kg/s | month | 2 | all |
| 11 | mfo (gilbraltar_strait) | $(35.8^{\circ}N, 5.6^{\circ}W$ $(36^{\circ}N, 5.6^{\circ}W)$ | full | OMIP | new | kg/s | month | 2 | all |
| 12 | mfo (iceland_faroe_channel) | $(13.6^{\circ}W, 64.9^{\circ}N)$ $(7.4^{\circ}W, 62.2^{\circ}N)$ | full | OMIP | same | kg/s | month | 2 | all |
| 13 | mfo (indonesian_throughflow) | $(100^{\circ}E, 6^{\circ}S)$ $(140^{\circ}E, 6^{\circ}S)$ | full | OMIP | same | kg/s | month | 2 | all |
| 14 | mfo (mozambique_channel) | $(39^{\circ}E, 16^{\circ}S)$ $(45^{\circ}E, 18^{\circ}S)$ | full | OMIP | same | kg/s | month | 2 | all |
| 15 | mfo (pacific_equatorial_undercurrent) | $(155^{\circ}W, 3^{\circ}S)$ $(155^{\circ}W, 3^{\circ}N)$ | 0-350m | OMIP | same | kg/s | month | 2 | all |
| 16 | mfo (taiwan_and_luzon_straits) | $(121.8^{\circ}E, 18.3^{\circ}N)$ $(121.8^{\circ}E, 22.3^{\circ}N)$ | full | OMIP | same | kg/s | month | 2 | all |

**Table 4.** This table summarizes the sections for archiving the depth integrated mass transport time series from the ocean component in CMIP6 simulations. Each time series is identified by the CF standard name sea_water_transport_across_line. Additionally, each geographical region has an associated string-valued coordinate given by the name in this table. All time series should be saved as monthly means in units of kg/s. Positive and negative numbers refer to total northward/eastward and southward/westward transports, respectively. The column indicating the experiment for saving the diagnostics generally says "all", in which case we recommend the diagnostic be saved for CMIP6 experiments in which there is an ocean model component, including the DECK, historical simulations, FAFMIP, DAMIP, DCPP, ScenarioMIP, and C4MIP, as well as the ocean-sea ice OMIP simulations.

regions of the ocean. Offline diagnostics of these transports, using the archived velocity and/or the barotropic streamfunction, can be subject to uncertainty, especially for models with complex horizontal and vertical grids. It is thus more direct and accurate for each participating model group to diagnose transports online.

In Table 4, we offer a list of recommended transports for CMIP6, with further details for these sections given in Table 5. Each transport section has an associated string valued coordinate given by the name. We present references to observational estimates in Table 4, though note the large uncertainties in many locations.

We make the following recommendations regarding the integrated mass transports.

– In Table 4, we note the approximate geographical longitude and latitude coordinates of the straits and currents. Given considerations of model grid resolution and grid orientation, precise values for the coordinates may differ for any particular model. In general, we recommend computing the simulated transport where the strait is narrowest and shallowest in the model configuration, and where the model grid is closely aligned with the section.




| DETAILS OF THE MASS TRANSPORT SECTIONS | | | | |
|---|---|---|---|---|
| ITEM | TRANSECT | OBSERVED ESTIMATE (Sv) | MEASUREMENT METHOD | REFERENCES |
| 1 | barents_opening<br>Spitsbergen to Norway | 2.0 net northward<br>3.2 northward, 1.2 southward | hydrography from 1997-2007 | Smedsrud et al. (2010) |
| 2 | bering_strait<br>Alaska to Siberia | 0.8 northward from Roach et al. (1995)<br>0.7 in 2001; 1.1 in 2011<br>seasonal range 0.4 to 1.2 | moorings A3 and A2 since 2001 | Woodgate et al. (2012)<br>heat/freshwater transports available |
| 3 | caribbean_windward_passage<br>east Cuba to northwest Haiti | 3.8 southward (ships: range -9.4 to 0.3)<br>3.6 southward (current meters: range -15 to 5) | Oct2003–Feb2005 moored current meters,<br>hydrographic surveys and lowered ADCP | Smith et al. (2007) |
| 4 | davis_strait<br>transport through Davis Strait | $-1.6 \pm .5$ | moorings and gliders in years 2004-2010 | Curry et al. (2014) |
| 5 | denmark_strait<br>Greenland to Iceland | $-3.4 \pm 1.4$<br>all deployments; 4337 days;<br>no significant trend over 1996-2011 | two moored ADCP from 1996-2011:<br>DS1: $66^\circ 4.6' N$, $27^\circ 5.6' W$ at 650 m<br>DS2: $66^\circ 7.2' N$, $27^\circ 16.2' W$ at 570 m | Jochumsen et al. (2012) |
| 6 | drake_passage<br>South America to Antarctica Peninsula | $136.7 \pm 6.9$ | based on 15 repeat hydrography cruises<br>from 1993–2009 | Meredith et al. (2011) |
| 7 | english_channel<br>Britain to continental Europe | 0.1-0.2 | | Otto et al. (1990) |
| 8 | faroe_scotland_channel<br>Faroe Islands to Scotland | 4.1 northward, 3.2 southward<br>0.9 net detided $\pm 0.1$ interannual | ADCP on a ferry since March 2008 | Rossby and Flagg (2012) |
| 9 | florida_bahamas_strait<br>Florida Current between<br>Florida & Bahamas near $27^\circ N$ | $31.6 \pm 2.7$ | submarine-cable since 1982 | Baringer and Larsen (2001) |
| 10 | fram_strait<br>Spitsbergen to Greenland | $6.6 \pm 0.4$ | Fram Strait moorings from 1997–2010 | Beszczynska-Möller et al. (2012) |
| 11 | gilbraltar_strait<br>Morrocco to Spain | $0.78 \pm 0.47$ Atlantic inflow<br>$-0.67 \pm 0.26$ Med outflow | 5 ADCP moorings 1997–1998 | Tsimplis and Bryden (2000) |
| 12 | iceland_faroe_channel<br>Iceland to Faroe Islands | 6.0 northward; 1.4 southward<br>$4.6 \pm 0.25$ detided | ADCP on ferry since March 2008 | Rossby and Flagg (2012) |
| 13 | indonesian_throughflow<br>through the Indonesian Archipelogo | -13 | INSTANT program 2004–2006 | Gordon et al. (2009) |
| 14 | mozambique_channel<br>Madagascar to the African continent | $-16.7 \pm 8.9$ | current meter mooring from 2004–2008 | Ridderinkhof et al. (2010) |
| 15 | pacific_equatorial_undercurrent<br>zonal transport in eq. undercurrent | 24–36 | inverse method | Lukas and Firing (1984)<br>Sloyan et al. (2003) |
| 16 | taiwan_and_luzon_straits<br>Taiwan to Philippines island of Luzon | $-2.4 \pm 0.6$ | inverse method | Yaremchuk et al. (2009) |

**Table 5.** This table details the mass transport sections from Table 4, including observational-based measures.

– For many ocean model grids, the requested transports can be diagnosed by aligning the section along a model grid axis. In this case, it is straightforward to assign a positive sign to transports going in a pseudo-north or pseudo-east direction, and negative signs for the opposite direction. We use the term *pseudo* here as it refers to an orientation according to the model grid lines, which in general may not agree with geographical longitude and latitude lines. The sign convention

5    chosen for the recorded transport should be indicated in the metadata information for the transport field.

– Some models may have a strait artificially closed, due to inadequate grid resolution. In this case, a zero or missing transport should be recorded for this strait.

– The full column depth integrated mass transport vanishes for mesoscale closures based on Gent et al. (1995), and the submesocale closures based on Fox-Kemper et al. (2008). Hence, when computing a column integrated mass transport,

10    it only involves the resolved advective transport.

– For the equatorial undercurrent, we ask for the residual mean zonal transport from the surface to 350 m.



## 8 Boundary fluxes

The ocean is a forced-dissipative system, with forcing largely at its boundaries. To develop a mechanistic understanding of ocean simulations, it is critical to have a clear sampling of the many forcing fields. Some of the following fields can be found in other parts of the CMIP6 archive as part of the sea ice or atmosphere components. However, these fields are typically on grids distinct from the ocean model grid. Fluxes on grids distinct from the ocean make accurate budget analyses difficult to perform. Additionally, these other componets are absent in the OMIP ocean/sea-ice simulations. We thus follow the CMIP5 approach, in which we request that CMIP6 models archive the precise boundary fluxes used to force the ocean model.

### 8.1 General comments on boundary fluxes

We offer here some general comments regarding the boundary flux fields. Specifications for the requested fluxes are given in the subsequent subsections.

#### 8.1.1 Area normalization

All fluxes (water mass, salt mass, heat, momentum) are normalized according to the horizontal area of the ocean model grid cell. In some cases (e.g., rainfall), the flux computation requires integrating the rainfall over the ice-free sea (to get a mass per time of rainfall) and then dividing by the ocean grid cell area (to get mass per time per area). For these fluxes, according to the CF metadata conventions, the cell_methods attribute for the fields should read

– area: mean where ice_free_sea over all_area_types.

In other cases (e.g., melting sea ice) the flux computation requires integrating the sea ice melt over the sea ice covered portion of the ocean grid cell, and then dividing by the ocean grid cell area. For these fluxes, according to the CF metadata conventions, the cell_methods attribute for the fields should read

– area: mean where sea_ice over all_area_types.

#### 8.1.2 Diagnosing transports from fluxes

Multiplication of a boundary flux by the ocean model grid cell area allows for computing the transport of mass, salt, heat, or momentum passed to the ocean.[21] This property must be maintained whether the fluxes are archived on the native model grid, or mapped onto a spherical grid according to the recommendations of Appendix B.

#### 8.1.3 Sign convention for fluxes

Momentum fluxes have a sign so that a positive flux in a particular direction will increase the momentum of the liquid ocean. Likewise, fluxes of mass, salt, and heat are positive if they increase the ocean content of mass, salt, and heat. This convention

---

[21]The units of these transports are mass transport = kilogram per second; salt transport = kilogram per second; heat transport = joule per second (or watt); momentum transport = joule per second (or watt).



for scalars follows that from CMIP5, with one exception. Namely, the evaporation diagnostic in CMIP5 was positive for water leaving the ocean. The CMIP6 convention aims to make all scalar fluxes compatible with one another, so that the net mass flux entering the ocean is the sum of all the component mass fluxes. With this convention, evaporation has a negative sign, whereas condensation is positive.

### 8.1.4 Coupling beneath the surface ocean grid cell

Many climate models place boundary fluxes just at the ocean surface. However, more general couplings are being considered (e.g., penetrative shortwave heating; sea ice and ice shelf models that interact with more than the surface ocean cell). To allow for such generality, we ask that those fluxes that are three-dimensional be archived with their full three dimensional structure.

### 8.1.5 Flux adjustments

The term "flux correction" in Tables 6, 7, 8, and 9 refers to the imposition of a prescribed boundary flux that has at most a monthly variability (sometimes only an annual mean is used) (Sausen et al., 1988; Weaver and Hughes, 1996; Gordon et al., 2000). These modifications of the prognostic fluxes have no interannual variability.

We prefer the term "flux adjustments" since the use of modified fluxes, though aiming to reduce flux errors relative to observations, should not be presumed to be "correct". They are included in some models for the purpose of reducing model drift, with such drift a function of nearly all aspects of the particular model configuration. Flux adjustments are rather uncommon in CMIP6 due to model improvments during the recent 10-20 years, largely due to improved representation of poleward heat transport in both atmosphere and ocean models (see, for example, Section 8.4.2 of McAvaney et al., 2001). In such cases, the flux adjustment fields are zero or simply not archived

Note that for the CMIP6/FAFMIP experiment (Section 3.7 and Gregory et al. (2016)), the prescribed perturbation fields should be saved in the appropriate flux correction diagnostics for mass (or virtual salt), heat, and momentum.

### 8.1.6 Heat content of water crossing ocean boundaries

Seawater carries salt, heat, carbon, and other trace matter. Some tracers are transferred across the ocean surface as mass enters or leaves the ocean. This "advective" mass transfer across ocean boundaries must be accounted for in the budget for ocean tracer content (an extensive property). Salt is generally not transferred across the air-sea interface. However, it is transferred across the ice-sea boundary, since sea ice has a non-zero salinity. It can also be advected into the ocean from salty estuaries. The ocean heat budget is affected by the boundary transfer of radiative (shortwave and longwave), turbulent (sensible and latent), and advective heat fluxes. For example, rain falling on the ocean increases the ocean mass, and in so doing it increases the ocean heat content, even if it does not alter the local sea surface temperature.

There is an arbitrariness in ocean heat content associated with the arbitrary temperature scale. However, time changes in the heat content are not affected, as we show in Appendix E. We find it convenient to measure the heat content using the celsius temperature scale, since that is the scale used in ocean models. Consequently, the associated heat content (as a flux and relative



to $0°C$) is requested in Table 8. Models that artificially preclude water to cross the ocean boundary (e.g., rigid lid models, or models with a virtual tracer flux discussed in Section 8.1.7) have zero contributions to these heat fluxes, in which case there is no need to archive the zero fields.

For precipitation and evaporation, the heat flux associated with water transport across the ocean boundaries generally represents a global net heat loss for the ocean. The reason is that evaporation transfers water away from the ocean at a temperature typically higher than precipitation adds water. Delworth et al. (2006) estimate a global mean for this heat flux from a coupled climate model (see their Section 3), arriving at the value

$$Q_{\text{advective}} \approx -0.15 \text{ W m}^{-2}. \tag{66}$$

Likewise, Griffies et al. (2014) determine a heat flux for forced ocean-ice simulations to be (see their Appendix A.4)

$$Q_{\text{advective}} \approx -0.30 \text{ W m}^{-2}. \tag{67}$$

Locally, the heat flux can be far larger in magnitude.

In a steady state, where the total ocean mass and heat content are constant, this heat loss due to advective mass transfer is compensated by ocean mass and heat transport. This ocean transport is in turn balanced by atmospheric transport. However, most atmospheric models do not account for heat content of its moisture field, and so the moisture field carries no temperature information. Hence, the atmospheric model represents only the moisture mass transport, but not the moisture heat content transport. The global heat budget is therefore not closed for these coupled climate models due to a basic limitation of the modelled atmospheric thermodynamics.

### 8.1.7  Virtual salt fluxes

Some ocean models do not allow for the passage of water mass across the liquid ocean boundaries. Virtual salt fluxes are instead formulated to parameterize the effects of changes in salinity on the density field (Huang, 1993; Griffies et al., 2001; Yin et al., 2010b). The models that use virtual fluxes do not have a physically correct water cycle, as they have zero exchange of water between the ocean and other components of the climate system. Correspondingly, they do not have a physically correct salt budget, since the real ocean system has a trivial net flux of salt across the air-sea boundary, contrasting with the nontrivial virtual salt fluxes. Additionally, they are missing the Goldsborough-Stommel circulation (Goldsbrough (1933), Stommel (1957), and Huang and Schmitt (1993)). Since virtual salt flux simulations are still in use, we continue to request that those models archive the relevant salt fluxes as part of CMIP6.

### 8.2  Boundary mass fluxes

The water mass fluxes in Table 6 aim to present the analyst with sufficient information to perform a water mass budget on the liquid ocean, and to map regionally where water enters or leaves the ocean through various physical processes. The following presents some general comments.





| BOUNDARY MASS FLUXES | | | | | | | | | |
|---|---|---|---|---|---|---|---|---|---|
| ITEM | CMOR NAME | SPONSOR | CMIP5/CMIP6 | UNITS | TIME | SHAPE | GRID | PRIORITY | EXPT |
| 1 | pr | OMIP/FAFMIP | same | kg/(m² s) | month | XY | sphere (native) | 2 | all |
| 2 | prsn | OMIP/FAFMIP | same | kg/(m² s) | month | XY | sphere (native) | 2 | all |
| 3 | evs | OMIP/FAFMIP | swap sign convention | kg/(m² s) | month | XY | sphere (native) | 2 | all |
| 4 | friver | OMIP/FAFMIP | same | kg/(m² s) | month | XY | sphere (native) | 2 | all |
| 5 | ficeberg | OMIP/FAFMIP | same | kg/(m² s) | month | XYZ | sphere (native), z/p | 2 | all |
| 6 | fsitherm | OMIP/FAFMIP | same | kg/(m² s) | month | XY | sphere (native) | 2 | all |
| 7 | wfo | OMIP | same | kg/(m² s) | month | XY | sphere (native) | 1 | all |
| 8 | wfonocorr | OMIP | same | kg/(m² s) | month | XY | sphere (native) | 1 | all |
| 9 | wfcorr | OMIP | same | kg/(m² s) | month | XY | sphere (native) | 1 | all |

| CMOR NAME RELATED TO CF STANDARD NAME | | |
|---|---|---|
| ITEM | CMOR NAME | CF STANDARD NAME |
| 1 | pr | rainfall_flux |
| 2 | prsn | snowfall_flux |
| 3 | evs | water_evaporation_flux |
| 4 | friver | water_flux_into_sea_water_from_rivers |
| 5 | ficeberg | water_flux_into_sea_water_from_icebergs |
| 6 | fsitherm | water_flux_into_sea_water_due_to_sea_ice_thermodynamics |
| 7 | wfo | water_flux_into_sea_water |
| 8 | wfonocorr | water_flux_into_sea_water_without_flux_correction |
| 9 | wfcorr | water_flux_correction |

**Table 6.** This table details the boundary fluxes of water mass to be saved from the ocean model component in CMIP6 simulations. Positive fluxes are into the ocean. Hence, for example, evaporating water represents a negative mass flux (this sign convention is opposite that for CMIP5). The column indicating the experiment for saving the diagnostics generally says "all", in which case we recommend the diagnostic be saved for CMIP6 experiments in which there is an ocean model component, including the DECK, historical simulations, FAFMIP, DAMIP, DCPP, ScenarioMIP, and C4MIP, as well as the ocean-sea ice OMIP simulations. The Priority=1 diagnostics should be saved as decadal time means at decadal intervals for the piControl-spinup. Entries with grids denoted "sphere (native)" denote diagnostics where spherical output is strongly recommended to facilitate analysis, though where native output is accepted if spherical is unavailable. The bottom sub-table lists the relation between the CMOR name for a diagnostic and its CF standard name.

- Liquid runoff is defined as liquid water that enters the ocean from land, such as through rainwater in rivers, or snow and ice meltwater in rivers. It may also incorporate melt water from sea ice, icebergs, and ice shelves.

- An iceberg model exports a certain amount of calved land ice away from the coasts. It is thus important to record where the icebergs melt (horizontal position and depth), hence the suggestion to include iceberg melt in Table 6.

- Models that employ a virtual salt flux, and so do not allow for the transfer of water mass across the liquid ocean boundary, will have zero for each of these mass flux fields. In that case, the mass flux diagnostics should not be reported. Instead, see Section 8.3 for virtual salt fluxes diagnostics. Note that virtual salt flux models do not add mass to or remove mass from the ocean.





We now present specifications for the diagnosed fields. As discussed in Section 8.1.1, the fluxes, which may be defined only over a portion of each ocean grid cell, are normalized by the full area of each ocean grid cell. As a result, multiplying the ocean grid cell horizontal area times the flux will render the mass per time of water entering or leaving an ocean grid cell.

- pr = mass flux of liquid precipitation from the atmosphere entering the ice-free portion of an ocean grid cell.

- prsn = mass flux of frozen precipitation (i.e., snow) from the atmosphere entering the ice-free portion of an ocean grid cell.

- evs = rate at which water crosses the air-sea interface due to evaporation and condensation, passing through the ice-free portion of an ocean grid cell. This flux is positive for water entering the liquid ocean through condensation, and negative when leaving the ocean through evaporation. This sign convention is opposite that used for CMIP5.

- friver = mass of liquid water runoff entering the ocean from land-surface boundaries.

- ficeberg = solid mass that enters the ocean from land-ocean boundaries will eventually melt in the ocean. This melt may occur just at the ocean-land boundary, be distributed seawards by a spreading scheme, or participate in the transport via icebergs. It may also be distributed with depth.

- fsitherm = contribution to liquid ocean mass due to the melt (positive mass flux) or freezing (negative mass flux) of sea ice.

- wfo = net flux of liquid water entering the liquid ocean.

- wfonocorr = mass flux due to physical processes absent the flux corrections. For models without flux corrections, wfonocorr = wfo.

- wfcorr = mass flux due to flux corrections. It is zero for models with no prescribed flux corrections/adjustments. However, for the CMIP6/FAFMIP experiment (Section 3.7), the prescribed perturbation water flux should be saved in the wfcorr diagnostic for FAFMIP models that make use of a real water flux.

The following equalities are satisfied by the requested water flux fields

$$\text{wfo} = \text{wfonocorr} + \text{wfcorr} \tag{68a}$$

$$\text{wfonocorr} = \text{pr} + \text{prsn} + \text{evs} + \text{friver} + \text{ficeberg} + \text{fsitherm}. \tag{68b}$$

## 8.3 Boundary salt fluxes

The salt fluxes in Table 7 aim to present the analyst with sufficient information to perform a salt budget on the liquid ocean, and to map regionally where salt enters or leaves the ocean through various physical processes. The following presents some details about the fields. Note that for models using real water fluxes, the virtual salt flux fields are all zero, so that there is no reason to submit these diagnostics.



| BOUNDARY SALT FLUXES | | | | | | | | | |
|---|---|---|---|---|---|---|---|---|---|
| ITEM | CMOR NAME | SPONSOR | CMIP5/CMIP6 | UNITS | TIME | SHAPE | GRID | PRIORITY | EXPT |
| 1 | vsfpr | OMIP/FAFMIP | same | kg/(m$^2$ s) | month | XY | sphere (native) | 2 | all |
| 2 | vsfevap | OMIP/FAFMIP | same | kg/(m$^2$ s) | month | XY | sphere (native) | 2 | all |
| 3 | vsfriver | OMIP/FAFMIP | same | kg/(m$^2$ s) | month | XY | sphere (native) | 2 | all |
| 4 | vsfsit | OMIP/FAFMIP | same | kg/(m$^2$ s) | month | XY | sphere (native) | 2 | all |
| 5 | vsf | OMIP/FAFMIP | same | kg/(m$^2$ s) | month | XY | sphere (native) | 2 | all |
| 6 | vsfcorr | OMIP/FAFMIP | same | kg/(m$^2$ s) | month | XY | sphere (native) | 2 | all |
| 7 | sfdsi | OMIP | same | kg/(m$^2$ s) | month | XY | sphere (native) | 1 | all |
| 8 | sfriver | OMIP | same | kg/(m$^2$ s) | month | XY | sphere (native) | 1 | all |

| CMOR NAME RELATED TO CF STANDARD NAME | | |
|---|---|---|
| ITEM | CMOR NAME | CF STANDARD NAME |
| 1 | vsfpr | virtual_salt_flux_into_sea_water_due_to_rainfall |
| 2 | vsfevap | virtual_salt_flux_into_sea_water_due_to_evaporation |
| 3 | vsfriver | virtual_salt_flux_into_sea_water_from_rivers |
| 4 | vsfsit | virtual_salt_flux_into_sea_water_due_to_sea_ice_thermodynamics |
| 5 | vsf | virtual_salt_flux_into_sea_water |
| 6 | vsfcorr | virtual_salt_flux_correction |
| 7 | sfdsi | downward_sea_ice_basal_salt_flux |
| 8 | sfriver | salt_flux_into_sea_water_from_rivers |

**Table 7.** This table provides a summary of the boundary fluxes of salt mass that should be saved from the ocean model component in CMIP6 simulations. Positive fluxes are into the ocean. The column indicating the experiment for saving the diagnostics generally says "all", in which case we recommend the diagnostic be saved for CMIP6 experiments in which there is an ocean model component, including the DECK, historical simulations, FAFMIP, DAMIP, DCPP, ScenarioMIP, and C4MIP, as well as the ocean-sea ice OMIP simulations. The Priority=1 diagnostics should be saved as decadal time means at decadal intervals for the piControl-spinup. Entries with grids denoted "sphere (native)" denote diagnostics where spherical output is strongly recommended to facilitate analysis, though where native output is accepted if spherical is unavailable. The lower sub-table relates the CMOR name to its CF standard name.

- vsfpr = virtual salt flux associated with liquid and solid precipitation.

- vsfevap = virtual salt flux associated with evaporation of water.

- vsfriver = virtual salt flux associated with liquid and solid runoff from land processes.

- vsfsit = virtual salt flux associated with melting or freezing of sea ice.

5  - vsf = total virtual salt flux entering the ocean. It is the sum of all of the above virtual salt fluxes, including the salt flux correction.

- vsfcorr = virtual salt flux arising from a salt flux correction. It is zero for models with no prescribed virtual salt flux correction/adjustment. For the CMIP6/FAFMIP experiment (Section 3.7) and for such models making use of a virtual salt flux, then the virtual salt flux corresponding to the prescribed perturbation water flux should be saved in the vsfcorr
10     diagnostic.





- sfdsi = salt transport from sea-ice to the ocean. The field arises since sea ice has a nonzero salinity, so it exchanges salt with the liquid ocean upon melting and freezing. This field is distinct from the virtual salt flux into sea water due to sea ice thermodynamics.

- sfriver = salt content of rivers. This field is typically zero, though some river models carry a non-zero salt concentration.

## 8.4 Boundary heat fluxes

- hfgeou = upward geothermal heat flux at sea floor

- hfrainds = heat content of liquid (rain), liquid condensate (precipitating fog), and solid (snow) precipitation (relative to $0°C$)

- hfevapds = heat content of water leaving the ocean (relative to $0°C$) due to evaporation or sea ice formation

- hfrunoffds = heat content of runoff in a liquid form (rivers) and solid form (calving land ice and icebergs) (relative to $0°C$)

- hfsifrazil = heat flux due to frazil ice formation

- hfsnthermds = latent heat flux due to snow melting

- hfibthermds = latent heat flux due to iceberg melting

- rlntds = surface net downward longwave flux

- hfls = surface downward latent heat flux (from evaporating vapor and from melting solids)

- hfss = surface downward sensible heat flux (including both air-sea and ice-sea sensible heating)

- rsntds = net downward shortwave flux at sea water surface

- rsdo = downwelling shortwave flux in sea water[22]

- hfcorr = heat flux correction

- hfds = net downward heat flux at sea surface (excluding any flux correction/adjustment)

These fluxes are summarized in Table 8. They present the analyst with sufficient information to perform a heat budget on the liquid ocean, and to map regionally where heat enters or leaves the ocean through various physical processes. We provide further details on the specification in the following.

---

[22]Note there was a mistake in the CMIP5 Xcel spread-sheet, with rsdo incorrectly listed as rsds. In fact, rsds is an atmospheric field.




| \multicolumn{9}{c}{BOUNDARY HEAT FLUXES} |
|---|---|---|---|---|---|---|---|---|
| ITEM | CMOR NAME | SPONSOR | CMIP5/CMIP6 | UNITS | TIME | SHAPE | GRID | PRIORITY | EXPT |
| 1 | hfgeou | OMIP/FAFMIP | same | W/m$^2$ | month | XY | sphere | 1 | all |
| 2 | hfrainds | OMIP/FAFMIP | same | W/m$^2$ | month | XY | sphere (native) | 2 | all |
| 3 | hfevapds | OMIP/FAFMIP | same | W/m$^2$ | month | XY | sphere (native) | 2 | all |
| 4 | hfrunoffds | OMIP/FAFMIP | same | W/m$^2$ | month | XYZ | sphere (native), z/p | 2 | all |
| 5 | hfsnthermds | OMIP/FAFMIP | same | W/m$^2$ | month | XYZ | sphere (native),z/p | 2 | all |
| 6 | hfsifrazil | OMIP/FAFMIP | same | W/m$^2$ | month | XYZ | sphere (native),z/p | 2 | all |
| 7 | hfibthermds | OMIP/FAFMIP | same | W/m$^2$ | month | XYZ | sphere (native), z/p | 2 | all |
| 8 | rlntds | OMIP/FAFMIP | CMIP5 called this rlds in Omon | W/m$^2$ | month | XY | sphere (native) | 2 | all |
| 9 | hfls | OMIP/FAFMIP | same | W/m$^2$ | month | XY | sphere (native) | 2 | all |
| 10 | hfss | OMIP/FAFMIP | same | W/m$^2$ | month | XY | sphere (native) | 2 | all |
| 11 | rsntds | OMIP/FAFMIP | same | W/m$^2$ | month | XY | sphere (native) | 2 | all |
| 12 | rsdo | OMIP/FAFMIP | same | W/m$^2$ | month | XYZ | sphere (native), z/p | 2 | all |
| 13 | hfcorr | OMIP | same | W/m$^2$ | month | XY | sphere (native) | 1 | all |
| 14 | hfds | OMIP | same | W/m$^2$ | month | XY | native (sphere) | 1 | all |

| \multicolumn{3}{c}{CMOR NAME RELATED TO CF STANDARD NAME} |
|---|---|---|
| ITEM | CMOR NAME | CF STANDARD NAME |
| 1 | hfgeou | upward_geothermal_heat_flux_at_sea_floor |
| 2 | hfrainds | temperature_flux_due_to_rainfall_expressed_as_heat_flux_into_sea_water |
| 3 | hfevapds | temperature_flux_due_to_evaporation_expressed_as_heat_flux_out_of_sea_water |
| 4 | hfrunoffds | temperature_flux_due_to_runoff_expressed_as_heat_flux_into_sea_water |
| 5 | hfsnthermds | heat_flux_into_sea_water_due_to_snow_thermodynamics |
| 6 | hfsifrazil | heat_flux_into_sea_water_due_to_frazil_ice_formation |
| 7 | hfibthermds | heat_flux_into_sea_water_due_to_iceberg_thermodynamics |
| 8 | rlntds | surface_net_downward_longwave_flux |
| 9 | hfls | surface_downward_latent_heat_flux |
| 10 | hfss | surface_downward_sensible_heat_flux |
| 11 | rsntds | net_downward_shortwave_flux_at_sea_water_surface |
| 12 | rsdo | downwelling_shortwave_flux_in_sea_water |
| 13 | hfcorr | heat_flux_correction |
| 14 | hfds | surface_downward_heat_flux_in_sea_water |

**Table 8.** This table provides a summary of the boundary fluxes of heat to be saved from the ocean component in CMIP6 simulations. Positive fluxes are into the ocean. The column indicating the experiment for saving the diagnostics generally says "all", in which case we recommend the diagnostic be saved for CMIP6 experiments in which there is an ocean model component, including the DECK, historical simulations, FAFMIP, DAMIP, DCPP, ScenarioMIP, and C4MIP, as well as the ocean-sea ice OMIP simulations. The Priority=1 diagnostics should be saved as decadal time means at decadal intervals for the piControl-spinup. For the geothermal heating, most models use a static geothermal heating, in which case only one time step need be archived. If time dependent, then monthly fields are requested. Note that many climate models place boundary fluxes at the ocean surface. However, more general couplings are being considered (e.g., a sea ice or ice shelf model that interacts with more than the surface ocean cell). To allow for such generality, we note that many of the fluxes can be three-dimensional. Note that the field "rsdo" was mistakenly included in the CMIP5 diagnostic excel spreadsheet as "rsds". Entries with grids denoted "sphere (native)" denote diagnostics where spherical output is strongly recommended to facilitate analysis, though where native output is accepted if spherical is unavailable. The lower sub-table relates the CMOR name to its CF standard name.





## 8.5 Relations satisfied by the heat fluxes

We summarize here some relations satisfied by the diagnosed heat fluxes.

- NET HEAT FLUX

  The net heat flux crossing the bottom and surface boundaries of the liquid ocean is given by

$$\text{net heat} = \text{hfgeou} + \text{hfds} + \text{hfcorr}. \tag{69}$$

- SURFACE HEAT FLUX WITHOUT HFCORR

  The net heat flux crossing the surface boundary of the liquid ocean, without flux adjustments, is given by

$$
\begin{aligned}
\text{hfds} = {} & \text{hfrainds} + \text{hfevapds} + \text{hfrunoffds} \\
& + \text{rlntds} + \text{hfls} + \text{hfss} + \text{rsntds} \\
& + \text{hfsifrazil}.
\end{aligned} \tag{70}
$$

  This is a critical identity to verify prior to making use of the various surface heat flux components for analysis.

- LATENT HEAT FLUX

  The net latent heat flux, hfls, contains contributions from the latent heat loss due to evaporating water, melting snow (hfsnthermds), melting icebergs (hfibthermds), and the melt/formation of sea ice.

- HEAT CONTENT OF PRECIPITATION

  The net heat content of precipitation, hfrainds, contains contributions from the mass of liquid precipitation (rain), solid precipitation (snow), sea ice melt, and condensed fog that falls into the ocean.

- HEAT CONTENT OF RUNOFF

  The net heat content of runoff, hfrunoffds, contains contributions from the mass of liquid runoff and solid runoff. The solid runoff may be exported from the coast via a spreading scheme and/or an iceberg model.

## 8.6 Specifications for the boundary heat fluxes

We here provide further specification for these diagnostics.

### 8.6.1 hfgeou

- hfgeou = geothermal heat flux

The geothermal heat flux is typically a static field. Models that use a time dependence should archive the monthly heat flux.



### 8.6.2 hfrainds

- hfrainds = heat content of liquid and solid precipitation with respect to $0°C$

For many climate models, this diagnostic is estimated using ocean properties

$$\text{hfrainds}(\text{W/m}^2) = c_p^o \left( Q_{\text{rain}} T_{\text{rain}} + Q_{\text{snow}} T_{\text{snow}} \right), \tag{71}$$

where $Q_{\text{rain}}$ and $Q_{\text{snow}}$ are the rain and snowfall mass fluxes, in $\text{kg/(m}^2\,\text{sec)}$, $T_{\text{rain}}$ is the temperature of rainfall in degrees celsius, and $T_{\text{snow}}$ is the temperature of snowfall in degrees celsius. Most climate models choose the rainfall and snowfall temperature equal to the ocean sea surface temperature. An assumption is needed since atmospheric models do not generally carry the temperature of their moisture field, and so do not provide heat content for the rain and snow (see discussion in Section 8.1.6 and Appendix E).

The field hfrainds is zero for ocean models employing a virtual tracer flux, in which there is no mass or volume transport of water across the ocean surface.

### 8.6.3 hfevapds

- hfevapds = heat content of water leaving the ocean, with respect to $0°C$

This diagnostic measures the heat content of water carried away from the liquid ocean via evaporation or sea ice formation. This heat is distinct from latent heat flux, which arises from a phase change. Instead, this heat content can be computed with respect to $0°C$ just as the diagnostic hfrainds. Here, if we make use of ocean fields, then

$$\text{hfevapds} = c_p^o \left( Q_{\text{evap}} T_{\text{evap}} + Q_{\text{ice form}} T_{\text{ice form}} \right), \tag{72}$$

where $Q_{\text{evap}}$ and $Q_{\text{ice form}}$ are the evaporative and ice formation mass fluxes, in $\text{kg/(m}^2\,\text{sec)}$, $T_{\text{evap}}$ is the temperature of evaporating water in degrees celsius, and $T_{\text{ice form}}$ is the temperature of water forming sea ice in degrees celsius. These temperatures are typically approximated by the sea surface temperature from the ocean model.

The field hfevapds is zero for models employing a virtual salt flux, in which there is no mass transport of water across the ocean surface.

### 8.6.4 hfrunoffds

- hfrunoffds = heat content of liquid runoff with respect to $0°C$

This diagnostic measures the heat content of liquid and solid runoff that enters the liquid ocean, with respect to $0°C$. This heat is typically estimated through use of ocean properties via

$$\text{hfrunoffds} = c_p^o \left( Q_{\text{runoff}} T_{\text{runoff}} + Q_{\text{icebergs}} T_{\text{icebergs}} \right), \tag{73}$$





where $Q_{\text{runoff}}$ and $Q_{\text{icebers}}$ are the liquid and solid runoff mass fluxes, in $\text{kg}/(\text{m}^2\,\text{sec})$, $T_{\text{runoff}}$ is the temperature of liquid runoff in degrees celsius, and $T_{\text{icebergs}}$ is the temperature of solid runoff in degrees celsius.

The field hfrunoffds is zero for models employing a virtual tracer flux, in which there is no mass transport of water across the ocean surface.

### 8.6.5  hfsifrazil

- hfsifrazil = frazil heat flux

As the temperature of seawater cools to the freezing point, sea ice is formed, initially through the production of frazil. Operationally in an ocean model, liquid water can be supercooled at any particular time step through surface fluxes and transport. An adjustment process heats the liquid water back to the freezing point, with this positive frazil heat flux extracted from the ice model as frazil sea ice is formed. This term is necessary to close the heat budget of the liquid ocean.

### 8.6.6  hfsnthermds

- hfsnthermds = latent heat of fusion required to melt snow

Snow entering the liquid ocean melts upon obtaining the latent heat of fusion from the ocean. This cooling of the liquid ocean is what is archived in hfsnthermds. Note that hfibthermds is included as part of the net latent heat flux diagnostic hfls.

### 8.6.7  hfibthermds

- hfibthermds = latent heat of fusion required to melt calving land ice and/or icebergs

Icebergs transport calved land ice from the land into the ocean. A rudimentary "iceberg" model may simply be the insertion of calving land ice/snow into the ocean, with an associated mass and heat transport (heat content of ice plus heat of fusion required to melt the ice). More realistic iceberg models are now becoming more common (Jongma et al., 2009; Martin and Adcroft, 2010; Marsh et al., 2015). Melting of the icebergs into the liquid ocean is associated with a transfer of the latent heat of fusion from liquid ocean, and so represents a cooling of the liquid ocean in regions where the icebergs melt. It is this heat flux that is archived in hfibthermds.[23] Note that hfibthermds is included as part of the net latent heat flux diagnostic hfls.

### 8.6.8  rlntds

- rlntds = downward flux of longwave radiation

This diagnostic measures the net downward flux of longwave radiation that enters the liquid ocean. Negative values cool the ocean.

---

[23]In testing the NEMO-ICB iceberg model, Marsh et al. (2015) considered icebergs wtih zero heat capacity. However, heat conservation in the coupled climate system requires that the latent heat of fusion used to create the ice on land must be given up by the liquid ocean as the icebergs melt. Consequently, icebergs with zero heat capacity should *not* be used in a coupled climate simulation for CMIP6.





### 8.6.9 hfls

- hfls = latent heat flux

This diagnostic provides the net latent heat flux, including contributions from the latent heat loss due to evaporating water, melting snow (hfsnthermds), melting icebergs (hfibthermds), and the melt/formation of sea ice. Negative values cool the ocean, as occurs when liquid water evaporates or solid ice or snow melts.

### 8.6.10 hfss

- hfss = sensible heat flux

This diagnostic measures the net downward flux of sensible heat acting on the liquid ocean. Positive values warm the ocean and negative values cool. The physical processes contributing to this heat flux include air-sea and ice-sea interactions.

### 8.6.11 rsntds

- rsntds = shortwave radiative flux at top of ocean

This diagnostic measures the net downward flux of shortwave radiation at the top of the ocean surface. Positive values warm the ocean.

### 8.6.12 rsdo and rsdoabsorb

- rsdo = downwelling shortwave flux in sea water

- rsdoabsorb = net rate of absorption of shortwave energy in ocean layer

Shortwave radiation penetrates into the ocean column and so heats the ocean interior (see Figure 8). The field rsdo measures the shortwave heat flux at the bottom of a tracer grid cell face. The field rsdoabsorb is the vertical convergence of rsdo, and is requested for the heat budget in Table 10.

### Theoretical considerations

The parameterization of oceanic absorption of downward solar radiation is generally written as

$$\text{rsdo} = I^{\text{penetrate}}(x,y,z) = I^{\text{down}}(x,y)\,\mathcal{F}(z), \tag{74}$$

where $I^{\text{down}}$, in units of $\mathrm{W\,m^{-2}}$, is the downwelling shortwave radiative heat per unit area incident at the ocean surface, and $\mathcal{F}(z)$ is a dimensionless attenuation function that depends on seawater optical properties.

Some models equate the downwelling shortwave, $I^{\text{down}}$, to the net incoming shortwave radiation, $I^{\text{net}} = \text{rsntds}$. However, some models decompose the net shortwave into downwelling and non-downwelling contributions

$$I^{\text{net}} = I^{\text{down}} + I^{\text{non-down}}. \tag{75}$$





Again, $I^{\text{down}}$ is the downwelling radiation that participates in shortwave penetrative radiation according to equation (74). In contrast, the non-downwelling portion, $I^{\text{non-down}}$, is deposited directly into the skin layer of the upper ocean (generally assumed to be in the upper ocean model grid cell), so that $I^{\text{non-down}}$ does not participate in the downwelling penetrative radiation.

Penetrative shortwave fluxes affect the heat budget according to

$$\text{rsdoabsorb}_k = c_p^o \left( \frac{\partial (\Theta \, \rho \, \mathrm{d}z)}{\partial t} \right)_k \tag{76a}$$

$$= (I_{k-1}^{\text{penetrate}} - I_k^{\text{penetrate}}) \tag{76b}$$

$$= \text{rsdo}_{k-1} - \text{rsdo}_k, \tag{76c}$$

where $k$ is the discrete vertical index increasing downward, and $\text{rsdo}_k = I_k^{\text{penetrate}}$ is the penetrative shortwave heat flux at the bottom of tracer cell $k$. The penetrative shortwave flux entering the top of a cell, $I_{k-1}^{\text{penetrate}}$ is larger than the flux leaving the cell bottom, $I_k^{\text{penetrate}}$, so that shortwave radiation is deposited within a tracer cell. The net shortwave absorbed by the tracer cell, rsdoabsorb, is asked for in the heat budget Table 10. Assuming a zero penetrative heat flux through the ocean bottom, $\text{rsdo}_{kmax} = 0$, we can infer the flux field $\text{rsdo}_k$ by integrating the convergence field $\text{rsdoabsorb}_k$ upwards.

We note here a peculiarity of penetrative shortwave heating implemented in some models. Namely, the net downwelling radiation at the top of the ocean, $I^{\text{down}}$, is sometimes included as part of the surface boundary condition portion of the code, rather than as part of the penetrative shortwave code. If this convention is followed, then rsdoabsorb will be negative (i.e., cooling) in the $k = 1$ grid cell, since in this case the $I^{\text{down}}$ contribution is part of rsntds. When performing a heat budget, care should be exercised to not double-count, or conversely to not account for, the contribution of $I^{\text{down}}$ to the $k = 1$ cell. We discuss this point further in Section 9.4.9.

### 8.6.13 hfcorr

- hfcorr = heat flux correction

This diagnostic records the heat flux correction acting at the liquid ocean surface. This field is zero for nearly all CMIP6 models. However, for the CMIP6/FAFMIP experiment (Section 3.7 and Gregory et al. (2016)), the prescribed FAFMIP perturbation heat should be saved in the hfcorr diagnostic (see Table 8).

### 8.6.14 hfds

- hfds = net surface downward heat flux at sea surface, excluding any flux correction/adjustment

This diagnostic measures the net heat flux passing across the ocean upper surface due to radiative, turbulent, latent, frazil, and heat content fluxes. It is related to the other diagnostic heat fluxes through equation (70). It does not include any heat flux correction/adjustment.

### 8.7 Boundary fluxes of momentum

- tauuo = downward surface x-stress



- tauvo = downward surface y-stress

- tauucorr = downward surface x-stress correction/adjustment

- tauvcorr = downward surface y-stress correction/adjustment

These fluxes are summarized in Table 9. They quantify the net momentum imparted to the liquid ocean surface, with positive
values accelerating the liquid ocean in the noted direction. These stresses arise from the overlying atmosphere, sea ice, icebergs,
ice shelf, etc. For models that do not apply a stress flux correction/adjustment, they will not report any diagnostics for either
tauucorr or tauvcorr. However, for the CMIP6/FAFMIP experiment (Section 3.7 and Gregory et al. (2016)), the prescribed
perturbation wind stress should be saved in the tauucorr and tauvcorr diagnostics.

| BOUNDARY MOMENTUM FLUXES | | | | | | | | | |
|---|---|---|---|---|---|---|---|---|---|
| ITEM | CMOR NAME | SPONSOR | CMIP5/CMIP6 | UNITS | TIME | SHAPE | GRID | PRIORITY | EXPT |
| 1 | tauuo | OMIP | same | N/m$^2$ | month | XY | sphere (native) | 1 | all |
| 2 | tauvo | OMIP | same | N/m$^2$ | month | XY | sphere (native) | 1 | all |
| 3 | tauucorr | OMIP | same | N/m$^2$ | month | XY | sphere (native) | 1 | all |
| 4 | tauvcorr | OMIP | same | N/m$^2$ | month | XY | sphere (native) | 1 | all |

| CMOR NAME RELATED TO CF STANDARD NAME | | |
|---|---|---|
| ITEM | CMOR NAME | CF STANDARD NAME |
| 1 | tauuo | surface_downward_x_stress |
| 2 | tauvo | surface_downward_y_stress |
| 3 | tauucorr | surface_downward_x_stress_correction |
| 4 | tauvcorr | surface_downward_y_stress_correction |

**Table 9.** This table presents the net surface stress applied at the liquid ocean surface due to air-sea plus ice-sea interactions. Positive fluxes
accelerate the ocean in the given direction. The column indicating the experiment for saving the diagnostics generally says "all", in which
case we recommend the diagnostic be saved for CMIP6 experiments in which there is an ocean model component, including the DECK,
historical simulations, FAFMIP, DAMIP, DCPP, ScenarioMIP, and C4MIP, as well as the ocean-sea ice OMIP simulations. We recommend
mapping each momentum flux component to an A-grid on the sphere, though native grid values are acceptable. The units N m$^{-2}$ are identical
to Pa. The lower sub-table relates the CMOR name to the CF standard name.





## 9 Budget terms for heat and salt

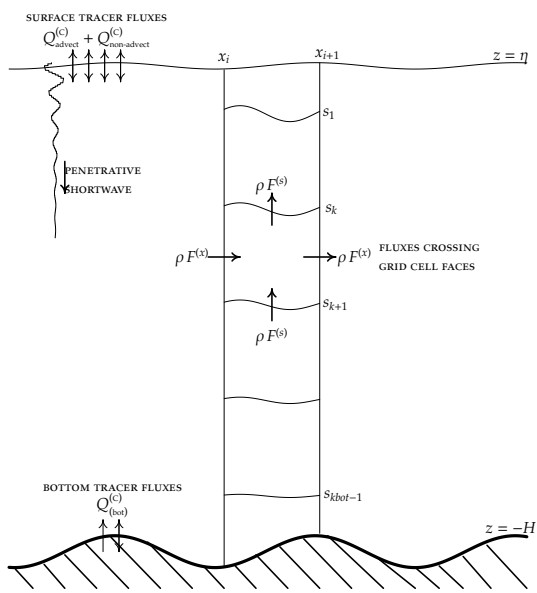

**Figure 1.** A longitudinal-vertical slice of ocean fluid from the surface at $z = \eta(x, y, t)$ to bottom at $z = -H(x, y)$, along with a representative column of discrete grid cells (a latitudinal-vertical slice is analogous). Ocean models used for large-scale climate studies assume the horizontal boundaries of a grid cell at $x_i$ and $x_{i+1}$ are static, meaning that the horizontal cross-sectional area is time independent. In contrast, the vertical extent, defined by surfaces of constant generalized vertical coordinate $s_k$ and $s_{k+1}$, are generally time dependent (e.g., $z^*$ surfaces, pressure surfaces, isopycnal surfaces, sigma surfaces, etc.). A general tracer flux $\rho \boldsymbol{F}$ (e.g., advective or subgrid scale flux) is decomposed into horizontal and dia-surface components, with the convergence of these fluxes onto a grid cell determining the evolution of tracer content within the cell. Amongst the fluxes crossing the ocean surface, the shortwave flux penetrates into the ocean column as a function of the optical properties of seawater (e.g., Manizza et al., 2005). This figure is based on Figure 1 of Griffies and Treguier (2013).

Heat and salt budget terms facilitate mechanistic information about model behaviour. Studies such as Gregory (2000), Palter et al. (2014), Griffies et al. (2015), Exarchou et al. (2014) and Kuhlbrodt et al. (2015) illustrate the value for physical interpretation provided by budget terms. The bulk of the science resulting from these budgets has thus far come from use of the annual mean fields, hence we request annual means, with these fields being Priority=1 for FAFMIP (Gregory et al., 2016) and Priority=3 for other experiments. Additionally, FAFMIP requests monthly mean fields at priority=2 in order to study seasonal variations of the budget terms.

### 9.1 Tracer budgets for a grid cell

To help ensure proper archival and use of heat and salt budgets terms, we provide here a tutorial for tracer budgets in a discrete ocean model, with this disussion following the finite volume formulation in Appendix F. This formulation leads to the following semi-discrete equations used as the basis for an ocean model tracer budget in surface, interior, and bottom grid cells,



| | | | | 3D HEAT AND SALT BUDGET TERMS | | | | | |
|---|---|---|---|---|---|---|---|---|---|
| ITEM | CMOR NAME | SPONSOR | CMIP5/CMIP6 | UNITS | TIME | SHAPE | GRID | PRIORITY | EXPT |
| 1 | opottempmint | FAFMIP | new | $(\text{kg m}^{-2})\,^\circ C$ | annual | XY | sphere (native) | 3 | all |
| 2 | ocontempmint | FAFMIP | new | $(\text{kg m}^{-2})\,^\circ C$ | annual | XY | sphere (native) | 3 | all |
| 3 | somint | FAFMIP | new | $(\text{kg m}^{-2}) * (1e-3)$ | annual | XY | sphere (native) | 3 | all |
| 4 | rsdoabsorb | FAFMIP | new | $\text{W m}^{-2}$ | annual | XYZ | sphere (native), z/p | 3 | all |
| 5 | opottemptend | FAFMIP | new | $\text{W m}^{-2}$ | annual | XYZ | sphere (native), z/p | 3 | all |
| 6 | opottemprmadvect | FAFMIP | new | $\text{W m}^{-2}$ | annual | XYZ | sphere (native), z/p | 3 | all |
| 7 | opottemppadvect | FAFMIP | new | $\text{W m}^{-2}$ | annual | XYZ | sphere (native), z/p | 3 | all |
| 8 | opottemppsmadvect | FAFMIP | new | $\text{W m}^{-2}$ | annual | XYZ | sphere (native), z/p | 3 | all |
| 9 | opottemppmdiff | FAFMIP | new | $\text{W m}^{-2}$ | annual | XYZ | sphere (native), z/p | 3 | all |
| 10 | opottempdiff | FAFMIP | new | $\text{W m}^{-2}$ | annual | XYZ | sphere (native), z/p | 3 | all |
| 11 | ocontemptend | FAFMIP | new | $\text{W m}^{-2}$ | annual | XYZ | sphere (native), z/p | 3 | all |
| 12 | ocontemprmadvect | FAFMIP | new | $\text{W m}^{-2}$ | annual | XYZ | sphere (native), z/p | 3 | all |
| 13 | ocontemppadvect | FAFMIP | new | $\text{W m}^{-2}$ | annual | XYZ | sphere (native), z/p | 3 | all |
| 14 | ocontemppsmadvect | FAFMIP | new | $\text{W m}^{-2}$ | annual | XYZ | sphere (native), z/p | 3 | all |
| 15 | ocontemppmdiff | FAFMIP | new | $\text{W m}^{-2}$ | annual | XYZ | sphere (native), z/p | 3 | all |
| 16 | ocontempdiff | FAFMIP | new | $\text{W m}^{-2}$ | annual | XYZ | sphere (native), z/p | 3 | all |
| 17 | osalttend | FAFMIP | new | $\text{kg m}^{-2}\,\text{s}^{-1}$ | annual | XYZ | sphere (native), z/p | 3 | all |
| 18 | osaltrmadvect | FAFMIP | new | $\text{kg m}^{-2}\,\text{s}^{-1}$ | annual | XYZ | sphere (native), z/p | 3 | all |
| 19 | osaltpadvect | FAFMIP | new | $\text{kg m}^{-2}\,\text{s}^{-1}$ | annual | XYZ | sphere (native), z/p | 3 | all |
| 20 | osaltpsmadvect | FAFMIP | new | $\text{kg m}^{-2}\,\text{s}^{-1}$ | annual | XYZ | sphere (native), z/p | 3 | all |
| 21 | osaltpmdiff | FAFMIP | new | $\text{kg m}^{-2}\,\text{s}^{-1}$ | annual | XYZ | sphere (native), z/p | 3 | all |
| 22 | osaltdiff | FAFMIP | new | $\text{kg m}^{-2}\,\text{s}^{-1}$ | annual | XYZ | sphere (native), z/p | 3 | all |

| | CMOR NAME RELATED TO CF STANDARD NAME | |
|---|---|---|
| ITEM | CMOR NAME | CF STANDARD NAME |
| 1 | opottempmint | integral_wrt_depth_of_product_of_sea_water_density_and_potential_temperature |
| 2 | ocontempmint | integral_wrt_depth_of_product_of_sea_water_density_and_conservative_temperature |
| 3 | somint | integral_wrt_depth_of_product_of_sea_water_density_and_salinity |
| 4 | rsdoabsorb | net_rate_of_absorption_of_shortwave_energy_in_ocean_layer |
| 5 | opottemptend | tendency_of_sea_water_potential_temperature_expressed_as_heat_content |
| 6 | opottemprmadvect | tendency_of_sea_water_potential_temperature_expressed_as_heat_content_due_to_residual_mean_advection |
| 7 | opottemppadvect | tendency_of_sea_water_potential_temperature_expressed_as_heat_content_due_to_parameterized_eddy_advection |
| 8 | opottemppsmadvect | tendency_of_sea_water_potential_temperature_expressed_as_heat_content_due_to_parameterized_submesoscale_advection |
| 9 | opottemppmdiff | tendency_of_sea_water_potential_temperature_expressed_as_heat_content_due_to_parameterized_mesoscale_diffusion |
| 10 | opottempdiff | tendency_of_sea_water_potential_temperature_expressed_as_heat_content_due_to_parameterized_dianeutral_mixing |
| 11 | ocontemptend | tendency_of_sea_water_conservative_temperature_expressed_as_heat_content |
| 12 | ocontemprmadvect | tendency_of_sea_water_conservative_temperature_expressed_as_heat_content_due_to_residual_mean_advection |
| 13 | ocontemppadvect | tendency_of_sea_water_conservative_temperature_expressed_as_heat_content_due_to_parameterized_eddy_advection |
| 14 | ocontemppsmadvect | tendency_of_sea_water_conservative_temperature_expressed_as_heat_content_due_to_parameterized_submesoscale_advection |
| 15 | ocontemppmdiff | tendency_of_sea_water_conservative_temperature_expressed_as_heat_content_due_to_parameterized_mesoscale_diffusion |
| 16 | ocontempdiff | tendency_of_sea_water_conservative_temperature_expressed_as_heat_content_due_to_parameterized_dianeutral_mixing |
| 17 | osalttend | tendency_of_sea_water_salinity_expressed_as_salt_content |
| 18 | osaltrmadvect | tendency_of_sea_water_salinity_expressed_as_salt_content_due_to_residual_mean_advection |
| 19 | osaltpadvect | tendency_of_sea_water_salinity_expressed_as_salt_content_due_to_parameterized_eddy_advection |
| 20 | osaltpsmadvect | tendency_of_sea_water_salinity_expressed_as_salt_content_due_to_parameterized_submesoscale_advection |
| 21 | osaltpmdiff | tendency_of_sea_water_salinity_expressed_as_salt_content_due_to_parameterized_mesoscale_diffusion |
| 22 | osaltdiff | tendency_of_sea_water_salinity_expressed_as_salt_content_due_to_parameterized_dianeutral_mixing |

**Table 10.** This table summarizes fields that support the study of three-dimensional ocean heat and salt budgets, listing here terms contributing to the time tendency of heat and salt in a model grid cell, mass and thickness of a cell, and depth integrated heat and salt. Annual means fields are Priority=1 for the FAFMIP experiment (Gregory et al., 2016), whereas the monthly means are Priority=2. For all other experiments, the annual means are Priority=3, as listed here. For models with prognostic temperature given by potential temperature, then these models should fill the potential_temperature fields and leave the conservative_temperature fields blank; conversely for models with Conservative Temperature as the prognostic temperature field. The column indicating the experiment for saving the diagnostics generally says "all", in which case we recommend the diagnostic be saved for CMIP6 experiments in which there is an ocean model component, including the DECK, historical simulations, FAFMIP, DAMIP, DCPP, ScenarioMIP, and C4MIP, as well as the ocean-sea ice OMIP simulations. Entries with grids denoted "sphere (native)" denote diagnostics where spherical output is strongly recommended to facilitate analysis, though where native output is accepted if spherical is unavailable. The lower sub-table lists the CMOR names and the corresponding CF standard names.





respectively

$$\frac{\partial (C\rho\,\mathrm{d}z)}{\partial t} = -\nabla_s \cdot [\rho\,\mathrm{d}z\,(\boldsymbol{u}\,C + \rho\,\boldsymbol{F})]$$
$$+ \left[ (w\,\rho\,C + \rho\,F^{(s)}) \right]_{s=s_{k=1}}$$
$$+ Q_{\mathrm{advect}}^{(c)} + Q_{\mathrm{non\text{-}advect}}^{(c)} + \mathcal{S}^{(c)} \tag{77}$$

$$\frac{\partial (C\rho\,\mathrm{d}z)}{\partial t} = -\nabla_s \cdot [\rho\,\mathrm{d}z\,(\boldsymbol{u}\,C + \boldsymbol{F})]$$
$$- [\rho\,(w\,C + F^{(s)})]_{s=s_{k-1}}$$
$$+ [\rho\,(w\,C + F^{(s)})]_{s=s_k} + \mathcal{S}^{(c)} \tag{78}$$

$$\frac{\partial (C\rho\,\mathrm{d}z)}{\partial t} = -\nabla_s \cdot [\rho\,\mathrm{d}z\,(\boldsymbol{u}\,C + \boldsymbol{F})]$$
$$- \left[ \rho\,(w\,C + F^{(s)}) \right]_{s=s_{kbot-1}}$$
$$+ Q_{(\mathrm{bot})}^{(c)} + \mathcal{S}^{(c)}. \tag{79}$$

These budgets are formulated as finite volume contributions to the tracer mass (or heat) per horizontal area of a grid cell, with the horizontal area of the grid cell assumed constant in time. The left hand side of these equations represents the net time tendency for the tracer content in a grid cell, per horizontal area of the cell. The right hand side arises from the convergence of

10 advective and subgrid scale fluxes crossing the faces of a grid cell, as well as boundary fluxes and sources.

A schematic of ocean model grid cells over an ocean column is shown in Figure 1. Grid cells generally have a non-constant thickness and non-constant density (although Boussinesq budgets have constant density factor $\rho \rightarrow \rho_o$). The lateral convergence operator acting on an advective or subgrid scale flux is formulated numerically so that multiplication by the respective area of a grid cell face leads to a difference operator acting on the lateral flux components crossing the tracer grid cell faces. That is, the

15 numerical discretization satisfies Gauss's Law (Section 9.9), as doing so allows us to retain the familiar finite volume budgets within the numerical model. We now detail terms in these budgets.

- $C$ is the potential (or Conservative) temperature of a grid cell, or the mass of tracer (e.g., salt or another material tracer) per mass of seawater within the cell (i.e., tracer concentration).

- $\rho\,\mathrm{d}z$ is the mass of seawater per horizontal area in a grid cell, with $\rho$ the *in situ* density and $\mathrm{d}z$ the thickness. For

Boussinesq models, the $\rho$ factor is replaced by a constant reference density $\rho_o$ (rhozero in Section 4). Furthermore, for Boussinesq models with fixed grid cell thicknesses (Section 4.8), the thickness factor $\mathrm{d}z$ is a temporal constant.

- The product $C\rho\,\mathrm{d}z$ is the mass per unit horizontal area of a grid cell if $C$ is a material tracer such as salinity. Since the horizontal area of the cell is constant in time, we may multiply by the horizontal area to recover a budget for the mass in the cell.





- The product $C\,\rho\,\mathrm{d}z$ is the heat per horizontal area if $C$ is potential or Conservative Temperature multiplied by the heat capacity. Since the horizontal area of the cell is constant in time, we may multiply by the horizontal area to recover a budget for the heat within the grid cell, in SI units of joule.

- The generalized vertical coordinate is denoted by $s$, and its discrete values $s_k$ determine the vertical grid cell.

- The horizontal velocity component is $\boldsymbol{u}$ and dia-surface component is $w = (\partial z/\partial s)\,\mathrm{D}s/\mathrm{D}t$, with $\mathrm{D}/\mathrm{D}t$ the material time derivative.

- The horizontal subgrid scale transport is $\rho\,\boldsymbol{F}$ and dia-surface component is $\rho\,F^{(s)}$.

- Tracer flux associated with the boundary water flux is accounted for by the term $Q_{\mathrm{advect}}^{(c)}$. That is, this term accounts for the heat content of the mass crossing the ocean surface, with discussion of this term given in Sections 8.4, 8.5, and 8.6.

- $Q_{(\mathrm{bot})}^{(c)}$ is the flux of tracer passed into the liquid ocean through the solid bottom boundary, such as through geothermal heating (Section 8.4).

- $Q_{\mathrm{non\text{-}advect}}^{(c)}$ is the non-advective flux of tracer crossing the ocean surface boundary. The sign is defined so that a positive value represents a flux of tracer into the ocean; e.g., positive sign adds heat, salt, carbon, or other tracers to the ocean. For the heat budget, this term arises from such terms as shortwave, longwave, latent, and sensible heat fluxes (Section 8.4).

- $\mathcal{S}^{(c)}$ is the tracer source term, which is critical for biogeochemical tracers as per the biogeochemical portion of OMIP (Orr et al., 2016). Sources are zero for heat and salt. Notably, CMIP6 models generally do *not* include joule heating.

### 9.2 Mass integrated prognostic temperature over an ocean column

We request the mass weighted depth integrated prognostic temperature. For models using potential temperature as their prognostic temperature field, the relevant diagnostic is opottempmint, whereas models that use Conservative Temperature as their prognostic temperature field (see Appendix D), the diagnostic is ocontempmint. These diagnostics are computed as a depth integrated sum

- opottempmint $= \sum\limits_{k} \theta\,\rho\,\mathrm{d}z$

- ocontempmint $= \sum\limits_{k} \Theta\,\rho\,\mathrm{d}z$

where the *in situ* density factor, $\rho$, is set to the reference density, $\rho = \rho_o$ (rhozero in Table 1), for Boussinesq fluids, and where $\theta$ is potential temperature and $\Theta$ is Conservative Temperature, $\Theta$.





## 9.3 Mass integrated salinity over an ocean column

To facilitate a quick assessment of the salt content in an ocean column, for purposes of closing the ocean model salt budget, it is useful to save annual mean mass weighted depth integrated salinity

- $\text{somit} = \sum_k S\rho \, \mathrm{d}z,$

where the *in situ* density factor, $\rho$, is set to the reference density, $\rho = \rho_o$ (rhozero in Table 1), for Boussinesq fluids.

## 9.4 Processes diagnosed for heat and salt budgets

There are numerous physical processes contributing to the evolution of heat and salt in a grid cell. It is not practical to request all such terms for CMIP. Rather, we aim to archive a suite of terms whose physical content is both interesting and generally nontrivial. In addition to the boundary salt fluxes detailed in Sections 8.3, and the boundary heat fluxes and penetrative

shortwave radiation detailed in Section 8.4, we recommend archiving the following three-dimensional terms associated with advective and parameterized subgrid scale transport. For the heat budget terms, save the potential temperature terms if that is the model prognostic temperature, otherwise save the Conservative Temperature terms.

### 9.4.1 Net tendency of heat and salt in a grid cell

- opottemptend = net time tendency for heat (via potential temperature) in a grid cell due to *all* processes.

- ocontemptend = net time tendency for heat (via Conservative Temperature) in a grid cell due to *all* processes.

- osalttend = net time tendency for salt in a grid cell due to *all* processes.

This term captures the net time tendency for heat and salt within a grid cell arising from all processes. For potential temperature, this term takes the form

$$\text{opottemptend} = c_p^o \left( \frac{\partial \left( \theta \rho \, \mathrm{d}z \right)}{\partial t} \right), \tag{80}$$

and likewise for Conservative Temperature and salinity. It is crucial that this term encompass all processes affecting the tracer, as its residual from other diagnosed terms will be used to infer contributions from unsaved processes.

### 9.4.2 Residual mean advection

- opottemprmadvect = convergence of residual mean advective fluxes of heat (via potential temperature)

- ocontemprmadvect = convergence of residual mean advective fluxes of heat (via Conservative Temperature)

- osaltrmadvect = convergence of residual mean advective fluxes of salt.



This term measures the time tendency of heat and salt due to the three-dimensional convergence of fluxes from the residual mean velocity, $v^{\dagger} = v + v^{*}$, where $v^{*}$ is the parameterized eddy-induced advection velocity, including both mesoscale and sub-mesoscale processes (see Section 6.1). If there are no eddy-advective parameterizations, then $v^{*} = 0$, in which case advection occurs solely from the resolved model prognostic velocity.

### 9.4.3 Net parameterized eddy advection

- opottemppadvect = convergence of parameterized eddy advective fluxes of heat (via potential temperature)

- ocontemppadvect = convergence of parameterized eddy advective fluxes of heat (via Conservative Temperature)

- osaltpadvect = convergence of parameterized eddy advective fluxes of salt

This term measures the time tendency of heat and salt due to the convergence of three-dimensional fluxes from just the parameterized eddy-induced velocity. The eddy-induced velocity can arise from mesoscale, submesoscale, and/or other processes. Notably, the eddy advection parameterizations can be implemented either as a traditional advection process, or as a skew transport, with their convergence the same in the continuum (Griffies, 1998). If there are no eddy advection parameterizations, then $v^{*} = 0$, in which case this budget term is absent.

### 9.4.4 Parameterized submesoscale eddy advection

- opottemppsmadvect = convergence of parameterized submesoscale eddy-advective fluxes of heat (via potential temperature)

- ocontemppsmadvect = convergence of parameterized submesoscale eddy-advective fluxes of heat (via Conservative Temperature)

- osaltpsmadvect = convergence of parameterized submesoscale eddy-advective fluxes of salt

This term measures the time tendency of heat and salt due to the convergence of three-dimensional fluxes from just the submesoscale eddy parameterization. The canonical form for this closure is that from Fox-Kemper et al. (2008) and Fox-Kemper et al. (2011). Note that Bachman and Fox-Kemper (2013) propose a diffusive component to the submesoscale parameterization. However, we know of no model making use of this scheme for CMIP6.

Note that if there only mesoscale plus submesoscale processes contribute to the eddy-induced advective transport, then knowledge of the net parameterized eddy transport advective transport, opottemppadvect for example, as well as the submesoscale contribution, opottemppsmadvect, allows for the mesoscale contribution to be inferred by subtraction.

### 9.4.5 Parameterized mesoscale diffusion

- opottemppmdiff = convergence of parameterized mesoscale eddy-diffusive fluxes of heat (via potential temperature)





- ocontemppmdiff = convergence of parameterized mesoscale eddy-diffusive fluxes of heat (via Conservative Temperature)

- osaltpmdiff = convergence of parameterized mesoscale eddy-diffusive fluxes of salt

This term measures the time tendency from the convergence of parameterized diffusive fluxes associated with mesoscale closures. Such diffusion is usually oriented according to neutral directions or isopycnal directions (Solomon (1971), Redi (1982), Griffies et al. (1998)). We note the added complexity associated with the use of an implicit time stepping for the diagonal vertical portion of the rotated diffusion tensor (Cox, 1987; Griffies et al., 1998). Care should be exercised to include this portion of the rotated diffusion as part of this online diagnostic calculation.

### 9.4.6 Parameterized vertical and dia-neutral diffusion

- opottempdiff = convergence of parameterized dianeutral / vertical diffusive fluxes of heat (via potential temperature)

- ocontempdiff = convergence of parameterized dianeutral / vertical diffusive fluxes of heat (via Conservative Temperature)

- osaltdiff = convergence of parameterized dianeutral / vertical eddy-diffusive fluxes of salt

This term measures the time tendency from the convergence of parameterized fluxes associated with dia-neutral (or diapycnal) processes as well as vertical boundary layer processes. These parameterizations are generally implemented as downgradient diffusion. In addition, this term can include mixing due to vertical convective adjustment. This budget term includes

1. convection via an enhanced vertical diffusivity

2. convection via a convective adjustment scheme (e.g., Rahmstorf (1993))

3. boundary layer mixing (e.g., Large et al. (1994))

4. interior shear driven mixing (e.g., Pacanowski and Philander (1981); Jackson et al. (2008))

5. gravity wave induced mixing (e.g., Simmons et al. (2004); Jayne (2009); Melet et al. (2013))

6. static background diffusion (e.g., Bryan and Lewis (1979))

7. other vertical diffusion processes.

### 9.4.7 Remaining processes

Contributions from remaining processes can be inferred as a residual by taking the difference of all diagnosed processes from the net tendency. Residual processes may include non-local KPP mixing (Large et al., 1994); and mixing from overflow schemes (e.g., Beckmann and Döscher, 1997; Campin and Goosse, 1999; Danabasoglu et al., 2010; Bates et al., 2012), and mixing across straits (e.g., see Section 3.5 of Griffies et al. (2005)).





### 9.4.8 Penetrative shortwave radiation

Heat absorbed in a tracer cell due to penetrative shortwave radiation is saved in the diagnostic rsdoabsorb. This diagnostic is discussed in Section 8.6.12.

### 9.4.9 Summary heat budget

5  The heat budget (via potential temperature or Conservative Temperature) for a surface grid cell is dependent on how shortwave radiation is treated (see discussion in Section 8.6.12). For models where the shortwave at the top of the ocean, rsntds, is included only in hfds, not in rsdoabsorb, then the surface grid cell budget takes the form

$$
\begin{aligned}
\text{opottemptend} = \text{hfds} &+ \text{rsdoabsorb} + \text{opottempadvect} \\
&+ \text{opottemppmdiff} + \text{opottempdiff} \\
&+ \text{other}.
\end{aligned}
\tag{81}
$$

A complement treatment assumes that all of the surface shortwave flux, rsntds, participates in penetrative shortwave heating 10  (see Section 8.6.12). In this case, rsdoabsorb accounts for the shortwave flux converging into the top cell. Consequently, the surface grid cell balance (81) reads

$$
\begin{aligned}
\text{opottemptend} = \text{hfds} &+ (\text{rsdoabsorb} - \text{rsntds}) + \text{opottempadvect} \\
&+ \text{opottemppmdiff} + \text{opottempdiff} \\
&+ \text{other}.
\end{aligned}
\tag{82}
$$

The budget for an interior grid cell is realized by removing the net surface heat flux hfds (Section 8.4). The budget for a bottom grid cell is the same, with hfds replaced by the bottom geothermal heat flux, hfgeou. The budget for salt is analogous, 15  yet without the penetrative shortwave heat flux, rsdoabsorb, nor the bottom geothermal heat flux.

### 9.5 Conventions for the heat budget terms

Following from the tracer budget given by equations (77)-(79), all heat budget terms take the form[24]

$$
Q_{\text{process(n)}}^{(\Theta)} = c_p^o \left( \frac{\partial (\rho \, \mathrm{d}z \, \Theta)}{\partial t} \right)_{\text{process(n)}},
\tag{83}
$$

where $n$ labels the particular physical process. The physical units for the heat budget terms are thus given by

20  $Q_{\text{process(n)}}^{(\Theta)} \; [\equiv] \; \mathrm{W \, m^{-2}}.$
            (84)

The area normalization for each budget term corresponds to the horizontal area of the tracer grid cell. Multiplication of any budget term by the tracer grid cell horizontal area thus yields the heat content change for that grid cell in units of watts.

---

[24]We use $\Theta$ in this section, as appropriate for TEOS-10 models. For pre-TEOS-10 models, they should archive tendencies appearing in the potential temperature, $\theta$ equation. See Appendix D for discussion of seawater thermodynamics.





### 9.6 Conventions for the salt budget terms

Following from the tracer budget given by equations (77)-(79), all salt budget terms for archiving into CMIP take the general form

$$Q_{\text{process(n)}}^{(S)} = \frac{1}{1000} \left( \frac{\partial (\rho \, dz \, S)}{\partial t} \right)_{\text{process(n)}} \quad \text{kg m}^{-2} \, \text{s}^{-1}, \tag{85}$$

where $S$ is the salinity in units of ppt = gram of salt per kilogram of seawater or psu, depending on the model salinity field (see Appendix D), and $n$ labels the particular physical process. Division by 1000 converts grams to kilograms. Multiplication of any budget term by the tracer grid cell horizontal area thus yields the salt content change for that grid cell in units of kilogram per second.

### 9.7 Temperature tendency terms

The heat budget term (83) scales according to the thickness of a cell. This is expected, since the budget determines the change in heat content per horizontal area of a cell, and this is the prognostic term in the ocean model.

For diagnostic purposes, it may be useful to consider a temperature tendency corresponding to the heat budget terms, with the temperature tendency in units of $^\circ C \, \text{s}^{-1}$. Doing so removes dependence on the grid cell thickness. That is, we may choose to consider the tendency for an intensive quantity, temperature, rather than the budget for an extensive quantity, heat. For this

purpose, we recommend dividing the heat budget terms in equation (83) by the annual mean mass per horizontal area of a grid cell, according to

$$\delta\Theta_{\text{process(n)}} = \frac{Q_{\text{process(n)}}^{(\Theta)}}{c_p^o \, \rho \, dz} \quad [\equiv] \,^\circ C \, \text{s}^{-1}. \tag{86}$$

The factor $\rho \, dz$ is the annual mean mass per unit area of a grid cell (masscello), requested in Section 5.3. In this way, we can map vertical sections of the tendency terms and thus remove dependence on the grid cell thicknesses. Note that for a Boussinesq

model with grid cell thicknesses that are time independent (Section 4.8), temperature tendency terms are trivially related to the heat budget terms.

### 9.8 Salinity tendency terms

Likewise, we may convert the salt budget terms into salinity tendencies in units of ppt/s. For this purpose, we may divide the salt budget terms in equation (85) according to

$$\delta S_{\text{process(n)}} = \frac{Q_{\text{process(n)}}^{(S)}}{\rho \, dz} \quad [\equiv] \, \text{ppt s}^{-1}. \tag{87}$$

The $\rho \, dz$ array is the annual mean mass per unit area of a grid cell, requested in Section 5.3. Note that for a Boussinesq model with grid cell thicknesses that are time independent, salinity tendency terms are trivially related to the salt budget terms.



## 9.9 Diagnosing a flux vector versus a flux convergence

In the mechanistic analysis of budgets, one is often interested in assessing budgets over a region, such as an ocean basin or within a subregion of a basin. These regional budgets help to identify dominant processes contributing to changes in heat and salt within the region, which in turn can help characterize physical mechanisms. For many purposes, this sort of analysis may
involve characterizing the fluxes of heat and salt crossing the regional boundaries, in which case the three components of a flux vector may be required.

However, it is ultimately the convergence of a flux vector into a region that causes the change in heat or salt in that region. Additionally, fluxes remain arbitrary up to the curl of a scalar, since the curl of a scalar has zero convergence. One therefore must be careful when focusing an analysis on fluxes. Further words of caution are summarized in the appendix of Gregory
(2000), who studied global and regional heat budgets, as well as in the study of equatorial Pacific heat budgets by Lee et al. (2004).

We are not advocating outright abandonment of flux components for mechanistic analyses, rather cautioning in their use absent consideration for their convergences into a region. It is largely for this reason that we prefer saving budget terms comprised of the convergence of fluxes associated with various physical processes. Besides saving archive space relative to
15 saving fluxes (by a factor of three), we are assured that the budget analysis is making use of terms that directly contribute to the changes in heat and salt within a region.

We furthermore note that integration of the divergence over a region leads, through Gauss's Law, to the sum of the fluxes crossing the boundary of the region

$$\iiint_{\mathcal{R}} \nabla \cdot \boldsymbol{F} \, \mathrm{d}V = \iint_{\partial \mathcal{R}} \hat{\boldsymbol{n}} \cdot \boldsymbol{F} \, \mathrm{d}S, \tag{88}$$

where $\mathcal{R}$ is an arbitrary volume of fluid, $\partial \mathcal{R}$ is the boundary of $\mathcal{R}$, and $\hat{n}$ is the outward normal on the boundary. Hence, by integrating a flux divergence (negative of the convergence) over a chosen volume (left hand side), one can garner mechanistic insight into the impacts from various physical processes in that region, without having to make direct use of flux components (right hand side).

## 10 Vertical/dianeutral subgrid scale parameterizations

In Table 11, we present fields to help characterize subgrid scale (SGS) parameterizations and their impact on the simulation, with focus on the vertical/dianeutral.

### 10.1 Vertical/dianeutral tracer diffusivities

- difvho = total vertical heat diffusivity

- difvso = total vertical salt diffusivity



| VERTICAL SGS PARAMETERIZATIONS | | | | | | | | | |
|---|---|---|---|---|---|---|---|---|---|
| ITEM | CMOR NAME | SPONSOR | CMIP5/CMIP6 | UNITS | TIME | SHAPE | GRID | PRIORITY | EXPT |
| 1 | difvho | OMIP/FAFMIP | (month → annual) and (hist → all) | $m^2/s$ | annual | XYZ | sphere (native), z/p | 3 | all |
| 2 | difvso | OMIP/FAFMIP | (month → annual) and (hist → all) | $m^2/s$ | annual | XYZ | sphere (native), z/p | 3 | all |
| 3 | tnpeo | OMIP/FAFMIP | (month → annual) and (hist → all) | $W/m^2$ | annual | XY | sphere (native) | 3 | all |

| CMOR NAME RELATED TO CF STANDARD NAME | | |
|---|---|---|
| ITEM | CMOR NAME | CF STANDARD NAME |
| 1 | difvho | ocean_vertical_heat_diffusivity |
| 2 | difvso | ocean_vertical_salt_diffusivity |
| 3 | tnpeo | tendency_of_ocean_potential_energy_content |

**Table 11.** This table summarizes diagnostics that support the study of vertical/dianeutral subgrid scale (SGS) parameterizations. The column indicating the experiment for saving the diagnostics generally says "all", in which case we recommend the diagnostic be saved for CMIP6 experiments in which there is an ocean model component, including the DECK, historical simulations, FAFMIP, DAMIP, DCPP, Scenari-oMIP, and C4MIP, as well as the ocean-sea ice OMIP simulations. We ask only for annual means from these fields, rather than the monthly means requested for most other diagnostics. Additionally, this table has been reduced from the 12 fields requested in CMIP5 to the three requested here. Furthermore, the field tnpeo was requested as a three-dimensional field in CMIP5, whereas for CMIP6 it is requested as a depth integrated two-dimensional field. Entries with grids denoted "sphere (native)" denote diagnostics where spherical output is strongly recommended to facilitate analysis, though where native output is accepted if spherical is unavailable. The lower sub-table lists the CMOR names and the corresponding CF standard names.

Vertical/dianeutral tracer diffusivities used in modern CMIP models typically consist of a static background value and a dynamically determined value. For the background diffusivity, some modellers choose a globally constant value, whereas others impose spatial dependence. There is evidence that the background diffusivity influences such processes as tropical currents (Meehl et al., 2001), and overturning strength (Bryan, 1987) in model simulations.

There are an increasingly large number of physical processes used by CMIP-class models that affect the vertical tracer diffusivity. For example, vigorous mixing processes in the upper ocean are associated with large mixing coefficients; more quiescent processes in the ocean pycnocline lead to much smaller coefficients; and enhanced mixing near the ocean bottom generally increases the mixing coefficients. The background, tidal, and boundary layer diffusivities are the same for temperature, salinity, and other tracers. However, the total diffusivities may differ if including a parameterization of double diffusive processes. We

request archival of just the net heat diffusivity in difvho and salt diffusivity in difvso.

### 10.2 Rate of work against stratification

- tnpeo = tendency of potential energy due to vertical mixing

### 10.2.1 Summary of the diagnostic

A vertical/dianeutral diffusivity impacts the solution predominantly where there are nontrivial vertical tracer gradients. A

measure of the impact can be deduced by mapping the rate at which work is done against the stratification by the tracer




diffusivity. This work against stratification also impacts the potential energy budget. We recommend mapping this work rate per horizontal area as a depth integrated two-dimensional field.

### 10.2.2 Theoretical considerations

The non-negative rate of work done against stratification by vertical/dianeutral diffusion of density is given by

$$\mathcal{P} \equiv \int \kappa_{\mathrm{d}} N^2 \rho \, \mathrm{d}V, \tag{89}$$

where $N^2$ is the squared buoyancy frequency (see the obvfsq diagnostic discussed in Section 5.21), and $\kappa_{\mathrm{d}}$ is the vertical/dianeutral diffusivity corresponding to a particular subgrid scale process. Equation (89) assumes the heat and salt diffusivities are the same, which is the case for tidal and background diffusivities. However, the full heat diffusivity, $\kappa_{\mathrm{d}}^{\theta}$, and salt diffusivity, $\kappa_{\mathrm{d}}^{S}$, can differ through effects from double diffusion. In this case, we split the integral as

$$\mathcal{P} \equiv -g \int \left( \kappa_{\mathrm{d}}^{\theta} \frac{\partial \rho}{\partial \theta} \frac{\partial \theta}{\partial z} + \kappa_{\mathrm{d}}^{S} \frac{\partial \rho}{\partial S} \frac{\partial S}{\partial z} \right) \mathrm{d}V. \tag{90}$$

This term should be archived as a two-dimensional map of depth integrated mixing work

$$\mathrm{tnpeo} = -g \int_{-H}^{\eta} \left( \kappa_{\mathrm{d}}^{\theta} \frac{\partial \rho}{\partial \theta} \frac{\partial \theta}{\partial z} + \kappa_{\mathrm{d}}^{S} \frac{\partial \rho}{\partial S} \frac{\partial S}{\partial z} \right) \mathrm{d}z. \tag{91}$$

Multiplication by the horizontal grid area, then summing over the globe, provides the global amount of work done by vertical mixing.

## 11 Lateral subgrid scale parameterizations

We now detail diagnostics helping to characterize lateral subgrid scale (SGS) parameterizations, with Table 12 summarizing the diagnostics. As for the vertical/dianeutral SGS parameterizations, we propose that dominant scientific use of the fields discussed in this subsection are realized by archiving *just* the annual mean fields.

### 11.1 Lateral tracer diffusivities

- diftrelo = diffusivity for parameterized epineutral mesoscale eddy-induced Laplacian diffusion

- diftrblo = diffusivity for parameterized mesoscale eddy-induced advection

These diffusivities are used for neutral diffusion (Solomon, 1971; Redi, 1982), and eddy-induced advective transport (Gent et al., 1995).

### 11.2 Eddy kinetic energy source from mesoscale parameterization

- tnkebto = tendency of eddy kinetic energy from parameterized eddy advection





| LATERAL SGS PARAMETERIZATIONS | | | | | | | | | |
|---|---|---|---|---|---|---|---|---|---|
| ITEM | CMOR NAME | SPONSOR | CMIP5/CMIP6 | UNITS | TIME | SHAPE | GRID | PRIORITY | EXPT |
| 1 | diftrblo | OMIP/FAFMIP | (month → annual) and (hist → all) | $m^2/s$ | annual | XYZ | sphere (native), z/p | 3 | all |
| 1 | diftrelo | OMIP/FAFMIP | (month → annual) and (hist → all) | $m^2/s$ | annual | XYZ | sphere (native), z/p | 3 | all |
| 2 | tnkebto | OMIP/FAFMIP | (month → annual) and (hist → all) | $W/m^2$ | annual | XY | sphere (native) | 3 | all |
| 4 | difmxylo | OMIP/FAFMIP | (month → annual) and (hist → all) | $m^2/s$ | annual | XYZ | sphere (native), z/p | 3 | all |
| 5 | difmxybo | OMIP/FAFMIP | (month → annual) and (hist → all) | $m^4/s$ | annual | XYZ | sphere (native), z/p | 3 | all |
| 6 | dispkexyfo | OMIP/FAFMIP | (month → annual) and (hist → all) | $W/m^2$ | annual | XY | sphere (native) | 3 | all |

| CMOR NAME RELATED TO CF STANDARD NAME | | |
|---|---|---|
| ITEM | CMOR NAME | CF STANDARD NAME |
| 1 | diftrblo | ocean_tracer_diffusivity_due_to_parameterized_mesoscale_advection |
| 2 | diftrelo | ocean_tracer_epineutral_laplacian_diffusivity |
| 3 | tnkebto | tendency_of_ocean_eddy_kinetic_energy_content_due_to_parameterized_eddy_advection |
| 4 | difmxylo | ocean_momentum_xy_laplacian_diffusivity |
| 5 | difmxybo | ocean_momentum_xy_biharmonic_diffusivity |
| 6 | dispkexyfo | ocean_kinetic_energy_dissipation_per_unit_area_due_to_xy_friction |

**Table 12.** This table summarizes diagnostics that support the study of lateral subgrid scale (SGS) parameterizations. The column indicating the experiment for saving the diagnostics generally says "all", in which case we recommend the diagnostic be saved for CMIP6 experiments in which there is an ocean model component, including the DECK, historical simulations, FAFMIP, DAMIP, DCPP, ScenarioMIP, and C4MIP, as well as the ocean-sea ice OMIP simulations. We ask only for annual means from these fields, rather than the monthly means requested for most other diagnostics. Additionally, this table has been reduced from the ten fields requested in CMIP5 to the six requested here. Furthermore, note that the fields tnkebto and dispkexyfo were requested as three-dimensional fields for CMIP5, whereas they are now requested as depth integrated two-dimensional fields. Entries with grids denoted "sphere (native)" denote diagnostics where spherical output is strongly recommended to facilitate analysis, though where native output is accepted if spherical is unavailable. The lower sub-table lists the CMOR names and the corresponding CF standard names.

An energetic analysis of the extraction of potential energy by the Gent et al. (1995) scheme indicates that it affects an increase in the eddy kinetic energy (Aiki and Richards, 2008; Eden and Greatbatch, 2008; Marshall and Adcroft, 2010). The rate of eddy kinetic energy increase, per unit horizontal area over an ocean column, is

$$\text{tnkebto} = \int_{-H}^{\eta} \kappa \, (N \, S)^2 \, \rho \, \mathrm{d}z. \tag{92}$$

In this expression, $N$ is the buoyancy frequency, $S$ is the magnitude of the neutral slope, $\kappa$ is the diffusivity setting the overall strength of the parameterization, $\rho \, \mathrm{d}z$ is the grid cell mass per horizontal area, with $\mathrm{d}z$ the cell thickness. In a Boussinesq model, the *in situ* density factor should be set to the constant Boussinesq reference density $\rho_o$ used by the model (see rhozero in Table 1). Note that the CMIP5 request asked for the full three-dimensional field, whereas for CMIP6 we only ask for the depth integrated two-dimensional field to reduce the archive burden.

Horizontal maps of the column integrated work from the mesoscale parameterization (92) can be readily compared across the suite of CMIP models. This depth integrated field also provides a means for directly comparing the work done by vertical





diffusion (Section 10.2). Furthermore, multiplication by the horizontal grid area, then summing over the globe, provides the global amount of work associated with the scheme.

### 11.3 Lateral momentum viscosities

- difmxylo = Laplacian viscosity

- difmxybo = biharmonic viscosity

These viscosities are generally time dependent. Note that we do not make the distinction between various methods used to compute the lateral momentum viscosities. Hence, we only recommend the total fields be archived from the ocean models in CMIP6.

### 11.4 Kinetic energy dissipation by lateral viscosity

- dispkexyfo = kinetic energy dissipation from lateral friction

#### 11.4.1 Summary of the diagnostic

As for the vertical/dianeutral viscosity, we recommend archiving the maps of energy dissipation from lateral viscous friction integrated over a full ocean column. The diagnostic dispkexyfo accounts for dissipation from the sum of Laplacian plus biharmonic friction active in the model.

#### 11.4.2 Theoretical considerations

The local energy dissipated in a hydrostatic model by a lateral Laplacian friction with isotropic viscosity $A$ and anisotropic viscosity $D$ is given by the non-positive quantity (see Section 17.8.2 of Griffies, 2004)

$$\mathcal{D} = -(\rho \, \mathrm{d}V) \left[ A \left( e_\mathrm{T}^2 + e_\mathrm{S}^2 \right) + 2 \, D \, \Delta^2 \right], \tag{93}$$

where

$$e_\mathrm{T} = (\mathrm{d}y) \, (u/\mathrm{d}y)_{,x} - (\mathrm{d}x) \, (v/\mathrm{d}x)_{,y} \tag{94a}$$
$$e_\mathrm{S} = (\mathrm{d}x) \, (u/\mathrm{d}x)_{,y} + (\mathrm{d}y) \, (v/\mathrm{d}y)_{,x} \tag{94b}$$

are the deformation rates,

$$2 \, \Delta = e_\mathrm{S} \cos 2\vartheta - e_\mathrm{T} \sin 2\vartheta \tag{95}$$

is a measure of the anisotropy of the viscous opearator with $\vartheta$ an angle that sets the alignment of the generally anisotropic

viscosity (Large et al., 2001; Smith and McWilliams, 2003), and $\mathrm{d}x$ and $\mathrm{d}y$ are the horizontal grid elements. We recommend





archiving depth integrated dissipation per horizontal area

$$\text{dispkexyfo}^{\text{Laplacian}} = -\int_{-H}^{\eta} (\rho \, dz) \left[ A(e_T^2 + e_S^2) + 2D\,\Delta^2 \right]. \tag{96}$$

The local energy dissipated in a hydrostatic model by a lateral biharmonic friction is given by the non-positive quantity (see Section 17.9.2 of Griffies, 2004)

$$\mathcal{D} = -(\rho \, dV) \, \boldsymbol{F} \cdot \boldsymbol{F}, \tag{97}$$

where $\rho \, dV \, \boldsymbol{F}$ is the lateral Laplacian friction vector used to build up the biharmonic operator. As for the dissipation from vertical viscosity, we recommend mapping the dissipation per horizontal area for each column of seawater, as given by

$$\text{dispkexyfo}^{\text{biharmonic}} = -\int_{-H}^{\eta} (\rho \, dz) \, \boldsymbol{F} \cdot \boldsymbol{F}. \tag{98}$$

Many models make use of *both* Laplacian and biharmonic friction. To account for the net dissipation of kinetic energy, the diagnostic dispkexyfo is comprised of the sum

$$\text{dispkexyfo} = \text{dispkexyfo}^{\text{Laplacian}} + \text{dispkexyfo}^{\text{biharmonic}}. \tag{99}$$

## 12 Summary and closing comments

OMIP has two components: an experimental protocol and a diagnostics protocol. The experimental protocol, detailed in Section 2, follows from the interannual Coordinated Ocean-ice Reference Experiments (CORE-II) (Section 2.1). The Large and Yeager (2009) atmospheric state used for CORE-II is used here for OMIP Phase I. Additionally, we are in the process of developing a Phase II of OMIP that aims to overcome some limitations of the Large and Yeager (2009) state, with the new efforts making use of the JRA-55 reanalysis (Kobayashi et al., 2015).

The OMIP diagnostic protocol is built from the CMIP5 ocean diagnostics of Griffies et al. (2009a). Describing the diagnostics formed the bulk of this paper. We could easily have asked for more diagnostics. Nonetheless, we must balance the desire to generate a vast diagnostic archive with the realities of limited human and archive resources. We furthermore cannot justify a substantial increase in ocean diagnostics without explicit sponsorship of the diagnostics under the expectation that the diagnostics will be used for peer-review studies. It is with this background in mind that we aimed to be true to the needs of analysts who, we hope, will extensively mine the CMIP6/OMIP model data. We also aimed to design a diagnostic suite attractive to process physicists who wish to look "under the hood" of the models.

It will take some years to determine the success of OMIP as an experimental protocol and as a diagnostic protocol. Regardless, it is our hope that this document will serve as a useful reference point for further endeavours such as this one. Quite generally, global models are firmly embedded in the science of oceanography and climate. Careful design of experiments and diagnostics, with correspondingly thorough and creative analysis of their output across a suite of models, offers ongoing opportunities for novel scientific insights and robust predictive skill.





**Data Availability**

The model output from the DECK and CMIP6 historical simulations described in this paper will be distributed through the Earth System Grid Federation (ESGF) with digital object identifiers (DOIs) assigned. As in CMIP5, the model output will be freely accessible through data portals after registration. In order to document CMIP6's scientific impact and enable ongoing

support of CMIP, users are obligated to acknowledge CMIP6, the participating modelling groups, and the ESGF centres. Details can be found on the CMIP Panel website at

http://www.wcrp-climate.org/index.php/wgcm-cmip/about-cmip.

Further information about the infrastructure supporting CMIP6, the metadata describing the model output, and the terms governing its use are provided by the WGCM Infrastructure Panel (WIP) in their invited contribution to this Special Issue

(Balaji et al., 2016).

Along with the data itself, the provenance of the data will be recorded, and DOI's will be assigned to collections of output so that they can be appropriately cited. This information will be made readily available so that published research results can be verified and credit can be given to the modelling groups providing the data.

The WIP is coordinating and encouraging the development of the infrastructure needed to archive and deliver this informa-

tion. In order to run the experiments, datasets for natural and anthropogenic forcings are required. These forcing datasets are described in separate invited contributions to this Special Issue. The forcing datasets will be made available through the ESGF with version control and DOIs assigned.

*Acknowledgements.* This document benefitted from the direct and indirect input of numerous ocean and climate scientists who contributed to the various CORE-II analysis papers (see Section 2.1). We also thank many scientists at various modelling centres who provided candid

and essential feedback needed to help optimize the scientific utility of OMIP. Thanks also go to Bill Hurlin and Jasmin John at GFDL for commenting on early drafts of this paper. NCAR is sponsored by the U.S. National Science Foundation. BFK was supported by NSF 1245944 and 1350795. The work of P.J.D., P.J.G. and K.E.T. from Lawrence Livermore National Laboratory is a contribution to the US Department of Energy, Office of Science, Climate and Environmental Sciences Division, Regional and Global Climate Modeling Program under contract DE-AC52-07NA27344. The work of Uotila [or FMI] was supported by the Academy of Finland (contracts 264358 and 283034). Y. Komuro

and H. Tsujino are supported by JSPS KAKENHI Grand Number 15H03726.

**Appendix A:  Grid cell volume and horizontal area**

In order to calculate ocean area integrals and volume integrals, information is needed to weight the grid cells in a manner consistent with conservation properties of the model. All CMIP6 ocean models have a fixed horizontal grid and hence constant horizontal cell areas. Cell areas (areacello) should be stored in each data file (Table 1) in order to keep this important information

within each diagnostic file. In contrast, the cell thicknesses (thkcello), and hence cell volumes and masses (masscello), may



be time dependent for many ocean models. We discuss in this section how to specify ocean grid cell volumes and masses for CMIP6.

## A1 Volume-conserving Boussinesq ocean models

Boussinesq ocean models are based on volume-conserving kinematics, with these models having been used since the early days of ocean modelling (Bryan, 1969). For budget purposes, Boussinesq models use a constant reference density for seawater, $\rho_o$. Hence, the grid cell mass, $\mathrm{d}M$, is equal to the grid cell volume, $\mathrm{d}V$, multiplied by the constant reference density (Section 5.3)

$$\mathrm{d}M = \rho_o \, \mathrm{d}V \qquad \text{Boussinesq models (kg).} \tag{A1}$$

A netCDF scalar variable containing the constant

$$\rho_o = \text{rhozero} \qquad (\text{kg m}^{-3}) \tag{A2}$$

should be archived in the same file (Table 1).

### A1.1 Boussinesq models with static grid cell volumes

Certain Boussinesq ocean models assume that grid cells have time-independent volumes, meaning they have static grid cell thicknesses. This property holds for geopotential Boussinesq ocean models based on barotropic dynamics using either the rigid lid approximation (Bryan, 1969; Pinardi et al., 1995) or linearized free surface (Dukowicz and Smith, 1994; Roullet and Madec, 2000). By construction, these models do not allow for changes in the volume associated with boundary water fluxes. Consequently, they must make use of virtual tracer flux boundary conditions discussed in Section 8.1.7 as well as Huang and Schmitt (1993) and Griffies et al. (2001).[25]

If the Boussinesq model has time-independent grid cell volumes, then the grid cell masses (equation (A1)) are also constant in time. For these models, the CMIP6 masscello field for cell mass (sea_water_mass_per_unit_area) is to be saved as a static XYZ variable measuring the mass per area of the tracer grid cell (Table 1). Furthermore, for cells which occupy the entire vertical extent of the grid cell layer (i.e. except for partial cells at the top or bottom of the ocean), the cell thickness can be calculated as the difference of the depth-bounds for the layer. This thickness should equal the cell mass per unit area divided by the Boussinesq reference density. Only for these models, the cell_thickness variable thkcello is not required.

### A1.2 Boussinesq models with time-dependent grid cell volumes

Many Boussinesq models have time dependent cell volumes, with examples including isopycnal models, terrain-following sigma models, and stretched depth-coordinate $z^*$ models (Section B5). For these models, the cell thickness, thkcello (Table 2), is time dependent. A separate masscello file is required for each distinct set of time coordinates at which other monthly

---

[25]Some models based on fully nonlinear split-explicit free surface methods also retain virtual salt fluxes for historical reasons.





XYZ scalar fields are provided (Tables 2, 10, 11, and 12). Doing so provides a one-to-one correspondence between the variable to be weighted (e.g. thetao) and the variable providing the weights (masscello). For Boussinesq models, the reference density rhozero should also be saved in each masscello file. The cell thickness, thkcello, is not required, since it can be easily diagnosed through

$$\text{thkcello} = \text{masscello}/(\text{areacello} * \text{rhozero}) \quad (\text{m}). \tag{A3}$$

In contrast, for typical non-Boussinesq models (see Section A2), both masscello and thkcello are required on the same time frequency as the primary fields (e.g., monthly).

## A2   Mass-conserving non-Boussinesq ocean models

Non-Boussinesq models are based on mass-conserving kinematics (Griffies and Greatbatch, 2012). When hydrostatic, such
models are naturally formulated using pressure, or a function of pressure, as the vertical coordinate (Huang et al., 2001; DeSzoeke and Samelson, 2002; Marshall et al., 2004). If based on pressure, then the mass of a grid cell remains constant in time, with the equations isomorphic to the depth-coordinate Boussinesq ocean equations.

In general, the cell thickness, thkcello, and cell mass, masscello (Table 2.2), are time dependent. A separate masscello file is required for each distinct set of time coordinates at which other XYZ scalar fields are provided (Tables 2, 10, 11, and 12).
Doing so provides a one-to-one correspondence between the variable to be weighted (e.g. thetao) and the variable providing the weights (masscello).

## A3   Details of the grid information

The link between a scalar data variable and the corresponding areacello and masscello variables is made using the cell_measures and associated_files attributes available in a netCDF file. For a field on an XY longitude-latitude horizontal grid, the file should
contain variables written as

```
float pr(time,latitude,longitude);
  pr:cell_measures="area: areacello";
  pr:standard_name="rainfall_flux";
  pr:units="kg m-2 s-1";
float areacello(latitude,longitude);
  areacello:standard_name="cell_area";
  areacello:units="m2";
```

The areacello variable is not required to have the variable name areacello. In cell_measures, "area: VARNAME" identifies the variable by name.





For a field on an XYZ grid, the file should contain variables written as

```
float thetao(time,depth,latitude,longitude);
  thetao:cell_measures="area: areacello mass_per_unit_area: masscello";
  thetao:associated_files="BASENAME";
  thetao:standard_name="sea_water_potential_temperature";
  thetao:units="degC";
float areacello(latitude,longitude);
  areacello:standard_name="cell_area";
  areacello:units="m2";
```

The field BASENAME is the basename (the last element of the path) of the file containing masscello for the same times as for the primary field. That mass file contains

```
  float masscello(time,depth,latitude,longitude);
    masscello:standard_name="sea_water_mass_per_unit_area";
    masscello:units="kg m-2";
```

where the time dimension and coordinate variable must have the same names and contents for the two files.

### Appendix B: Details for spatial sampling

In this Appendix, we offer further details regarding how to sample scalar and vector fields in space. Our discussion raises issues about both vertical and horizontal sampling. Further issues related to gridding and remapping for CMIP6 are discussed more thoroughly in the WGCM Infrastructure Panel (WIP) contribution to this CMIP6 special issue (Balaji et al., 2016).

#### B1   Integration over spatial regions

We start with an easy question: how to sample fields to be integrated over a spatial region, such as a basin or section? The answer is to compute the integral using all model grid points within the relevant domain and time average using all model time steps. There should be no sub-sampling in space or time.

#### B2   Horizontal and vertical remapping for scalars

We offer here some specific comments about remapping scalar fields, such as tracers.

#### B2.1   Horizontal remapping

Horizontal grids are static for ocean climate modelling. Hence, remapping to a spherical grid can occur offline. Conservative remapping of scalars may be desirable for some purposes, such as for those diagnostics meant for budget analysis (e.g., boundary fluxes in Tables 6, 7, 8, and budget terms in Table 10). In this case, all horizontal area factors must be properly handled, with care required especially in the presence of the complex ocean geometry. For other purposes, conservative remapping may not be necessary nor ideal, particularly as some conservative methods can introduce noise in the regridded fields. In either case,





groups aiming to archive data on a one-degree class of grid are encouraged to map to standard spherical grid defined by the one-degree used for the World Ocean Atlas (Levitus, 1982; Locarnini et al., 2013).

## B2.2    Vertical remapping

In models with time dependent grid cell volumes/masses (e.g., isopycnal models, sigma coordinate models, vertical ALE models), it is critical that vertical remapping occur online for each model time step to include correlations between the fluctuating grid cell geometry and the scalar field. Remapping subsampled fields in such cases generally leads to erroneous results; it must be avoided.

- For models based on $z$ (geopotential coordinate), stretched depth $z^*$, pressure, or stretched pressure $p^*$, there is no need to perform a depth remapping, unless aiming to remap to a standard vertical grid such as the 33-levels used by the World Ocean Atlas (Levitus, 1982; Locarnini et al., 2013).

- For models with a time dependent grid cell thickness that do not use $z, z^*, p$, or $p^*$ vertical coordinates (e.g., isopycnal and terrain following), the vertical remapping step should be computed each model time step to ensure exact conservation. See Appendix B5 for more details of these coordinates.

- Vertical remapping should occur onto a vertical coordinate based on depth (for Boussinesq models) or pressure (for non-Boussinesq models).

- Pressure based vertical grids should be measured in dbar, in order to facilitate easy comparison to depth-based models using metres.

- Depth and pressure increase downward from the ocean surface, whereas the vertical geopotential $z$ increases upward starting from the resting ocean surface at $z = 0$.

- For those choosing to coarsen the vertical resolution of their archived diagnostics, modellers are encouraged to map onto the 33-levels used for the World Ocean Atlas (Levitus, 1982; Locarnini et al., 2013).

## B3    Vector fields

It is mathematically straightforward to transform (e.g., rotate) a continuum vector field from one coordinate system to another using methods of tensor analysis (e.g., Chapter 20 in Griffies (2004)). Unfortunately, these continuum mathematical methods are ambiguous for discrete vector fields. For example, the commonly used C-grid has horizontal velocity components sitting at distinct spatial positions, thus breaking the tensorial character of the continuum vector field. Tracer fluxes are likewise positioned at the tracer cell sides for all finite volume models.

We recommend that any remapping of vector components onto a sphere (into north-south and east-west vector components) be based on a high order (higher than linear) interpolation scheme, thus ensuring smoothness and accuracy. But further manipulations of the remapped vector fields is discouraged, since the remapped vector fields can have spurious divergence and curls.





Conservation is generally not needed for vector components. Furthermore, for native vector components sitting on a C-grid, these components should be mapped onto an A-grid or B-grid, depending on what is more convenient based on the native model grid. Interpolating vector components to a single point greatly facilitates routine interpretation. See Balaji et al. (2016) for more details.

## B4   Zig-zag method for estimating poleward transports

We recommend that each group using non-spherical grids develop a native-grid algorithm that computes the closest native grid approximation to the basin integrated poleward transports. That is, transports across a section (e.g. meridional overturning at a given latitude, transport through a passage, or vertically integrated poleward heat transport) should be computed consistent with the native grid by finding a nearly equivalent path to the section that has been "snapped" to the native grid (often resulting in a "zig-zag" path) (e.g., see Figure C2 of Forget et al. (2015)). This approach retains the native grid variables, and so allows for conservation of transports. It also avoids ambiguities associated with defining a remapped land/sea mask. The resulting transports should be made available as a function of latitude (even though the integrations are not exactly along latitude circles). The latitude spacing should be comparable to that of the model grid spacing.

## B5   More details for vertical sampling

There are two questions to answer regarding the vertical coordinate:

- Should model output be remapped in the vertical to a common vertical coordinate?

- If remapped, then what is a scientifically relevant vertical coordinate?

There is no ambiguity regarding the vertical grid when working with Boussinesq rigid lid geopotential-coordinate ocean models, as each grid has a fixed vertical position. It was thus sensible for WGCM (2007) to recommend that output in the vertical be on a geopotential grid, preferably remapped to the 33 depth levels used by (Levitus, 1982; Locarnini et al., 2013). The more recent trend towards free surface geopotential models raises only trivial issues with the surface grid cell, and these issues can be ignored without much loss of accuracy.[26] However, the move towards pressure, isopycnal, terrain following, and general/hybrid models increases the complexity of vertical coordinate questions.

We make the following observations and clarifications regarding the recommendations for vertical remapping.

- For isopycnal, terrain following, and general/hybrid models, we recommend remapping to $z^*$ for Boussinesq models and $p^*$ for non-Boussinesq ($z^*$ and $p^*$ are defined below). This recommendation is based on the predominant needs of analysis. The one exception concerns the overturning streamfunction, which is archived on both geopotential/pressure *and* density surfaces (Sections 6.5 and 6.6).

---

[26]We know of no group that considers the question of remapping model fields in the top model cell of a free surface geopotential model to a pre-defined geopotential level. Indeed, there is little reason to do so, as the top cell, whether it has a center at $z = -1$m or $z = 1$m, for example, still provides the model version of sea surface properties.





- Conservative vertical remapping with straightforward linear interpolation is reasonably accurate so long as the remapping is done every model time step. Remapping subsampled fields can lead to erroneous analysis, especially with isopycnal models, and so must be avoided.

- Contrary to the situation in the horizontal, separate vector components can be treated as scalars for the purpose of remapping in the vertical.

### B5.1 Rescaled geopotential for Boussinesq models

For Boussinesq models, it is natural to consider remapping to the *rescaled geopotential* coordinate (Stacey et al., 1995; Adcroft and Campin, 2004)

$$z^* = H\left(\frac{z-\eta}{H+\eta}\right). \tag{B1}$$

In this equation, $z$ is the geopotential, $z = -H(x,y)$ is the ocean bottom, and $z = \eta(x,y,t)$ is the deviation of the free surface from a resting ocean at $z = 0$. To better understand the ratio, note that $z - \eta$ is the thickness of seawater above a particular geopotential, and $H + \eta$ is the total thickness of seawater in the fluid column. Surfaces of constant $z^*$ correspond to geopotentials when $\eta = 0$. For most practical applications of global ocean modelling, $z^*$ surfaces only slightly deviate from constant geopotential surfaces even with nonzero $\eta$ fluctuations. The advantage of $z^*$ over geopotential is that it has a time independent range $-H \leq z^* \leq 0$, thus allowing for a more straightforward mapping from a free surface isopycnal or terrain following model.

### B5.2 Rescaled pressure for non-Boussinesq models

The *rescaled pressure* coordinate is defined as

$$p^* = p_\mathrm{b}^o\left(\frac{p-p_\mathrm{a}}{p_\mathrm{b}-p_\mathrm{a}}\right), \tag{B2}$$

where $p$ is the pressure at a grid point; $p_\mathrm{a}(x,y,t)$ is the pressure applied at the ocean surface due to overlying atmosphere, sea ice, and/or ice shelves; $p_\mathrm{b}(x,y,t)$ is the pressure at the ocean bottom; and $p_\mathrm{b}^o(x,y)$ is a static reference bottom pressure, such as the initial bottom pressure. To better understand the ratio, note that in a hydrostatic ocean, $g^{-1}(p-p_\mathrm{a})$ is the mass per horizontal area of seawater situated above a pressure level $p$, and $g^{-1}(p_\mathrm{b}-p_\mathrm{a})$ is the total mass per horizontal area of seawater in the fluid column. For most practical applications of global modelling, constant $p^*$ surfaces only slightly deviate from constant pressure surfaces, even with nonzero fluctuations of $p_\mathrm{b}$. The advantage of $p^*$ over pressure is that $p^*$ has a time independent range $0 \leq p^* \leq p_\mathrm{b}^o$, thus allowing for a more straightforward mapping from a non-Boussinesq model making use of alternative vertical coordinates.

### B5.3 Visualization and analysis purposes

For visualization purposes, the distinction between geopotential (or rescaled geopotential) and pressure (or rescaled pressure) can be ignored to within great accuracy, so long as geopotential is measured in metres and pressure is measured in decibars.



For analysis purposes, the distinction between geopotential (or rescaled geopotential) and pressure (or rescaled pressure) can be ignored when working with model native scalars and fluxes. The differences *cannot* be ignored when performing off-line integration of velocity components to approximate fluxes. This is a central reason that we request mass fluxes in addition to velocity components (Section 6).

## Appendix C: Data precision of archived diagnostics

Besides ensuring proper practices for temporal and spatial sampling, it is important to understand the needs for data precision of the archived diagnostics. This issue is important for two reasons, firstly to ensure that analysts are able to maintain accuracy when calculating derived diagnostics, and secondly to minimize storage footprint, particularly as model resolution and diagnostic requests increase.

### C1    Features of netCDF4

All model data for CMIP6 follows the netCDF4 protocol. This recommendation contrasts to the use of netCDF3 in CMIP5, which was a sensible recommendation since netCDF4 has matured during the recent years after the CMIP5 recommendations were finalized in 2009. The CMIP5 archives adhered to the netCDF3 protocol, meaning that CMIP5 data was written using single precision (32-bit float) format. In contrast, key features of netCDF4 that motivate its use for CMIP6 include lossless compression (deflation) and file access/read performance tools of chunking and shuffling. Furthermore, most standard analysis packages now support netCDF4 formatted data (see Section C2), thus placing no burden on the analyst.

In Table C1, we provide information about precision features of different data formatting within the netCDF4 protocol. For most analysis purposes, single precision (seven significant digits) is sufficient. However, length and area factors from grids may usefully be saved in double precision, given that area factors are the basis for statistical analyses and remapping. Notably, it is rare to find observational-based oceanographic data with significance greater than half precision (three significant digits).

### C2    Software packages supporting netCDF4

NetCDF4 is now a standard library for many software packages, including those listed in Table C2. Consequently, using netCDF4 deflation (and reducing file sizes by roughly 50% in a lossless format) should pose no hindrance for CMIP6 analysis. Note that the 50% compression assumes that the supplied netCDF4 libraries are built with HDF5 and zlib support, which are needed to garner the compression functionality.

## Appendix D: Seawater thermodynamics

We offer specifications for treating seawater thermodynamics, in particular for temperature, salinity, and associated heat and salt transports. These specifications are made in light of the endorsement by the international oceanography community of the Thermodynamic Equation of State 2010 (TEOS-10) (IOC et al., 2010). TEOS-10 is based on a consistent theory of seawater





## DATA PRECISION

| NAME | DESCRIPTION | NETCDF4 TYPE | PRECISION | MINIMUM | MAXIMUM | DECIMAL DIGITS | EXAMPLE (35.1234567891) | DELTA |
|---|---|---|---|---|---|---|---|---|
| i8 | 8-bit signed integer | NC_byte (byte) | - | -128 | 127 | - | 35.0708661417 | -0.052590647391170364 |
| u8 | 8-bit unsigned integer | NC_ubyte (unsigned byte) | - | 0 | 255 | - | 35.0708661417 | -0.052590647391170364 |
| i16 | 16-bit signed integer | NC_short (short) | - | -32768 | 32767 | - | 35.1235694449 | 0.00011265574485719299 |
| u16 | 16-bit unsigned integer | NC_ushort (unsigned short) | - | 0 | 65535 | - | 35.1235694449 | 0.00011265574485719299 |
| i32 | 32-bit signed integer | NC_int (int) | - | -2147483648 | 2147483647 | - | 35.123456786 | -3.1449545190298522e-09 |
| u32 | 32-bit unsigned integer | NC_uint (unsigned int) | - | 0 | 4294967295 | - | 35.123456786 | -3.1449545190298522e-09 |
| i64 | 64-bit signed integer | NC_int64 (long long) | - | -9223372036854775808 | 9223372036854775807 | - | 35.1234567891 | 0.0 |
| u64 | 64-bit unsigned integer | NC_uint64 (unsigned long long) | - | 0 | 18446744073709551615 | - | 35.1234567891 | 0.0 |
| f16/binary16 | 16-bit floating-point | - | half | $-\infty$ | $\infty$ | 3.31 | 35.125 | 0.0015432108765480734 |
| f32/binary32 | 32-bit floating-point | NC_float (float) | single | $-\infty$ | $\infty$ | 7.22 | 35.123455 | -1.7415160300515709e-06 |
| f64/binary64 | 64-bit floating-point | NC_double (double) | double | $-\infty$ | $\infty$ | 15.95 | 35.1234567891 | 0.0 |

**Table 13.** Listing of available data types using the netCDF4 data model, and example data precision for ocean salinity using these data types. Plausible observed salinity ranges (PSS-78; Lewis and Perkin (1981)) are between 2 and 42 (open ocean salinities normally range from 32 to 38), which is the range over which the Practical Salinity Scale (1978) provides coverage. This scale is used to define the valid_min and valid_max from which the representative salinity precision (last column) is calculated. For reference, the PSS-78 scale specifies an *in-situ* temperature range of $-2°C$ to $35°C$, and a pressure range of 0 to $10^4$ dbar (Lewis and Perkin, 1981). For further details on the precision, see `http://en.wikipedia.org/wiki/IEEE_floating_point#Basic_formats` and `https://www.unidata.ucar.edu/software/netcdf/docs/netcdfc/NetCDF_002d4AtomicTypes.html`

| SOFTWARE SUPPORTING NETCDF4 | |
|---|---|
| NAME | EARLIEST VERSION |
| CDO | 1.5 |
| CDAT | 5.2 |
| Ferret | 6.6 |
| IDL | 7 |
| Matlab | R2010b |
| NCO | 3.1 |
| NCL | 6.1.1 |
| Python | 2.6 |
| CF-Python | 1.0 |

**Table 14.** Table of software products that support netCDF4. Listed are the software packages and the earliest version of that software that supports netCDF4.





thermodynamics, as well as empirical measurements updated since the UNESCO-80 equation of state. TEOS-10 represents a major move forward in the fundamental science and practice of seawater thermodynamics.

### D1    Balancing the needs

This document aims to provide a rational and practical framework for meaningful comparisons across climate models and
observational based measurements. Meeting this aim supports the primary means whereby analysis of CMIP simulations contributes to climate science. In offering recommendations for seawater thermodynamics, we must balance the desire to remain true to IOC et al. (2010), while acknowledging the practical needs for a successful model intercomparison.

The document

http://www.teos-10.org/pubs/Getting_Started.pdf

provides a starting point for incorporating TEOS-10 into ocean models, and Roquet et al. (2015) offer more specific steps. Nonetheless, many modelling groups are just now incorporating TEOS-10 into their CMIP6 models, with some unable to realize this transition in time for CMIP6.

Hence, CMIP6 will contain models based on TEOS-10, and others based on pre-TEOS-10. Furthermore, there remain unanswered research questions raised by IOC et al. (2010), in particular regarding the treatment of salinity. For CMIP6, we
cannot impose strict standards defining what it means to be "TEOS-10 compliant", when research remains incomplete. Indeed, at this time, there are zero peer-reviewed publications using ocean climate simulations based on the suite of recommendations from TEOS-10. In short, the community is in a transition stage from pre-TEOS-10 to TEOS-10. For CMIP6, we thus offer a cosmopolitan approach rather than one based on a well defined territory.

### D2    Specification for temperature

Regardless the model thermodynamics, modellers should archive potential temperature, $\theta$. For models using pre-TEOS-10 ocean thermodynamics, no change is required relative to previous CMIPs. For models using TEOS-10 thermodynamics, in which Conservative Temperature, $\Theta$, is the model prognostic field, we still recommend archiving potential temperature to allow for meaningful comparisons. Doing so requires an online diagnostic calculation to convert at each time step from Conservative Temperature to potential temperature. Additionally, we request TEOS-10 based models to archive Conservative Temperature,
anticipating that future CMIPs will naturally see more models based on Conservative Temperature rather than potential temperature.

### D3    Specification for heat content

The air-sea flux of heat is exactly the air-sea flux of potential enthalpy (since the reference gauge pressure of potential enthalpy is 0 dbar). Apart from warming caused by the dissipation of turbulent kinetic energy (as well as another smaller term), potential
enthalpy is a conservative variable in the ocean (McDougall, 2003; Graham and McDougall, 2013), meaning that it satisfies a scalar conservation equation analogous to a source-free material tracer. Because of these properties of potential enthalpy, we



are justified in calling it the heat content of seawater. That is, the heat content (in joules) of a seawater parcel or an ocean model grid cell is

$$\text{seawater heat content} = h^o \, \rho \, \mathrm{d}V \qquad (D1)$$

with

5 $$h^o = c_p^o \, \Theta \qquad (D2)$$

the potential enthalpy per mass, $\Theta$ the Conservative Temperature, $\mathrm{d}V$ the parcel or grid cell volume, and $\rho$ the *in situ* seawater density. The seawater heat capacity, as *defined* by TEOS-10, is the constant

$$c_p^o = 3991.86795711963 \, \mathrm{J \, kg^{-1} \, K^{-1}}. \qquad (D3)$$

The 15 significant digits in $c_p^o$ is a based on a numerical fit. The observation-based data used in this fit are measured to a
10 precision no greater than three or four significant digits. Hence, there is no physics in $c_p^o$ beyond roughly four significant digits.

Ocean climate models measure heat content (in joules) of a grid cell according to

$$\text{model heat content} = c_p^o * \text{prognostic temperature} * \rho \, \mathrm{d}V. \qquad (D4)$$

We now comment on this model practice and relate it to TEOS-10 and CMIP6.

### D3.1   Boussinesq reference density

For a Boussinesq fluid, mass, tracer, and momentum budgets replace the *in situ* density, $\rho$, with a constant reference density, $\rho_o$ (except for the buoyancy force, where $\rho \, g$ retains the *in situ* density). Not all groups use the same constant (see Roquet et al. (2015) for a discussion of various choices). Modellers should therefore archive in CMIP this constant according to the request in Table 1.

### D3.2   Heat capacity

The ocean model heat capacity, $c_p^o$, is constant. However, the ocean model heat capacity is not always equal to the TEOS-10 recommended value given by equation (D3). We thus ask to archive the model heat capacity in Table 1. We note that the TEOS-10 heat capacity $c_p^o$ (equation (D3)) was chosen so that the surface area average (and ocean mass average) of $c_p^o \, \theta$ closely matches the corresponding surface area (and ocean mass) averages of potential enthalpy. We thus highly recommend models choose a heat capacity $c_p^o$ for both pre-TEOS-10 and TEOS-10 usage.

### D3.3   Heat content

Expression (D4) is the heat content for the respective TEOS-10 and pre-TEOS-10 ocean models. This expression is relevant for CMIP6 since the model prognostic temperature field evolves according to grid cell budgets. Hence, pre-TEOS-10 models



should *not* archive heat content by diagnosing the Conservative Temperature. Rather, they should measure heat content as always done for previous CMIPs, using the model prognostic potential temperature. Likewise, TEOS-10 models should measure heat content using the TEOS-10 recommendation (D2), using the model prognostic Conservative Temperature field.

### D3.4 Boundary heat fluxes

As noted by McDougall (2003), boundary heat fluxes affect the ocean potential temperature, with a tendency proportional to the reciprocal of the specific isobaric heat capacity of seawater. Importantly, this heat capacity varies by 5% over the ocean. However, no ocean climate model makes use of a non-constant specific isobaric heat capacity, even though the temperature field of ocean models is most often interpreted as potential temperature. This inconsistency is motivated by the desire to have the model ocean heat content related directly to the model prognostic temperature field, with that temperature field time stepped according to conserved budget equations. Turning this inconsistency into an opportunity, McDougall (2003) noted that ocean models using a constant heat capacity, $c_p^o$, may in fact be interpreted as using Conservative Temperature rather than potential temperature. There are errors associated with this interpretation arising from the calculation of *in situ* density and boundary heat fluxes. Nonetheless, these errors may in fact be smaller than those associated with ignoring the non-constant heat capacity. Research is needed to further pursue this interpretation.

### D3.5 Heat transport

Heat transport and its convergence are determined by various transport processes (e.g., advection, diffusion) impacting on the grid cell heat content (D4). We ask for the archival of such transports and convergences in Tables 3 and 10.

### D4 Specification for salinity

Ocean models based on pre-TEOS-10 thermodynamics carry a salinity variable that approximates the observed quantity of Practical Salinity - the observed variable from which most ocean model initial states are obtained (often from a version of the World Ocean Atlas such as Zweng et al. (2013)). The model version of Practical Salinity is influenced by transport in the ocean interior, and through atmosphere and terrestrial boundary freshwater fluxes that alter the salt mass concentration. However, these similarities between model and observations are more subtle when we consider the new salinity definitions provided with TEOS-10 (IOC et al., 2010).

TEOS-10 aims to better quantify a poorly constrained aspect of the observed quantity of ocean salinity. The new salinity definitions are preferred over Practical Salinity because the thermodynamic properties of observed seawater are directly influenced by the mass of dissolved constituents whereas Practical Salinity depends only on conductivity (and coincident temperature). For example, exchange a small mass of pure water with the same mass of silicate in an otherwise isolated seawater sample maintained at constant temperature and pressure. Since silicate is predominantly non-ionic, the conductivity (and therefore Practical Salinity) is unchanged to measurement precision. In contrast, the Absolute Salinity and density is increased. Similarly, if a small mass of sodium chloride (NaCl) is added and the same mass of silicate is removed, the salinity mass fraction





will not have changed (and so the density will also remain near constant) but the Practical Salinity (measured by conductivity) will have increased. The TEOS-10 Reference Composition of sea salt comprises 15 chemical species, of which Cl− and Na+ comprise 55% and 31% respectively, with $SO_4^{2-}$ and $Mg^{2+}$ the next most abundant species at 7.7% and 3.6% (Pawlowicz et al., 2016).

While the new TEOS-10 salinities are more representative of the real world sea water constituents, the observing platforms and techniques currently used to obtain sea water salinity measurements have not changed.[27] This situation has led to TEOS-10 advocating for a continuation of the current practice of Practical Salinity (measured through well defined conductivity relationships) being the stored quantity in observed oceanographic databases. However, when observed analyses are being undertaken, TEOS-10 recommends that Absolute Salinity, rather than Practical Salinity be used.

The enhanced treatment of ocean salinity defined by TEOS-10 leads, unfortunately, to a divergence between observed and modelled salinity quantities. Most models that are contributing to CMIP6 are based on pre-TEOS-10 thermodynamics and carry only salinity that approximates the observed quantity of Practical Salinity. Therefore, to facilitate comparisons across models and with observations, salinity comparisons in CMIP6 will be made against Practical Salinity regardless of the model thermodynamics. This practice represents exact correspondence to earlier CMIPs, and reflects the same role of modelled

salinity when considering modelled sea water thermodynamics.

### D5    Specification for salt content

The salt content in a grid cell is *not* given by the grid cell mass times Practical Salinity. Instead, it is given by the grid cell mass times Absolute Salinity. However, for CMIP6, the differences between Practical Salinity and Absolute Salinity will likely be ignored by all modelling groups, given the early stages of such research. In this case, salt content is approximated by the grid

cell mass times the Practical Salinity.

### Appendix E:  Temperature scales and ocean heat content

Ocean heat content is arbitrary up to specification of the ocean temperature scale. However, the evolution of ocean heat content is invariant when shifting ocean temperature scales, such as when changing from kelvin to celsius. To show this property, consider the heat content of the global ocean as written in Appendix A.4 of Griffies et al. (2014)

$$\mathcal{H} = c_p^o \mathcal{M} \langle \Theta \rangle^\rho. \tag{E1}$$

In this expression, $\mathcal{M}$ is the total mass of seawater,

$$\langle \phi \rangle = \mathcal{V}^{-1} \int \phi \, dV \tag{E2}$$

is the volume mean operator with $\mathcal{V}$ the total ocean volume,

$$\langle \Theta \rangle^\rho = \frac{\langle \rho \, \Theta \rangle}{\langle \rho \rangle} \tag{E3}$$

---

[27] A historical survey of these practices is provided by Durack et al. (2013).





is the density weighted mean Conservative Temperature, and $\langle \rho \rangle = \mathcal{M}/\mathcal{V}$ is the mean ocean density. As discussed in Section 8.1.6, time changes in ocean heat content are affected by non-advective and advective heat fluxes crossing the ocean boundary

$$\frac{\mathrm{d}\mathcal{H}}{\mathrm{d}t} = \mathcal{A}\left(\overline{Q}_{\text{non-advect}} + \overline{Q}_{\text{advect}}\right). \tag{E4}$$

In this equation, $\mathcal{A}$ is the surface area of the ocean, $\overline{Q}_{\text{non-advect}}$ comprise the area mean radiative and turbulent heat fluxes, and $\overline{Q}_{\text{advect}}$ is the area mean advective heat flux. The non-advective heat fluxes are determined by the thermodynamic temperature; i.e., kelvin. Likewise, a portion, $\overline{Q}_{\text{advect}}^{\text{other}}$, of the advective heat flux is generally determined outside of the ocean; for example, when the atmosphere or river model transfers the heat content of precipitation or river runoff to the ocean. Another portion of the advective heat flux is determined by the ocean. We write that heat content in the form

$$\overline{Q}_{\text{advect}} = \overline{Q}_{\text{advect}}^{\text{other}} + c_p^o \overline{Q^{\mathrm{m}}\Theta^{\mathrm{m}}}, \tag{E5}$$

where $Q^{\mathrm{m}}$ is that portion of the boundary mass flux whose heat content is determined by the ocean, and $\Theta^{\mathrm{m}}$ is the Conservative Temperature of the boundary mass flux. In practice, we often approximate $\Theta^{\mathrm{m}}$ by the surface ocean temperature, but that is not necessary for the present arguments. Bringing these results together leads to the time changes in the heat content

$$\frac{\mathrm{d}\mathcal{H}}{\mathrm{d}t} = \mathcal{A}\left(\overline{Q}_{\text{non-advect}} + \overline{Q}_{\text{advect}}^{\text{other}} + c_p^o \overline{Q^{\mathrm{m}}\Theta^{\mathrm{m}}}\right). \tag{E6}$$

When Conservative Temperature is measured in kelvin, the ocean heat content is related to the celsius heat content by the offset (see definition of the heat content in equation (E1))

$$\mathcal{H}^{(\kappa)} = \mathcal{H}^{(c)} + c_p^o \mathcal{M}\,\Theta^{(\text{C2K})}, \tag{E7}$$

where $\Theta^{(\text{C2K})} = -273.15$ is the constant offset in the scales. Hence, their time derivatives are related by

$$\frac{\mathrm{d}\mathcal{H}^{(\kappa)}}{\mathrm{d}t} = \frac{\mathrm{d}\mathcal{H}^{(c)}}{\mathrm{d}t} + c_p^o \mathcal{A}\overline{Q^{\mathrm{m}}}\,\Theta^{(\text{C2K})}, \tag{E8}$$

where

$$\frac{\mathrm{d}\mathcal{M}}{\mathrm{d}t} = \mathcal{A}\overline{Q^{\mathrm{m}}} \tag{E9}$$

is the time change of the global ocean mass. When the ocean mass is constant, evolution of ocean heat content does not care what temperature scale is used. To show that this result holds in general, without loss of generality assume that the ocean heat budget equation (E6) holds when measuring ocean Conservative Temperature in kelvin, $\Theta_{(\kappa)}$, so that

$$\frac{\mathrm{d}\mathcal{H}^{(\kappa)}}{\mathrm{d}t} = \mathcal{A}\left(\overline{Q}_{\text{non-advect}} + \overline{Q}_{\text{advect}}^{\text{other}} + c_p^o \overline{Q^{\mathrm{m}}\Theta_{(\kappa)}^{\mathrm{m}}}\right). \tag{E10}$$

Now shift to the celsius temperature scale, $\Theta_{(c)}^{\mathrm{m}}$, bringing the right hand side to the form

$$\frac{\mathrm{d}\mathcal{H}^{(\kappa)}}{\mathrm{d}t} = \mathcal{A}\left(\overline{Q}_{\text{non-advect}} + \overline{Q}_{\text{advect}}^{\text{other}} + c_p^o \overline{Q^{\mathrm{m}}\Theta_{(c)}^{\mathrm{m}}} + c_p^o \overline{Q^{\mathrm{m}}}\,\Theta^{(\text{C2K})}\right). \tag{E11}$$



Equating the budgets as written by equations (E8) and (E11) then yields the heat content budget using the celsius scale

$$\frac{\mathrm{d}\mathcal{H}^{(c)}}{\mathrm{d}t} = \mathcal{A}(\overline{Q}_{\text{non-advect}} + \overline{Q}_{\text{advect}}^{\text{other}} + c_p^o\,\overline{Q^{\text{m}}\Theta_{(c)}^{\text{m}}}). \tag{E12}$$

As anticipated, the ocean heat budget using the celsius scale takes the same form as equation (E10), which is the budget for ocean heat using the kelvin temperature scale. We may thus use either temperature scale when analyzing ocean heat budgets.

For OMIP, it is more convenient to use the celsius scale, as that is the scale used for prognostic temperature in ocean models.

### Appendix F: Finite volume scalar equations

In this appendix, we outline a finite volume framework that underlies our requests for heat and salt budget terms in Section 9. This framework is based on the discretization of budget equations for *extensive* fluid properties. Extensive properties include scalars such as seawater mass and tracer mass, as well as vectors such as linear momentum. The scalar concentration, $C$, is an

10 *intensive* property that measures the mass of a scalar field (e.g., salt) in a region per mass of seawater in that region

$$C = \left(\frac{\text{mass of trace matter}}{\text{mass of seawater}}\right). \tag{F1}$$

Enthalpy (heat) also follows this formalism (McDougall, 2003), where the "heat concentration" is the Conservative Temperature. Note that for a Boussinesq fluid, mass can be replaced by volume through division by the constant Boussinesq reference density, $\rho_o$. Correspondingly, when making the Boussinesq approximation, *in situ* density in this appendix is replaced by the

15 reference density.

The finite volume framework offers a useful means to formulate the discretization of budget equations for extensive fluid properties. It is useful since it provides a means to ensure that proper accounting is made for transport of extensive properties between model grid cells. That is, what enters through the ocean boundaries or through source/sink terms fully accounts for the net amount of tracer content within the ocean. Without such *conservative* numerical methods, spurious accumulation or

20 destruction of tracer content can arise from non-conservative numerical methods, with such spurious sources/sinks compromising the physical integrity of the simulation. This point was discussed in Section 5.12. Furthermore, it was illustrated for heat in Appendix C of Griffies et al. (2014), where it was noted that non-conservative numerical choices can lead to spurious heat sources of nontrivial magnitude.

Given a finite volume framework, the problem of how to formulate the discrete ocean model equations shifts from funda-

25 mentals to realizations. Realizations of the framework numerically differ, for example, by the choice of grid cell shape (e.g., quadrilateral, icosohedral, etc.); parameterization of subgrid scale fields; estimation of fluxes on the grid cell faces; representation of domain boundaries; and time discretization. Sorting through these details forms the content of finite volume methods used in computational fluid dynamics. For examples focused on geophysical fluid problems, see Adcroft et al. (1997), Adcroft et al. (2008), Ringler (2011), and Adcroft (2013). Importantly, all methods must respect the conservation equation (F4) derived

below in order to ensure conservation properties of tracer content for the grid cell, as well as for any larger domain in the numerical ocean.





## F1 Formulating the finite volume equations

For a finite fluid region, $\mathcal{R}$, with boundary $\partial\mathcal{R}$, the conservation of mass for an arbitrary tracer can be written

$$\frac{\partial}{\partial t}\left(\iiint\limits_{\mathcal{R}} \rho\, C\, \mathrm{d}V\right) = -\iint\limits_{\partial\mathcal{R}} (\rho\, C\, \delta\boldsymbol{v} + \rho\,\boldsymbol{F})\cdot\hat{\boldsymbol{n}}\,\mathrm{d}A_{(\hat{\boldsymbol{n}})}. \tag{F2}$$

In this equation, $\rho\, C\,\delta\boldsymbol{v}\cdot\hat{\boldsymbol{n}}$ is the advective tracer flux penetrating the boundary with outward normal $\hat{\boldsymbol{n}}$, and $\delta\boldsymbol{v}$ is the velocity
5   of a parcel relative to the velocity of the boundary. $\rho\,\boldsymbol{F}$ is the subgrid scale tracer flux and $\mathrm{d}A_{(\hat{\boldsymbol{n}})}$ is the area element on
the boundary. We ignored sources for brevity, though they can be trivially introduced as needed for biogeochemical tracers.
The volume integral is taken over the region, and the area integral is over the region boundary. That component of the flux
$\rho\,(C\,\delta\boldsymbol{v} + \boldsymbol{F})$ that penetrates the bounding surface alters the scalar content within the region. Now introduce the discrete finite
volume fields with discrete label $J$

$$V_{J} \equiv \iiint\limits_{\mathcal{R}_{J}} \mathrm{d}V \tag{F3a}$$

$$V_{J}\,\rho_{J} \equiv \iiint\limits_{\mathcal{R}_{J}} \rho\,\mathrm{d}V \tag{F3b}$$

$$V_{J}\,\rho_{J}\,C_{J} \equiv \iiint\limits_{\mathcal{R}_{J}} \rho\, C\,\mathrm{d}V. \tag{F3c}$$

For our purposes, an ocean model grid cell forms the canonical example of a finite volume. The discrete field $C_{J}$ is the volume
mean of the continuous tracer concentration, $C$, over the finite domain; it is *not* the value of the continuous tracer evaluated at
15   a point. Making use of definitions (F3a)-(F3c) transforms the continuous conservation equation (F2) into a discrete form

$$\frac{\partial\,(\rho_{J}\,C_{J}\,V_{J})}{\partial t} = -\sum_{\partial\mathcal{R}_{J}} (\rho\, C\,\delta\boldsymbol{v} + \rho\,\boldsymbol{F})\cdot\hat{\boldsymbol{n}}\,A_{(\hat{\boldsymbol{n}})}. \tag{F4}$$

Consequently, the continuous flux form scalar equation (F2) is readily transformed to a finite volume spatially discrete scalar
equation (F4). Likewise, the discrete mass equation is realized by setting the tracer concentration to unity and dropping the
subgrid scale term

$$\frac{\partial\,(\rho_{J}\,V_{J})}{\partial t} = -\sum_{\partial\mathcal{R}_{J}} \rho\,\delta\boldsymbol{v}\cdot\hat{\boldsymbol{n}}\,A_{(\hat{\boldsymbol{n}})}. \tag{F5}$$

The spatially discrete tracer (F4) and mass (F5) equations provide the basis for finite volume scalar equations.

## F2 Thickness weighting in ocean models

Ocean circulation models generally assume the horizontal area of a grid cell is constant in time. It is only the cell thickness that
fluctuates. A time dependent cell thickness applies to models using generalized level or layer coordinates, such as $z^{*}$, terrain





following, isopycnal, and Arbitrary Lagrangian Eulerian (ALE).[28] The finite volume method is quite useful for such models, in which we specialize the finite volume expressions (F3a)-(F3c) to thickness weighted fields

$$A_J \Delta z_J \equiv \iint_{\mathcal{R}_J} \mathrm{d}A \int \mathrm{d}z \qquad (\text{F6a})$$

$$A_J \Delta z_J \rho_J \equiv \iint_{\mathcal{R}_J} \mathrm{d}A \int \rho \, \mathrm{d}z \qquad (\text{F6b})$$

$$A_J \Delta z_J \rho_J C_J \equiv \iint_{\mathcal{R}_J} \mathrm{d}A \int \rho C \, \mathrm{d}z, \qquad (\text{F6c})$$

where $A_J = \iint_{\mathcal{R}_J} \mathrm{d}A$ is the time independent horizontal area of the grid cell, and $\Delta z_J$ is the time dependent cell thickness. Since the horizontal grid cell area is constant in time, the discrete finite volume tracer budget (F4) becomes a budget for the mass of tracer per horizontal area in a grid cell

$$\frac{\partial \left( \Delta z_J \rho_J C_J \right)}{\partial t} = -\frac{1}{A_J} \sum_{\partial \mathcal{R}_J} \left( C \rho \, \delta \boldsymbol{v} + \rho \mathbf{F} \right) \cdot \hat{\mathbf{n}} A_{(\hat{\mathbf{n}})}. \qquad (\text{F7})$$

Likewise, the discrete mass budget for a grid cell, (F5), becomes an equation for the mass per horizontal area,

$$\frac{\partial \left( \Delta z_J \rho_J \right)}{\partial t} = -\frac{1}{A_J} \sum_{\partial \mathcal{R}_J} \rho \, \delta \boldsymbol{v} \cdot \hat{\mathbf{n}} A_{(\hat{\mathbf{n}})}. \qquad (\text{F8})$$

Again, for a Boussinesq fluid, the *in situ* density in the scalar equations is replaced by the constant reference density, $\rho_o$, so that equation (F8) becomes the cell thickness equation.

Vertical cell faces, oriented normal to the horizontal directions $\hat{\boldsymbol{x}}$ and $\hat{\boldsymbol{y}}$, have fixed horizontal positions. Hence, the relative velocity $\delta \boldsymbol{v}$ is just the horizontal velocity so that

$$\delta \boldsymbol{v} \cdot \hat{\boldsymbol{x}} \, \mathrm{d}A_{(\hat{\mathbf{n}})} = u \, \mathrm{d}y \, \mathrm{d}z \qquad (\text{F9a})$$

$$\delta \boldsymbol{v} \cdot \hat{\boldsymbol{y}} \, \mathrm{d}A_{(\hat{\mathbf{n}})} = v \, \mathrm{d}x \, \mathrm{d}z. \qquad (\text{F9b})$$

Likewise, for the vertical faces, defined by surfaces of constant vertical coordinate,

$$\delta \boldsymbol{v} \cdot \hat{\boldsymbol{n}} \, \mathrm{d}A_{(\hat{\mathbf{n}})} = w^{(s)} \, \mathrm{d}x \, \mathrm{d}y, \qquad (\text{F10})$$

where $w^{(s)}$ is the dia-surface velocity component (see Section 6.7 of Griffies (2004) or Section 2.2 of Griffies and Adcroft (2008)).

### F3 Implications for ocean model diagnostics

To ensure conservation of tracer content, ocean models time step the tracer mass per area, $\Delta z_J \rho_J C_J$, as per equation (F7). Furthermore, the time dependent mass per horizontal area, $\Delta z_J \rho_J$, is time stepped through the mass equation (F8). With an updated tracer mass per horizontal area, and an updated seawater mass per horizontal area, we can update the tracer concentration

---

[28]Boussinesq geopotential rigid lid models, or linear free surface models, assume grid cell area and thickness to be time independent. Finite difference methods are sufficient to formulate the discrete equations for these models.





via the division

$$C_J = \frac{\Delta z_J \, \rho_J \, C_J}{\Delta z_J \, \rho_J}. \tag{F11}$$

This update is performed each model time step so as to have access to the tracer concentration at each time step. Tracer concentration is needed, for example, to evaluate the equation of state and freezing point using the salinity (salt concentration), temperature (heat concentration), and pressure. Temperature is also need for computing air-sea fluxes. For OMIP, we therefore request archives for the tracer concentration $C_J$, with time averaging performed online each model time step.

To produce regional maps of an extensive quantity such as salt content or heat content, it is necessary diagnose the mass weighted tracer concentration online to account for temporal correlations between tracer concentration and cell mass. Absent this online calculation, errors are generally unacceptable when working with isopycnal models, where layer thicknesses can vanish. However, experience at GFDL with Boussinesq $z^*$ models suggests that errors are minor when working offline using time averaged thickness and time averaged tracer concentration. Nonetheless, to help reduce errors for general purposes, in Table 10 we request the mass integrated temperature and salinity fields

$$\text{opottempmint} = \sum_k \theta_k \, \Delta z_k \, \rho_k \tag{F12a}$$

$$\text{ocontempmint} = \sum_k \Theta_k \, \Delta z_k \, \rho_k \tag{F12b}$$

$$\text{somint} = \sum_k S_k \, \Delta z_k \, \rho_k, \tag{F12c}$$

where $k$ is the vertical cell index and the sum extends over the full ocean depth. Since the grid cell mass per horizontal area is generally time dependent, the depth integral must be computed online each time step. From this diagnostic, we can compute global and regional integrated heat content and salt content. Furthermore, OMIP requests the online calculation of the mass averaged temperature and salinity (see Table 2)

$$\text{thetaoga} = \frac{\sum_{i,j,k} \theta_{i,j,k} \, \Delta z_{i,j,k} \, \rho_{i,j,k} \, A_{i,j}}{\sum_{i,j,k} \Delta z_{i,j,k} \, \rho_{i,j,k} \, A_{i,j}} \tag{F13a}$$

$$\text{ocontempmint} = \frac{\sum_{i,j,k} \Theta_{i,j,k} \, \Delta z_{i,j,k} \, \rho_{i,j,k} \, A_{i,j}}{\sum_{i,j,k} \Delta z_{i,j,k} \, \rho_{i,j,k} \, A_{i,j}} \tag{F13b}$$

$$\text{somint} = \frac{\sum_{i,j,k} S_{i,j,k} \, \Delta z_{i,j,k} \, \rho_{i,j,k} \, A_{i,j}}{\sum_{i,j,k} \Delta z_{i,j,k} \, \rho_{i,j,k} \, A_{i,j}}. \tag{F13c}$$



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
