# Peer review of "OMIP contribution to CMIP6: experimental and diagnostic protocol for the physical component of the Ocean Model Intercomparison Project"

_Geoscientific Model Development, 2016_

## Short Comment (SC1) · 13 Apr 2016

Dear authors,

In agreement with the CMIP6 panel members, the Executive editors of GMD would like to establish a common naming convention for the titles of the CMIP6 experiment description papers.

The title of CMIP6 papers should include both the acronym of the MIP, and CMIP6, so that it is clear this is a CMIP6-Endorsed MIP.

Additionally, we strongly recommend to add a version number to the MIP descrip­tion. The reason for the version numbers is so that the MIP protocol can be updated later, normally in a second short paper outlining the changes. See, for example: http://www.geosci-model-dev.net/special_issue11.html,

Good formats for the title include:

'XYZMIP (v1.0) contribution to CMIP6: Name of project'

or

'Name of Project (XYZMIP v1.0) contribution to CMIP6'

If you want to include a more descriptive title, the format could be along the lines of,

'XYZMIP (v1.0) contribution to CMIP6: Name of project - descriptive title'

or

'Name of Project (XYZMIP v1.0) contribution to CMIP6: descriptive title.'

When you revise your manuscript, please correct the title of your manuscript accordingly.

Yours,

Astrid Kerkweg

––––––––––––––––––––––––––––––––

---

## Author Comment (AC1) · 22 Apr 2016

Many thanks for the comments regarding the title of the manuscript. We are happy to follow the title format whereby the title takes the form:

**OMIP contribution to CMIP6: experimental and diagnostic protocol for the physical component of the Ocean Model Intercomparison Project**

We will make the title change in the revised manuscript.

However, in agreement with the CMIP Panel, we will not include version numbering as part of the title.

[Figure]

Sincerely, Stephen Griffies for the co-authors

---

## Short Comment (SC2) · 29 Apr 2016

There is an inconsistency in the requested shape of variables hfx and hfy: In table 3 the requested shape is XYZ indicating a 3D array but in the text describing the variables you say "We follow CMIP5 in requesting the vertically integrated x-ward and y-ward heat transport from all ocean processes."

From our perspective (using NEMO) I think either is possible but it would require a small amount of effort to change the code to output 3D fields instead of 2D depth integrated fields.
* * *

---

## Author Comment (AC2) · 2 May 2016

Hi,

Thanks for the comment. There is indeed an inconsistency in the hfx and hfy diagnostic requests. Upon further consultation with the group originally advocating for 3d terms, we have now agreed to return to the CMIP5 approach, whereby hfx and hfy are the depth integrated heat transports. We are thus not asking for depth information. The revised manuscript will reflect this change.

Thanks again, Stephen Griffies

---

## Referee Comment (RC1) · F. O. Bryan (Referee) · 9 May 2016

This manuscript provides an in-depth and lucid description and justification for the physical ocean quantities that are requested for the CMIP6 archives, as well as the experimental protocol for the OMIP component of CMIP6. The later is relatively brief, but supported by the recent CORE-II analysis papers. The overwhelming majority of the material is devoted to the former. It is my understanding that the protocol and variables lists have been negotiated through various international panels and CMIP governance processes, and are not really open to criticism in this review of the manuscript. Rather, I have been directed to focus my review on issues of clarity of presentation. that said, in the comments below, I do question some omissions.

[Figure]

Overall, the manuscript is an outstanding piece of work and reflects the depth of expertise represented in the author list, and especially of the first author. The level of detail makes the manuscript rise to a level that can serve as a standard reference in ocean modeling. I expect that it will be used by the ocean and climate modeling communities for purposes well outside and beyond the scope of CMIP6.

I have only one comment of substance. This is in regards to the issue of remapping the output. The authors make the case that for the purpose of comparative analyses, it is critical to have the output remapped to a common grid, and they encourage the use of a standard 1 deg. grid commensurate with that used in the World Ocean Atlas (notably, this is coarser than the native grids of many CMIP6 models). I have no fundamental objection to this position. Where my concern arises is the equally strong emphasis of the authors on being able to diagnose and test for exact conservation of mass, heat, salt, etc. In the presence of complex topography, these two objectives run counter to one another. In particular, any remapping will necessarily require the definition of new land-mask and topography fields, inevitably different from those of the native model grid. It is not trivial, and perhaps not even possible, to retain conservation when the surface area and volume of the ocean differ on the two grids. Perhaps the authors envision use of partial cell type ideas to recover such properties from both grids. However, there is ambiguity in how the partial cell volumes and areas should be partitioned between surface area and cell thickness/mass. Indeed, for the case of variable cell thickness/mass models, I wonder if all remapping (vertical + horizontal) would need to be done online to guarantee conservation? If the authors are going to push for these twin constraints, then they need to provide more complete guidance on how they are to be mutually satisfied. At several points in the manuscript they defer to the unpublished Balaji et al manuscript (which I did not have access to), but I have my doubts whether this problem will be adequately addressed there.

Detailed Comments:

pg 7, line 9-10: WOA13v1 or WOA13v2 ?

[Figure]

pg 7, line 22: "an implied surface temperature restoring" a matter of semantics, but I think it would be better said as "a negative feedback on SST anomalies"

pg 9, line 21:-23: I would argue that for more complex manipulations, the analysis needs to be done on the native grid with the analysis output targeted to the common grid

pg 10, line 23-25: This is not clear. You are convolving temporal variability with spatial sampling. The spatial covariances of subgrid-scale (on the target grid) structure needs to be properly accounted for.

pg 11, line 3-4: See above. A locally conservative remapping of a variable does not necessarily guarantee global conservation if the global areas or volume change.

pg 12, line 3: what happens to native grid levels with depths greater than the 5500m max depth of the Levitus grid?

pg 12, line 23: did any of the final fields meet this criteria (I can't find any)

pg 10, line 9: This would seem to apply to all variables on the remapped grid

pg 19, line 8-10: This seems like an inconsistent level of detail. The region mask is going to be blurred on the remapped domain, and remapped staggered quantities will likely sit on the edge between to regions.

pg 19: section 4.9 This section needs considerable expansion to deal with the issue in my general comments

pg 22, line 25-28: How is vertical staggering to be handled with respect to recording cell volumes or thicknesses?

pg 27, line 7: "are not trustworthy" This is a bit presumptuous - the CMIP6 land ice models have not yet been assessed.

pg 31, line 5: "a measure of simulation drift" will also include a component of true,

forced low-frequency variability

pg 33, line 5-8: What is the rationale for diagnosing sub-daily variance in SST? Why not in surface velocity, SSH or SSS?

pg 40, table 3: why is wo (vertical velocity) excluded?

pg 42, line 16-17: only for the _steady_state_, rigid-lid, Boussinesq case

pg 44, line 15-16: "comparable to the model native grid" Why was the same specification not included for overturning streamfunction? Presumably this is the prescription for the "native" resolution with a further decimation to the spherical grid resolution?

pg 53, line 10: Why is river runoff prescribed as a surface (XY) flux rather than a lateral flux?

pg 60, line 17: Figure 1

pg 62, Table 9: Why is wind work excluded. Several variables related to energy dissipation are included.

---

## Referee Comment (RC2) · W. Weijer (Referee) · 20 May 2016

This paper documents the experimental protocol for the CMIP6 Ocean Model Inter-comparison Project (OMIP); as well as a recommended suite of diagnostics to analyze the ocean component of OMIP and other CMIP6 simulations. This is a very thorough paper that provides excellent guidance for modeling centers participating in the CMIP6 experiments, as well as a reference for analysts. I found the paper very well-written and well-documented. I have only trivial comments and a few corrections, see below, and so I recommend the paper be accepted with only minor modifications.

p. 21, footnote 10: Should g not be added as part of the archive?

p. 29, ll. 24-25, ". . .the first year. . .": Do you mean initial state instead?

Section 5.24: I often find the maximum mixed layer depth over a given averaging interval quite useful as well.

Section 6.7: So hfx and hfy will reflect total heat transport, not broken up in individual contributions?

Section 6.8: In the hfbasin diagnostics I don't see the contribution by the resolved flow called out. Is the idea that this can be calculated from the difference between the total and parameterized contributions?

p. 44, l. 12: . . .should ALSO (?) compute. . .

p. 49, l. 6: componeNts

p. 51, l. 29: Goldsbrough

p. 68, l. 24: remove there

p. 79, l. 15: It is my understanding that Dukowicz & Smith's a free-surface formulation /does/ allow for changing surface layer thickness.

Appendix E: It's probably good to capitalize Kelvin and Celsius.

p. 105, l. 34: Leeuwen

p. 106, l. 8: Carson

---

## Short Comment (SC3) · 2 Jun 2016

Dear OMIP authors,

The CMIP Panel is undertaking a review of the CMIP6 GMD special issue papers to ensure a level of consistency in answering the key questions that were outlined in our request to submit a paper to all co-chairs of CMIP6-Endorsed MIPs. These questions are outline in the overview paper (Eyring et al, GMD, 2016) and the relevant section is summarised below:

'Each of the 21 CMIP6-Endorsed MIPs is described in a separate invited contribution to this Special Issue. These contributions will detail the goal of the MIP and the major scientific gaps the MIP is addressing, and will specify what is new compared to CMIP5 and previous CMIP phases. The contributions will include a description of the experimental design and scientific justification of each of the experiments for Tier 1 (and possibly beyond), and will link the experiments and analysis to the DECK and CMIP6 historical simulations. They will additionally include an analysis plan to fully justify the resources used to produce the various requested variables, and if the analysis plan is to compare model results to observations, the contribution will highlight possible model diagnostics and performance metrics specifying whether the comparison entails any particular requirement for the simulations or outputs (e.g. the use of observational simulators). In addition, possible observations and reanalysis products for model evaluation are discussed and the MIPs are encouraged to help facilitate their use by contributing them to the obs4MIPs/ana4MIPs archives at the ESGF (see Section 3.3). In some MIPs additional forcings beyond those used in the DECK and CMIP6 historical simulations are required, and these are described in the respective contribution as well.'

We very much welcome the OMIP contribution and the hugely valuable detailing of the diagnostic output that you currently cover in sections 3-8. OMIP is clearly providing leadership on the ocean diagnostics that will provide an important protocol for CMIP6.

However we would like to suggest that for consistency with the other papers these sections (3-8) are documented in an appendix rather than in the main body of the paper.

Additionally, we would like to see some more detail on some of the issues raised above, notably;

1. More discussion on the goal of OMIP in CMIP6 and what science gaps it is attempting to fill. Currently you do not mention the 3 science questions or the WCRP grand-challenges around which CMIP6 is organised. It would seem clear that OMIP is focussed on 'understanding systematic biases' and hence would be good to include this and also discuss what OMIP is hoping to achieve that is new.

2. The discussion of the CORE II experiments is not framed in terms of the CMIP6 ideas of MIPs having tiered experiments. Again for consistency it would be good to include this in section 2.2

3. All MIPs have been asked to demonstrate connectivity to the DECK experiments and the CMIP6 historical simulations as one of the 10 endorsement criteria (see Table 1 in Eyring et al., 2016). Please document this for OMIP.

4. You have not provided an analysis plan for the science community engaged in OMIP. How are you going to use the experiments and diagnostics? Are you committing to analyse all the data that you are requesting (or can you point to other MIPs that will do so)?

5. You make a strong argument about the potential to compare the modelling data with new observations. Can you highlight diagnostics that will enable this comparison – do they make any particular demand on the model outputs? Are/Could the new observations you describe in section 1.1 be made easily available to the modelling community (e.g. through Obs4MIPs?)

We hope you agree that some level of consistency across the MIP papers in this special issue is valuable and that the above suggestions can be accommodated in your paper.

Other comments:

- The first sentence in the data availability section seems wrong "The model output from the DECK and CMIP6 historical simulations described in this paper will be distributed through the Earth System Grid Federation (ESGF) with digital object identifiers (DOIs) assigned." This paper is not describing the DECK and CMIP6 historical simulations. Please change. The data availability section could also be shortened. The details on the WIP contribution seems unnecessary here.

- Somewhere at the beginning of the manuscript it should say that this is one of the 21 CMIP6-Endorsed MIPs.

- For the diagnostic sections (3-8), what is the link to the CMIP6 data request? Perhaps you need to clarify where is the definitive documentation of what is actually being output from the models (e.g. via a link to the actual data request) and to reference the GMD paper by Martin Jukes?

With many thanks for your ongoing efforts in the CMIP6 process.

The CMIP Panel

---

## Author Comment (AC4) · 27 Jun 2016

We sincerely thank Wilbert Weijer for his comments and for his encouragement. Here are our responses.

**Reviewer comment**:

This paper documents the experimental protocol for the CMIP6 Ocean Model Intercomparison Project (OMIP); as well as a recommended suite of diagnostics to analyze the ocean component of OMIP and other CMIP6 simulations. This is a very thorough paper that provides excellent guidance for modeling centers participating in the CMIP6 experiments, as well as a reference for analysts. I found the paper very well-written

and well-documented. I have only trivial comments and a few corrections, see below, and so I recommend the paper be accepted with only minor modifications.

**Author response**: Many thanks for your encouraging comments.

**Reviewer comments and author responses**

p. 21, footnote 10: Should g not be added as part of the archive?

–>We will consider doing so in the future. The issue concerns the role of static equilibrium sea level and tides, each of which are a consideration for future CMIPs. So adding gravitational acceleration to the CF diagnostic suite is not as trivial as one may think/hope.

p. 29, ll. 24-25, ". . .the first year. . .": Do you mean initial state instead?

–>We mean the first year of the simulation, which is generally taken at the end of a spin up. This point has been clarified in the revised draft.

Section 5.24: I often find the maximum mixed layer depth over a given averaging interval quite useful as well.

–>We agree, and have added these two fields (max and min MLD for a month) to the diagnostic request.

Section 6.7: So hfx and hfy will reflect total heat transport, not broken up in individual contributions?

–>Correct, as detailed in this section.

Section 6.8: In the hfbasin diagnostics I don't see the contribution by the resolved flow called out. Is the idea that this can be calculated from the difference between the total and parameterized contributions?

–>correct.

p. 44, l. 12: . . .should ALSO (?) compute. . .

–>agree and corrected

p. 49, l. 6: componeNts

–>corrected

p. 51, l. 29: Goldsbrough

–>corrected

p. 68, l. 24: remove there

–>corrected

p. 79, l. 15: It is my understanding that Dukowicz  Smith's a free-surface formulation /does/ allow for changing surface layer thickness.

–>From equations (6) and (7) of Dukowicz and Smith (1994), their algorithm assumes a linearized free surface formulation, in which the top grid cell has an upper surface strictly at z=0 rather than at z=eta. Therefore, POP cannot conservatively incorporate real water fluxes; it must instead use virtual salt fluxes.

Appendix E: It's probably good to capitalize Kelvin and Celsius.

–>According to http://www.nist.gov/pml/wmd/metric/writing-metric.cfm we should write Celsius in capital, but Kelvin is lowercse.

p. 105, l. 34: Leeuwen

–>corrected

p. 106, l. 8: Carson

–>Thanks for identifying the typo. We corrected the reference.

---

## Author Response (AR1)

**Author responses to reviewer comments**

We sincerely thank the editor (Robert Marsh), the three reviewers (Frank Bryan, Wilbert Weijer, and Cath Senior), as well as Astrid Kerkweg and Tim Graham for their efforts to help improve our manuscript detailing the OMIP contribution to CMIP6. We are pleased that the original draft was largely agreeable to the reviewers, and we are happy to address each of the issues they identified. Here is a summary of what we have provided.

- We have responded online to the individual comments from the reviewers. We also include those responses in the present document.

- The first comment from the CMIP6 panel (represented by Cath Senior) requested a substantial reorganization of the sections in the manuscript. Doing so brings the OMIP manuscript in line with other contributions to the CMIP6 special issue. We have thus followed her suggestion.

  As a result, all of the diagnostics details are now placed into appendices. Doing so has modified nearly all of the text so that a "diff" is no longer able to clearly identify relevant edits to the manuscript. Hence, the typical mark-up software packages fail. Consequently, we have decided to not submit a marked-up draft. We communicated this issue with the editor, Robert Marsh, on 09June2016. We trust the editor will find it sufficient to read our detailed responses to the reviewer comments to identify how we have responded to reviewr comments.

- Based on comments from colleagues, we have added a figure to illustrate the geographical location of the 16 mass transport sections to be saved for OMIP (Figure 1 in the revised draft).

- In revising the draft, we have updated references where needed, corrected typos, and further refined presentation where prompted by reviewer comments. Each of these edits has resulted in a more solid manuscript for publication in GMD.

**1 Responses to Robert Marsh**

- **Reviewer comment**: Also, following up on footnote 23 (p.59), I agree with the statement, but I can add that the latent heat of iceberg melting (as a negative ocean heat flux) is now included in NEMO-ICB, as implemented at the UK Met Office for CMIP6.

- **Author comment**: Thanks for letting us know about the update to the NEMO-ICB. We have thus modified the footnote to read:

  "In testing the NEMO-ICB iceberg model, Marsh et al. (2015) considered icebergs with zero heat capacity. However, heat conservation in the coupled climate system requires that the latent heat of fusion used to create the ice on land must be given up by the liquid ocean as the icebergs melt. Consequently, icebergs with zero heat capacity should not be used in a coupled climate simulation for CMIP6. In fact, the nonzero heat capacity of icebergs is now included in the NEMO-ICB for use in CMIP6 (Robert Marsh, personal communication)."

**2   Responses to Astrid Kerkweg**

- **Reviewer comment**:  In agreement with the CMIP6 panel members, the Executive editors of GMD would like to establish a common naming convention for the titles of the CMIP6 experiment description papers. The title of CMIP6 papers should include both the acronym of the MIP, and CMIP6, so that it is clear this is a CMIP6-Endorsed MIP.

   Additionally, we strongly recommend to add a version number to the MIP description. The reason for the version numbers is so that the MIP protocol can be updated later, normally in a second short paper outlining the changes. See, for example: http://www.geosci-model-dev.net/special_issue11.html,

   Good formats for the title include: "XYZMIP (v1.0) contribution to CMIP6: Name of project" or "Name of Project (XYZMIP v1.0) contribution to CMIP6"

   If you want to include a more descriptive title, the format could be along the lines of, "XYZMIP (v1.0) contribution to CMIP6: Name of project - descriptive title" or "Name of Project (XYZMIP v1.0) contribution to CMIP6: descriptive title."

   When you revise your manuscript, please correct the title of your manuscript accordingly.

- **Author response**:

   Many thanks for the comments regarding the title of the manuscript. We are happy to follow the title format whereby the title takes the form:

   *OMIP contribution to CMIP6: experimental and diagnostic protocol for the physical component of the Ocean Model Intercomparison Project*

   We have made that title change in the revised manuscript. However, in agreement with the CMIP Panel, we will not include version numbering as part of the title.

**3   Responses to Tim Graham**

- **Reviewer comment**:

   There is an inconsistency in the requested shape of variables hfx and hfy: In table 3 the requested shape is XYZ indicating a 3D array but in the text describing the variables you say "We follow CMIP5 in requesting the vertically integrated x-ward and y-ward heat transport from all ocean processes."

   From our perspective (using NEMO) I think either is possible but it would require a small amount of effort to change the code to output 3D fields instead of 2D depth integrated fields.

- **Author response**: There is indeed an inconsistency in the hfx and hfy diagnostic requests. Many thanks for identifying the problem. Upon further consultation with the group originally advocating for 3d terms, we have now agreed to return to the CMIP5 approach, whereby hfx and hfy are the depth integrated heat transports. We are thus not asking for depth information. The revised manuscript reflects this change.

**4   Responses to Frank Bryan**

**General comments**

- **Reviewer comment**:

This manuscript provides an in-depth and lucid description and justification for the physical ocean quantities that are requested for the CMIP6 archives, as well as the experimental protocol for the OMIP component of CMIP6. The later is relatively brief, but supported by the recent CORE-II analysis papers. The overwhelming majority of the material is devoted to the former. It is my understanding that the protocol and variables lists have been negotiated through various international panels and CMIP governance processes, and are not really open to criticism in this review of the manuscript. Rather, I have been directed to focus my review on issues of clarity of presentation. That said, in the comments below, I do question some omissions.

Overall, the manuscript is an outstanding piece of work and reflects the depth of expertise represented in the author list, and especially of the first author. The level of detail makes the manuscript rise to a level that can serve as a standard reference in ocean modeling. I expect that it will be used by the ocean and climate modeling communities for purposes well outside and beyond the scope of CMIP6.

- **Author response**:

Many thanks for your encouraging comments. We too hope that this manuscript will be of use for the broader ocean and climate modelling communities, even beyond CMIP6.

**Issues with regridding**

- **Reviewer comment**:

I have only one comment of substance. This is in regards to the issue of remapping the output. The authors make the case that for the purpose of comparative analyses, it is critical to have the output remapped to a common grid, and they encourage the use of a standard 1 deg. grid commensurate with that used in the World Ocean Atlas (notably, this is coarser than the native grids of many CMIP6 models). I have no fundamental objection to this position.

Where my concern arises is the equally strong emphasis of the authors on being able to diagnose and test for exact conservation of mass, heat, salt, etc. In the presence of complex topography, these two objectives run counter to one another. In particular, any remapping will necessarily require the definition of new land-mask and topography fields, inevitably different from those of the native model grid. It is not trivial, and perhaps not even possible, to retain conservation when the surface area and volume of the ocean differ on the two grids. Perhaps the authors envision use of partial cell type ideas to recover such properties from both grids. However, there is ambiguity in how the partial cell volumes and areas should be partitioned between surface area and cell thickness/mass. Indeed, for the case of variable cell thickness/mass models, I wonder if all remapping (vertical + horizontal) would need to be done online to guarantee conservation?

If the authors are going to push for these twin constraints, then they need to provide more complete guidance on how they are to be mutually satisfied. At several points in the manuscript they defer to the unpublished Balaji et al manuscript (which I did not have access to), but I have my doubts whether this problem will be adequately addressed there.

- **Author Response**:

We were originally hopeful that a robust remapping method could be identified during the many months developing this manuscript. Alas, no such tool is available or has been championed. We therefore conclude, as suggested by the reviewer, that it is not sensible to strongly encourage regridding Priority=1 output to a spherical grid. We have taken a more modest perspective into the revised draft, and modified the text where appropriate (see in particular Sections 3.1 and A3).

The new draft continues to acknowledge the dilemna facing the community (i.e., ease of analysis versus integrity of the model data). We now, however, more strongly emphasize native grid diagnostics as the common element to be saved across all model submissions. Regridding, if desired, will be facilitated only for scalar fields by area weights requested for the grids through the WIP contribution to this special issue (Balaji et al (2016) in prep). We now make these points clear in the revised draft (see Sections 3.1 and A3).

**Detailed reviewer comments and author responses**

- pg 7, line 9-10: WOA13v1 or WOA13v2 ?

  –>WOA13v2 is now noted.

- pg 7, line 22: "an implied surface temperature restoring" a matter of semantics, but I think it would be better said as "a negative feedback on SST anomalies"

  –>Agree; wording changed as suggested.

- pg 9, line 21:-23: I would argue that for more complex manipulations, the analysis needs to be done on the native grid with the analysis output targeted to the common grid

  –>Text is now fully rewritten, with new version consistent with this comment (now in Appendix A3).

- pg 10, line 23-25: This is not clear. You are convolving temporal variability with spatial sampling. The spatial covariances of subgrid-scale (on the target grid) structure needs to be properly accounted for.

  –>We agree. The questionable text has now been removed given re-focus on native grids, and computation of products online rather than offline.

- pg 11, line 3-4: See above. A locally conservative remapping of a variable does not necessarily guarantee global conservation if the global areas or volume change.

  –>We agree. We now recommend saving native data, with new discussion noting problems with conservation when regridding.

- pg 12, line 3: what happens to native grid levels with depths greater than the 5500m max depth of the Levitus grid?

  –>This is indeed an ambiguous situation. We offer no firm recommendation, but do identify the issue.

- pg 12, line 23: did any of the final fields meet this criteria (I can't find any)

  –>Agree; this is an obsolete point, now removed in revised draft.

- pg 10, line 9: This would seem to apply to all variables on the remapped grid

  –>Text has been removed in revised draft.

- pg 19, line 8-10: This seems like an inconsistent level of detail. The region mask is going to be blurred on the remapped domain, and remapped staggered quantities will likely sit on the edge between to regions.

  –>Agree for the sphere, but useful if native. Point is now clarified.

- pg 19: section 4.9 This section needs considerable expansion to deal with the issue in my general comments

  –>As noted above, we are no longer emphasizing spherical regridding. Instead, we are relying on Balaji et al (2016) to provide details for the area weights required for regridding scalars. These details are best presented in Balaji et al (2016) since the WIP is coordinating that work.

- pg 22, line 25-28: How is vertical staggering to be handled with respect to recording cell volumes or thicknesses?

  –>We are only asking for the tracer cell thickness, so no need to consider vertical staggering between velocity and tracer.

- pg 27, line 7: "are not trustworthy". This is a bit presumptuous - the CMIP6 land ice models have not yet been assessed.

  –>Agree; text is now modified to reflect this point, and to acknowledge that some groups may have an ice sheet model. We also point to the ice sheet model comparison project now part of CMIP6 (Nowicki et al. 2016).

- pg 31, line 5: "a measure of simulation drift" will also include a component of true, forced low-frequency variability.

  –>Agree, with text now added to reflect this point.

- pg 33, line 5-8: What is the rationale for diagnosing sub-daily variance in SST? Why not in surface velocity, SSH or SSS?

  –>We presently request tossq (SST*SST) on daily and monthly time sampling. The daily sampling is requested to help diagnose Tropical Instability Waves (TIWs). We have not had requests for similar sampling of SSS or SSH, although note that daily mean SSS is requested.

- pg 40, table 3: why is wo (vertical velocity) excluded?

  –>This was an oversight. This field is now been requested in Section J4.

- pg 42, line 16-17: only for the steady state, rigid-lid, Boussinesq case

  –>Agree, and corrected.

- pg 44, line 15-16: "comparable to the model native grid". Why was the same specification not included for overturning streamfunction? Presumably this is the prescription for the "native" resolution with a further decimation to the spherical grid resolution?

  —>This is admittedly imprecise; it remains unclear what is the best and most practical approach. Text to this effect has been added.

- pg 53, line 10: Why is river runoff prescribed as a surface (XY) flux rather than a lateral flux?

  –>Agree; text changed to allow for lateral flux with friver now XYZ field in general.

- pg 60, line 17: Figure 1

  –>corrected

- pg 62, Table 9: Why is wind work excluded. Several variables related to energy dissipation are included.

  –>As no one has requested it, we did not include it. Additionally, we are not requesting budget terms for the kinetic energy. The energy dissipation terms that are requested are meant to diagnose impact from subgrid scales. As mentioned in the Closing Comments section, we anticipate future CMIPs to address more budgets, beyond the new heat and salt budgets requested in CMIP6.

**5 Responses to Wilbert Weijer**

**General comments**

- **Reviewer comment**:

  This paper documents the experimental protocol for the CMIP6 Ocean Model Intercomparison Project (OMIP); as well as a recommended suite of diagnostics to analyze the ocean component of OMIP and other CMIP6 simulations. This is a very thorough paper that provides excellent guidance for modeling centers participating in the CMIP6 experiments, as well as a reference for analysts. I found the paper very well-written and well-documented. I have only trivial comments and a few corrections, see below, and so I recommend the paper be accepted with only minor modifications.

- **Author response**: Many thanks for your encouraging comments.

**Detailed reviewer comments and author responses**

- p. 21, footnote 10: Should g not be added as part of the archive?

  –>We have discussed this diagnostic among a sub-group, and have concluded that adding a precise definition of the gravitational acceleration to the CF diagnostic suite is not trivial. The issue concerns the role of static equilibrium sea level and tides, each of which are a consideration for future CMIPs. When considering such processes, the gravitational acceleration is a function of space and time. We suggest that diagnosing the gravitational acceleration for CMIP should occur when such issues are routinely considered in climate simulations.

- p. 29, ll. 24-25, ". . .the first year. . .": Do you mean initial state instead?

  –>We mean the first year of the simulation, which is generally taken at the end of a spin up. This point has been clarified in the revised draft.

- Section 5.24: I often find the maximum mixed layer depth over a given averaging interval quite useful as well.

  –>We agree, and have added these two fields (max and min MLD for a month) to the diagnostic request.

- Section 6.7: So hfx and hfy will reflect total heat transport, not broken up in individual contributions?

–>Correct, as detailed in this section.

- Section 6.8: In the hfbasin diagnostics I don't see the contribution by the resolved flow called out. Is the idea that this can be calculated from the difference between the total and parameterized contributions?

  –>correct.

- p. 44, l. 12: . . .should ALSO (?) compute. . .

  –>agree and corrected

- p. 49, l. 6: componeNts

  –>corrected

- p. 51, l. 29: Goldsbrough

  –>corrected

- p. 68, l. 24: remove there

  –>corrected

- p. 79, l. 15: It is my understanding that Dukowicz & Smith's free-surface formulation /does/ allow for changing surface layer thickness.

  –>From equations (6) and (7) of Dukowicz and Smith (1994), their algorithm assumes a linearized free surface formulation, in which the top grid cell has an upper surface strictly at $z = 0$ rather than at $z = \eta$. Therefore, POP cannot conservatively incorporate real water fluxes; it must instead use virtual salt fluxes.

- Appendix E: It's probably good to capitalize Kelvin and Celsius.

  –>According to http://www.nist.gov/pml/wmd/metric/writing-metric.cfm we should write Celsius in capital, but kelvin is lowercse.

- p. 105, l. 34: Leeuwen

  –>corrected

- p. 106, l. 8: Carson

  –>Thanks for identifying the typo. We corrected the reference.

**6   Responses to Cath Senior and the CMIP6 panel**

**Move diagnostics into appendices**

- **Reviewer comment**:

  We very much welcome the OMIP contribution and the hugely valuable detailing of the diagnostic output that you currently cover in sections 3-8. OMIP is clearly providing leadership on the ocean diagnostics that will provide an important protocol for CMIP6. However we would like to suggest that for consistency with the other papers these sections (3-8) are documented in an appendix rather than in the main body of the paper.

- **Author response**:

  Many thanks for your encouraging remarks. We agree to move the diagnostics details (located in Sections 3 through 11 rather than 3 through 8) into appendices. We have done so in the revised draft.

**More discussion on the goal of OMIP in CMIP6**

- **Reviewer comment**:

  Additionally, we would like to see some more detail on some of the issues raised above, notably;

  More discussion on the goal of OMIP in CMIP6 and what science gaps it is attempting to fill. Currently you do not mention the 3 science questions or the WCRP grand-challenges around which CMIP6 is organised. It would seem clear that OMIP is focussed on "understanding systematic biases" and hence would be good to include this and also discuss what OMIP is hoping to achieve that is new.

- **Author response**:

  Agree. These points are now discussed in the abstract and in the introduction. In particular, we now state:

  "OMIP addresses CMIP6 science questions investigating the origins and consequences of systematic model biases. It does so by providing a framework for evaluating (including assessment of systematic biases), understanding, and improving ocean, sea ice, tracer, and biogeochemical components of climate and earth system models contributing to CMIP6. Among the WCRP Grand Challenge in climate science (GCs), OMIP primarily contributes to the regional sea-level rise and near-term (climate / decadal) prediction GCs."

**OMIP framed in terms of CMIP6**

- **Reviewer comment**:

  The discussion of the CORE II experiments is not framed in terms of the CMIP6 ideas of MIPs having tiered experiments. Again for consistency it would be good to include this in section 2.2

- **Author response**:

  Agree. These points are now discussed. Namely, the present paper focuses on a Tier 1 physical experimental protocol. The companion paper from Orr et al (2016) discusses the inert chemistry and the biogeochemical elements of OMIP, and propose a Tier 1 and a Tier 2 simulation.

**OMIP connectivity to DECK**

- **Reviewer comment**:

  All MIPs have been asked to demonstrate connectivity to the DECK experiments and the CMIP6 historical simulations as one of the 10 endorsement criteria (see Table 1 in Eyring et al., 2016). Please document this for OMIP.

- **Author response**:

  We now more clearly make note of the connection of OMIP to CMIP6 goals in the abstract and introduction, where we now state (as noted above):

  "OMIP addresses CMIP6 science questions investigating the origins and consequences of systematic model biases. It does so by providing a framework for evaluating (including assessment of systematic biases), understanding, and improving ocean, sea ice, tracer, and biogeochemical components of climate and earth system models contributing to CMIP6. Among the WCRP Grand Challenge in climate science (GCs), OMIP primarily contributes to the regional sea-level rise and near-term (climate / decadal) prediction GCs."

**Analysis plans**

- **Reviewer comment**:

  You have not provided an analysis plan for the science community engaged in OMIP. How are you going to use the experiments and diagnostics? Are you committing to analyse all the data that you are requesting (or can you point to other MIPs that will do so)?

- **Author response**:

  In Section 3.2 of the revised draft, we detail areas of ocean and climate science having effected the design of the diagnostics. We here list five CMIP6 sanctioned MIPs (OMIP, FAFMIP, C4MIP, HighResMIP, DCPP) that have either proposed diagnostics, or have indicated their needs. We also list five science communities (GSOP, AMOC, Southern Ocean Regional Panel, Ecosystem community, Ocean Mixing) that have representatives on the author list of this paper, each having expressed the need for various diagnostics to further their science research using CMIP6 ocean fields.

  More generally, our aim is to design a diagnostic protocol that enables a wide suite of ocean related research to emerge from CMIP6. Analysis plans for this research take various shapes, from detailed projects ongoing by various OMIP authors, to an understanding that some of the best science from CMIP6 will emerge from questions that have yet to be asked. We trust that if these diagnostics are archived for CMIP6, there will be extensive science enabled in the coming years.

**Connections to observations**

- **Reviewer comment**:

  You make a strong argument about the potential to compare the modelling data with new observations. Can you highlight diagnostics that will enable this comparison - do they make any particular demand on the model outputs? Are/Could the new observations you describe in section 1.1 be made easily available to the modelling community (e.g. through Obs4MIPs?)

- **Author response**:

  We comment on this point in the introduction. Most notably, improvements to the measurement of ocean heat enable a huge suite of key research into the energy balance of the climate system. These observations are being made available to the modelling community through the development of databases and their analyses. Obs4MIPS is aware of this work

and is endeavoring to bring the data into the hands of modellers (note, Peter Gleckler and Karl Taylor are co-authors of this OMIP document, and leads on Obs4MIPs).

**Data availability edits**

- **Reviewer comment**:

  The first sentence in the data availability section seems wrong "The model output from the DECK and CMIP6 historical simulations described in this paper will be distributed through the Earth System Grid Federation (ESGF) with digital object identifiers (DOIs) assigned." This paper is not describing the DECK and CMIP6 historical simulations. Please change. The data availability section could also be shortened. The details on the WIP contribution seems unnecessary here.

- **Author response**:

  Yes, we agree and have edited the text.

**OMIP is an endorsed MIP**

- **Reviewer comment**:

  Somewhere at the beginning of the manuscript it should say that this is one of the 21 CMIP6-Endorsed MIPs.

- **Author response**: Agree and done.

**Link to CMIP6 data request**

- **Reviewer comment**:

  For the diagnostic sections (3-8), what is the link to the CMIP6 data request? Perhaps you need to clarify where is the definitive documentation of what is actually being output from the models (e.g. via a link to the actual data request) and to reference the GMD paper by Martin Juckes?

- **Author response**:

  We are working closely with Martin Juckes. His data request spreadsheets include all of the diagnostics requested in the OMIP manuscript. We indicate such to be the case in the revised version (start of Section 3). However, he has informed us that there is no Juckes et al. manuscript yet to cite, so we cannot do so at this time. Instead, we cite "Martin Juckes, personal communication".